# UNSUPERVISED DOMAIN ADAPTATION WITH AN UNOBSERVABLE SOURCE SUBPOPULATION

## ABSTRACT

We study an unsupervised domain adaptation problem where the source domain consists of subpopulations defined by the binary label $Y$ and a binary background (or environment) $A$. We focus on a challenging setting in which one such subpopulation in the source domain is unobservable. Naively ignoring this unobserved group can result in biased estimates and degraded predictive performance. Despite this structured missingness, we show that the prediction in the target domain can still be recovered. Specifically, we rigorously derive both background-specific and overall prediction models for the target domain. For practical implementation, we propose the distribution matching method to estimate the subpopulation proportions. We provide theoretical guarantees for the asymptotic behavior of our estimator, and establish an upper bound on the prediction error. Experiments on both synthetic and real-world datasets show that our method outperforms the naive benchmark that does not account for this unobservable source subpopulation.

## 1 INTRODUCTION

Unsupervised domain adaptation (UDA) (Kouw & Loog, 2019) addresses the challenge of transferring predictive models from a labeled source domain to an unlabeled target domain under distributional shifts (Koh et al., 2021; Sagawa et al., 2022). In this area, research methods aim to reduce domain discrepancy by aligning feature distributions, using statistical measures such as maximum mean discrepancy (MMD) (Tzeng et al., 2014) and higher-order moment matching (HoMM) (Chen et al., 2020). Deep adaptation frameworks, such as deep adaptation network (DAN) (Long et al., 2015) and domain-adversarial neural network (DANN) (Ganin et al., 2016), are also popularly used due to their strong empirical performance. There are also other approaches that integrate reconstruction objectives to disentangle domain-invariant and domain-specific components (Ghifary et al., 2016). These approaches often assume access to a representative and diverse set of source examples. However, real-world datasets may violate this assumption in systematic and non-random ways.

In this work, we focus on a more challenging and practically relevant UDA setting where a *structured subpopulation is entirely missing from the source domain*. Specifically, we consider binary label $Y$ and a binary background or environment variable $A$, and study the case where one subpopulation—defined by a particular combination of $Y$ and $A$—is unobserved in the source. This structured missingness is not merely a sampling artifact, but often reflects real-world constraints in data collection. For instance, in the widely studied Waterbirds dataset (Sagawa et al., 2019), waterbirds ($Y = 1$) photographed in water environments ($A = 1$) can be rare or entirely absent due to the difficulty of capturing such images in the wild. This issue arises in many other disciplines as well. In healthcare, certain patient subgroups—defined jointly by disease status and demographics—may be underrepresented or absent in historical datasets due to restrictive inclusion criteria or changes in clinical practice over time. When such models are applied to broader populations, unobserved subgroups can suffer from systematic mispredictions. This structured missingness (Mitra et al., 2023) fundamentally changes some statistical properties when comparing the source and target domains, and, if unaddressed, can lead to severely biased estimation and unreliable prediction in the target domain. These structured gaps pose new challenges that are not adequately addressed by conventional UDA techniques, which motivates our work.

To tackle this challenge, we develop a theoretical framework that accounts for the structured absence of a subpopulation, such as $(Y = 1, A = 1)$, in the source domain. Our key idea is to model how

prediction in the target domain can still be recovered by relating it to the observable parts of the source and target data. Under a mild assumption that the distribution of features $X$ given $(Y, A)$ stays the same across domains, we derive closed-form expressions for making accurate predictions in the target domain. These expressions depend on the proportions of different subgroups in the target, which are unknown. To estimate them, we propose a practical method based on distribution matching that avoids modeling complex feature distributions directly. Specifically, we frame the problem as estimating finite-dimensional mixture proportions under structured conditional invariance, and propose a KL-divergence-based objective that can be optimized using only observable quantities. We also provide theoretical guarantees, showing that our approach yields statistically consistent estimates and deriving upper bounds on the prediction error of the resulting target-domain classifiers. Overall, our framework provides the first rigorous characterization of model adaptation under structured subpopulation absence, and enables robust domain adaptation in such a challenging scenario.

We validate our approach through experiments on both synthetic and real-world datasets. We simulate domain adaptation scenarios where one subpopulation is systematically excluded from the source data and evaluate our method against baseline approaches that do not account for this missing group. Across a range of settings, our method consistently achieves higher accuracy and F1 scores, particularly on the subpopulation absent from the source. These results highlight the practical value of explicitly modeling structured missingness and demonstrate that our approach leads to more reliable predictions in the target domain. To summarize, this paper makes the following novel contributions:

- We consider a new unsupervised domain adaptation setting where an entire label-background subpopulation is missing from the source domain, a scenario motivated by real-world data collection constraints.

- We develop a theoretical framework that enables accurate prediction in the target domain by estimating subpopulation proportions through distribution matching, and we provide rigorous guarantees and error bounds for our method.

- We demonstrate the effectiveness of our approach on both synthetic and real-world datasets. Our method outperforms standard baselines that ignore structured missingness, particularly in recovering performance on the unobserved subpopulation.

## 2 PROBLEM SETUP AND NOTATION

In our UDA setting, $Y \in \{0, 1\}$ denotes the binary label, which is observed in the source domain but not in the target. Let $A \in \{0, 1\}$ be a binary background or environment variable and $\mathbf{X} \in \mathbf{R}^q$ a vector of all other attributes. Let $R \in \{0, 1\}$ be a domain indicator, with $R = 1$ corresponding to the source and $R = 0$ to the target. In our notation, we consistently use the order of $(R, Y, A)$ for indicator function $I_{\{\cdot\}}$, sample size $n_{\{\cdot\}}$, and population probability $p_{\{\cdot\}}$.

We define $\pi = \mathrm{pr}(R = 1)$. For $y = 1, 0$, $a = 1, 0$, we define $\alpha_{ya} = \mathrm{pr}(Y = y, A = a \mid R = 1)$, and $\beta_{ya} = \mathrm{pr}(Y = y, A = a \mid R = 0)$. For clarity, the total source sample size is $n_1 = n_{101} + n_{110} + n_{100}$, and the target sample size is $n_0 = n_{0\cdot1} + n_{0\cdot0}$, so that the total sample size is $n = n_1 + n_0$. Table 1 summarizes the observed data structure and key notation.

Table 1: Data structure and key notation used throughout the paper.

| | $R$ | $Y$ | $A$ | $\mathbf{X}$ | Sample Size | Proportion | Prediction Models |
|---|---|---|---|---|---|---|---|
| Source | 1 | 0 | 1 | ✓ | $n_{101}$ | $p_{101} = \alpha_{01}\pi$ | $\xi_1(\mathbf{x}) = \mathrm{pr}(Y = 1 \mid \mathbf{X} = \mathbf{x}, A = 1, R = 1)$ |
| | 1 | 1 | 0 | ✓ | $n_{110}$ | $p_{110} = \alpha_{10}\pi$ | $\xi_0(\mathbf{x}) = \mathrm{pr}(Y = 1 \mid \mathbf{X} = \mathbf{x}, A = 0, R = 1)$ |
| | 1 | 0 | 0 | ✓ | $n_{100}$ | $p_{100} = \alpha_{00}\pi$ | $\xi(\mathbf{x}) = \mathrm{pr}(Y = 1 \mid \mathbf{X} = \mathbf{x}, R = 1)$ |
| Target | 0 | ? | 1 | ✓ | $n_{0\cdot1}$ | $p_{0\cdot1} = (\beta_{11} + \beta_{01})(1 - \pi)$ | $\eta_1(\mathbf{x}) = \mathrm{pr}(Y = 1 \mid \mathbf{X} = \mathbf{x}, A = 1, R = 0)$ |
| | 0 | ? | 1 | ✓ | | | $\eta_0(\mathbf{x}) = \mathrm{pr}(Y = 1 \mid \mathbf{X} = \mathbf{x}, A = 0, R = 0)$ |
| | 0 | ? | 0 | ✓ | $n_{0\cdot0}$ | $p_{0\cdot0} = (\beta_{10} + \beta_{00})(1 - \pi)$ | $\eta(\mathbf{x}) = \mathrm{pr}(Y = 1 \mid \mathbf{X} = \mathbf{x}, R = 0)$ |
| | 0 | ? | 0 | ✓ | | | |

In our context, we have $\alpha_{10} + \alpha_{01} + \alpha_{00} = 1$, $\alpha_{11} = 0$, and $0 < \alpha_{10}, \alpha_{01}, \alpha_{00} < 1$. The parameters can be consistently estimated by

$$\widehat{\alpha}_{10} = n_{110}/n_1, \quad \widehat{\alpha}_{01} = n_{101}/n_1, \quad \widehat{\alpha}_{00} = n_{100}/n_1, \quad \widehat{\pi} = n_1/n. \tag{1}$$

More formally, $\alpha_{11} = 0$ is the following structured missingness condition:

$$\mathrm{pr}(Y = 1, A = 1 \mid R = 1) = 0. \tag{2}$$

Note that this assumption is made without loss of generality, as alternative combinations–such as $(Y = 0, A = 1)$, $(Y = 1, A = 0)$, or $(Y = 0, A = 0)$–can be similarly assumed to have zero probability. To characterize the distributional connection between the two domains, we impose a structured conditional invariance assumption:

$$p(\mathbf{X} \mid Y, A, R = 1) = p(\mathbf{X} \mid Y, A, R = 0) = p(\mathbf{X} \mid Y, A) \equiv p_{ya}(\mathbf{X}), \tag{3}$$

that is, the conditional distribution of features $\mathbf{X}$ given $(Y, A)$ remains the same across domains. This can be regarded as a conditional version, or, more nuanced version, of label shift where the marginal distribution of labels (now, the combination of both label and background) varies across domains (e.g., Du Plessis & Sugiyama, 2014; Garg et al., 2020; Iyer et al., 2014; Lipton et al., 2018; Nguyen et al., 2016; Tasche, 2017; Tian et al., 2023; Zhang et al., 2013). It indicates, conditional on background $A$, the label shift assumption holds. Mathematically, it states that $p(\mathbf{X}|Y, A, R = 1) = p(\mathbf{X}|Y, A, R = 0)$. It is equivalent to $p(R|\mathbf{X}, Y, A) = p(R|Y, A)$, the independence between $R$ and $\mathbf{X}$, conditional on $(Y, A)$. In practice, this assumption may be suitable in many applications. Below we give two examples to illustrate the rationality of this assumption. For instance, we aim to predict user clicks on advertisements for a new batch of users (target domain, $R = 0$) using historical data (source domain, $R = 1$). Conditional on the advertisement type $A$ and whether the user clicks $Y$, the distribution of browsing behavior features $\mathbf{X}$ is assumed to remain stable across time periods. This is because user clicks are fundamentally determined by ad content and user interests, not by the time period in which data are collected. As another example, suppose we have datasets from two hospitals ($R = 1$ indicates the source hospital and $R = 0$ indicates the target hospital). Here, $\mathbf{X}$ represents imaging features, $Y$ is the disease type, and $A$ denotes patient attributes such as gender or age group. Then, conditional on the disease type $Y$ and demographic attributes $A$, the distribution of imaging features $\mathbf{X}$ is expected to remain the same across hospitals. This is because imaging characteristics for a given disease and demographic group are not systematically altered by the hospital. The main difference between hospitals lies in sampling proportions rather than in conditional distributions.

This framework captures real-world scenarios in which a certain label-background subpopulation is absent from the source domain. For example, in the Waterbirds dataset, waterbirds on water backgrounds (label $Y = 1$, background $A = 1$) are rarely observed–or even completely absent–in the training set, making the adaptation to target domains particularly challenging. For illustration purposes, Table 2 below shows the three observed subpopulations in the source as well as the four subpopulations in the target in two real-world datasets.

Table 2: Illustrations in Waterbirds and CelebA datasets. Note that the $(Y = 1, A = 1)$ combination does not exist in the source domain but does in the target domain.

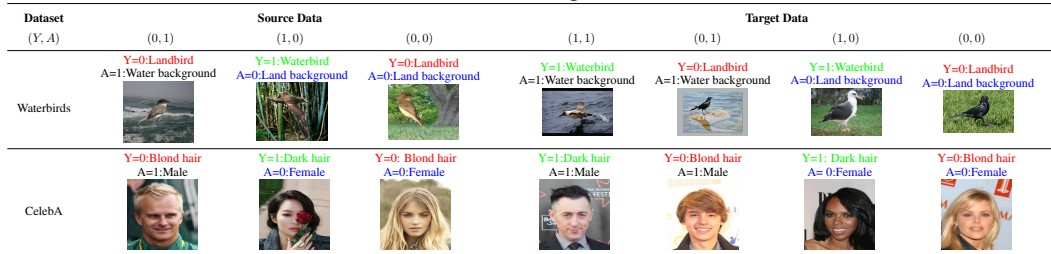

## 3 PROPOSED METHODOLOGY

Our goal in this work is to correctly identify and successfully implement, under our UDA setting, the two background-specific prediction models $\eta_1(\mathbf{x})$ and $\eta_0(\mathbf{x})$ and the overall prediction model $\eta(\mathbf{x})$, in the target domain. All of the three models were precisely defined in Table 1.

### 3.1 The naive benchmark

As the benchmark, one may naively apply the three source domain prediction models $\xi_1(\mathbf{x})$, $\xi_0(\mathbf{x})$, and $\xi(\mathbf{x})$ to the target. First, one can use the observed data to implement the overall source domain prediction model $\xi(\mathbf{x})$ and one background-specific prediction model $\xi_0(\mathbf{x})$ as

$$\xi(\mathbf{x}) = \mathrm{pr}(Y = 1 \mid \mathbf{x}, R = 1), \text{ and } \xi_0(\mathbf{x}) = \mathrm{pr}(Y = 1 \mid \mathbf{x}, R = 1, A = 0). \tag{4}$$

Afterwards, even though the subpopulation $(Y = 1, A = 1)$ is entirely absent in the source, still one can compute the other background-specific prediction model

$$\xi_1(\mathbf{x}) = \frac{\xi(\mathbf{x}) - \xi_0(\mathbf{x})\{1 - \tau_1(\mathbf{x})\}}{\tau_1(\mathbf{x})}, \text{ where} \tag{5}$$

$$\tau_1(\mathbf{x}) = \mathrm{pr}(A = 1 \mid \mathbf{X} = \mathbf{x}, R = 1), \tag{6}$$

can also be implemented using the observed data.

### 3.2 Model adaptation from source to target

The most challenging aspect of this work is to adapt the model for the $A = 1$ background since the component $(Y = 1, A = 1)$ is entirely absent in the source. Nevertheless, we can still correctly derive the three prediction models for the target domain, as shown below.

**Proposition 3.1.** *Define the model $\tau_0(\mathbf{x}) = pr(A = 1 \mid \mathbf{X} = \mathbf{x}, R = 0)$ and the model*

$$\kappa(\mathbf{x}) = pr(R = 1 \mid \mathbf{x}, A = 1), \tag{7}$$

*both of which can be implemented using the observed data in our UDA setting. Then the three prediction models in the target domain are given by:*

$$\eta_1(\mathbf{x}) = 1 - \frac{\beta_{01}}{\alpha_{01}} \cdot \frac{1 - \pi}{\pi} \cdot \frac{\kappa(\mathbf{x})}{1 - \kappa(\mathbf{x})}, \quad \eta_0(\mathbf{x}) = \frac{\frac{\beta_{10}}{\alpha_{10}}\xi_0(\mathbf{x})}{\frac{\beta_{10}}{\alpha_{10}}\xi_0(\mathbf{x}) + \frac{\beta_{00}}{\alpha_{00}}\{1 - \xi_0(\mathbf{x})\}}, \text{ and} \tag{8}$$

$$\eta(\mathbf{x}) = \eta_1(\mathbf{x})\tau_0(\mathbf{x}) + \eta_0(\mathbf{x})\{1 - \tau_0(\mathbf{x})\}.$$

The proof of this result is provided in Appendix A.1. Proposition 3.1 illustrated that, in general, the naive method presented in Section 3.1 fails. There are no explicit relations between $\eta_1(\mathbf{x})$ and $\xi_1(\mathbf{x})$ or between $\eta(\mathbf{x})$ and $\xi(\mathbf{x})$. For the relation between $\eta_0(\mathbf{x})$ and $\xi_0(\mathbf{x})$, they coincide only in the special case that $\beta_{10}/\alpha_{10} = \beta_{00}/\alpha_{00}$, which corresponds to a proportionality condition between the class-conditional densities across domains. Outside of this narrow scenario, the naive approach systematically misestimates the target posterior, leading to biased predictions.

This result also implies that model adaptation fundamentally relies on estimating the proportions of key subgroups in the target population. In particular, for individuals with $A = 1$, one only needs to estimate $\beta_{01}$, while for those with $A = 0$, it suffices to estimate the ratio $\beta_{10}/\beta_{00}$. Denote $\boldsymbol{\beta} = (\beta_{10}, \beta_{00})^{\mathrm{T}}$. It can be seen that, accurate estimation of the parameter $\boldsymbol{\beta}$ in the target domain enables valid model adaptation across domains. Before developing methods for estimating $\boldsymbol{\beta}$ in Section 3.4, we first present some model identification considerations.

### 3.3 Model identification considerations

The identifiability structure of our problem closely resembles that of the *open set label shift* (OSLS) framework (Garg et al., 2022). Note that our target distribution consists of a mixture over four joint distributions: $\mathrm{pr}(Y = 1, A = 1)$, $\mathrm{pr}(Y = 1, A = 0)$, $\mathrm{pr}(Y = 0, A = 0)$, and $\mathrm{pr}(Y = 0, A = 1)$. By treating the joint label $(Y, A)$ as the response, this setting can be viewed as a special case of the OSLS framework. However, our setup is considerably simpler due to the availability of the auxiliary variable $A$ in the target domain. As a result, we can restrict attention to the subset $A = 1$, thereby discarding the $A = 0$ portion of the distribution. This reduction simplifies the problem to recovering $\mathrm{pr}(Y = 1, A = 1)$ from a mixture of $\mathrm{pr}(Y = 1, A = 1)$ and $\mathrm{pr}(Y = 0, A = 1)$, given direct access to $\mathrm{pr}(Y = 0, A = 1)$. This is a canonical *positive-unlabeled (PU) learning* problem. Identifiability in this setting is governed by the standard *anchor set condition* (see Definition 8 of Ramaswamy et al. (2016)): there exists a measurable subset $\mathbf{x}_{\mathrm{anchor}} \in \mathcal{X}$ such that

$$p(\mathbf{X} \in \mathbf{x}_{\mathrm{anchor}}|Y = 0, A = 1) > 0 \quad \text{and} \quad \frac{p(\mathbf{X} \in \mathbf{x}_{\mathrm{anchor}}|Y = 1, A = 1)}{p(\mathbf{X} \in \mathbf{x}_{\mathrm{anchor}}|Y = 0, A = 1)} = 0.$$

This condition ensures that the positive class $(Y = 1, A = 1)$ has no support on a subset of the feature space that is occupied by the negative class $(Y = 0, A = 1)$, which is necessary for identifiability. Under the assumption (3), the primary difficulty arises from the fact that the component $p_{11}(\mathbf{x})$, corresponding to the subgroup $(Y = 1, A = 1)$, is not directly observable in either the source or target domain.

To elucidate this observation, we denote $\mathbf{p}_0(\mathbf{x}) = \{p_{10}(\mathbf{x}), p_{00}(\mathbf{x})\}^{\mathrm{T}}$, and then the observed data log-likelihood of one generic observation in our UDA setting is proportional to:

$$I_{110}\log p_{10}(\mathbf{x}) + I_{101}\log p_{01}(\mathbf{x}) + I_{100}\log p_{00}(\mathbf{x})$$
$$+ I_{0\cdot1}\log\left\{\beta_{11}p_{11}(\mathbf{x}) + (1 - \beta_{11} - \boldsymbol{\beta}^{\top}\mathbf{1})p_{01}(\mathbf{x})\right\} + I_{0\cdot0}\log\left\{\boldsymbol{\beta}^{\top}\mathbf{p}_0(\mathbf{x})\right\}.$$

In this formulation, the parameter with finite dimension is $\boldsymbol{\beta}$. The model involves four nonparametric nuisance components: $p_{11}(\mathbf{x})$, $p_{10}(\mathbf{x})$, $p_{01}(\mathbf{x})$, and $p_{00}(\mathbf{x})$.

**Lemma 3.2.** *Assume $\beta_{11} = 0$ and $p_{10}(\mathbf{x}) \neq p_{00}(\mathbf{x})$, then all components except $p_{11}(\mathbf{x})$ are identifiable. Assume $0 < \beta_{11} < 1$ and is known, and $p_{10}(\mathbf{x}) \neq p_{00}(\mathbf{x})$, then all components in the model are identifiable.*

The proof of Lemma 3.2 is provided in Appendix A.1. The identification conditions in Lemma 3.2 are intuitive and reasonable. If $\beta_{11} = 0$, it degenerates to the situation that the source and target domains have the same support on both label $Y$ and background $A$, then the component $p_{11}(\mathbf{x})$ is no longer relevant. Also, if $p_{10}(\mathbf{x}) = p_{00}(\mathbf{x})$, the subpopulations of $(Y = 1, A = 0)$ and $(Y = 0, A = 0)$ become indistinguishable, and hence the individual probabilities $\beta_{10}$ and $\beta_{00}$ are not separately identifiable. Overall, these conditions are natural to ensure the problem is well-posed.

### 3.4 ESTIMATING PARAMETERS OF INTEREST

To estimate the parameter $\boldsymbol{\beta}$, we consider the distribution of attributes $\mathbf{x}$ in the subpopulation defined by $(R = 0, A = 0)$. By the law of total probability, we have

$$p(\mathbf{x} \mid R = 0, A = 0)\mathrm{pr}(R = 0, A = 0) = p_{10}(\mathbf{x})\,\beta_{10}(1 - \pi) + p_{00}(\mathbf{x})\,\beta_{00}(1 - \pi), \qquad (9)$$

subject to the constraint

$$\mathrm{pr}(R = 0, A = 0) = \beta_{10}(1 - \pi) + \beta_{00}(1 - \pi). \qquad (10)$$

Note that the distribution $p(\mathbf{x} \mid R = 0, A = 0)$ is identifiable from the target population. The distributions $p_{10}(\mathbf{x})$ and $p_{00}(\mathbf{x})$ can be consistently estimated from the source population subgroups $(R = 1, Y = 1, A = 0)$ and $(R = 1, Y = 0, A = 0)$, respectively. Thus, the parameters $\boldsymbol{\beta} = (\beta_{00}, \beta_{10})^{\mathrm{T}}$ can be estimated by minimizing a suitable discrepancy measure between the two sides of equation (9), such as an $L_2$ norm or a divergence-based criterion (e.g., Kullback–Leibler divergence), subject to the constraint in equation (10). Therefore, we reformulate the estimation of $\boldsymbol{\beta}$ as a constrained distribution matching problem:

$$\widehat{\boldsymbol{\beta}} = \mathrm{argmin}_{\boldsymbol{\beta}} D\left\{\widehat{p}(\mathbf{x} \mid R = 0, A = 0)\|\{\widehat{p}_{10}(\mathbf{x})\beta_{10} + \widehat{p}_{00}(\mathbf{x})\beta_{00}\}/\widehat{\mathrm{pr}}(A = 0|R = 0)\right\}, \qquad (11)$$

subject to $\widehat{\mathrm{pr}}(A = 0|R = 0) = \beta_{10} + \beta_{00}$, where $D$ denotes a discrepancy measure between probability distributions over the covariate space $\mathcal{X}$. Among various choices for $D$, we adopt the Kullback–Leibler (KL) divergence due to its favorable analytical and computational properties. To facilitate optimization, we relax the constraint in (11) and reformulate the objective under KL divergence, as summarized in the following lemma.

**Lemma 3.3.** *Let $D$ be the Kullback–Leibler divergence. Then the solution $\widehat{\beta}_{10}$ to the minimization problem (11) is given by*

$$argmax_{\beta_{10}}\,\widehat{E}\left(\log[\widehat{\xi}_0(\mathbf{X})\widehat{b}_1^{-1}\beta_{10} + \{1 - \widehat{\xi}_0(\mathbf{X})\}(1 - \widehat{b}_1)^{-1}(\widehat{\varrho} - \beta_{10})]\,\Big|\,R = 0, A = 0\right), \qquad (12)$$

*where, for simplicity, $b_1 = pr(Y = 1|R = 1, A = 0)$, $\varrho = pr(A = 0|R = 0)$ and $\widehat{E}$ represents the empirical average.*

The proof of Lemma 3.3 is provided in Appendix A.1. A key advantage of minimizing the KL divergence is that it circumvents the need to explicitly estimate the generative models $p_{10}(\mathbf{x})$ and

$p_{00}(\mathbf{x})$, which are often difficult to model accurately in high dimensions. Instead, it suffices to estimate one background-specific prediction model $\xi_0(\mathbf{x})$ using standard classification techniques on the source domain restricted to $A = 0$.

The above method adopts the idea of distribution matching. Alternatively, one may consider matching only certain moments rather than the full distribution. Due to space constraints, we defer the details to Appendix A.1.

---

**Algorithm 1** Implementation details of our proposed method.

---

**Input:** Observed source domain data $\{(\mathbf{X}_i, Y_i, A_i, R_i = 1)\}_{i=1}^{n_1}$ and target domain data $\{(\mathbf{X}_i, A_i, R_i = 0)\}_{i=1}^{n_0}$.

**Output:** Estimated benchmark prediction models $\widehat{\xi}(\mathbf{x})$, $\widehat{\xi}_1(\mathbf{x})$ and $\widehat{\xi}_0(\mathbf{x})$, proposed prediction models for the target $\widehat{\eta}(\mathbf{x})$, $\widehat{\eta}_1(\mathbf{x})$ and $\widehat{\eta}_0(\mathbf{x})$; and subpopulation proportions $\widehat{\alpha}_{ya}$, $\widehat{\beta}_{ya}$.

1: Estimate $\xi(\mathbf{x})$ (defined in (4)) using data $\{(\mathbf{X}_i, Y_i, R_i = 1) : i = 1, \cdots, n_1\}$, as $\widehat{\xi}(\mathbf{x})$;
2: Estimate $\xi_0(\mathbf{x})$ (defined in (4)) using data $\{(\mathbf{X}_i, Y_i, A_i = 0, R_i = 1) : i = 1, \cdots, n_1\}$, as $\widehat{\xi}_0(\mathbf{x})$;
3: Estimate $\tau_r(\mathbf{x})$ (defined in (6)) using data $\{(\mathbf{X}_i, A_i, R_i = r) : i = 1, \cdots n_r\}, r = 0, 1$, as $\widehat{\tau}_r(\mathbf{x})$;
4: Estimate $\xi_1(\mathbf{x})$ following equation (5), as $\widehat{\xi}_1(\mathbf{x})$;
5: Estimate $\kappa(\mathbf{x})$ (defined in (7)) using data $\{(\mathbf{X}_i, R_i, A_i = 1) : i = 1, \cdots n\}$, as $\widehat{\kappa}(\mathbf{x})$;
6: Estimate $\boldsymbol{\beta}$ and $\alpha_{y,a}$ following equations (12) and (1), as $\widehat{\boldsymbol{\beta}}$ and $\widehat{\alpha}_{ya}$ for $(y, a) \in \{0, 1\}$;
7: Estimate $\eta_1(\mathbf{x})$, $\eta_0(\mathbf{x})$ and $\eta(\mathbf{x})$ following equation (8), as $\widehat{\eta}_1(\mathbf{x})$, $\widehat{\eta}_0(\mathbf{x})$ and $\widehat{\eta}(\mathbf{x})$.

---

Finally, based on all of the above discussions, we summarize the implementation details of our proposed method in Algorithm 1.

## 3.5 DOWNSTREAM TASKS

With any loss function $\ell(\cdot)$, for the background-specific prediction model with $A = 0$, the conditional risk is

$$E[\ell\{h(\mathbf{X}), Y\} \mid R = 0, A = 0] = E[\ell\{h(\mathbf{X}), Y\} \, w(Y) \mid R = 1, A = 0], \tag{13}$$

where, for simplicity, we write $w(y) = \frac{\text{pr}(y|R=0,A=0)}{\text{pr}(y|R=1,A=0)}$. One can derive that $w(1) = \frac{\beta_{10}(\alpha_{00}+\alpha_{10})}{\alpha_{10}(\beta_{00}+\beta_{10})}$ and $w(0) = \frac{\beta_{00}(\alpha_{00}+\alpha_{10})}{\alpha_{00}(\beta_{00}+\beta_{10})}$. To evaluate the performance of the prediction model, it can be approximated as $\widehat{E}[\ell\{h(\mathbf{X}), Y\} \, \widehat{w}(Y) \mid R = 1, A = 0]$. Furthermore, the model can be fine-tuned specifically for the target subgroup by minimizing the reweighted empirical risk:

$$\widehat{h}_{\widehat{w}} \in \arg\min_{h \in \mathcal{F}} \widehat{E}[\ell\{h(\mathbf{X}), Y\} \, \widehat{w}(Y) \mid R = 1, A = 0], \tag{14}$$

where $\mathcal{F}$ is a suitable function class.

For the interest of space, for the other two prediction models, we only present the relations analogous to (13) without elaborations. For the background-specific prediction model with $A = 1$, one can derive $E[\ell\{h(\mathbf{X}), Y\}|R = 0, A = 1]$ as

$$E[\ell\{h(\mathbf{X}), Y = 1\}|R = 0, A = 1]$$
$$-E\left([\ell\{h(\mathbf{X}), Y = 1\} - \ell\{h(\mathbf{X}), Y = 0\}] \,|Y = 0, A = 1\right) \frac{\beta_{01}}{\beta_{01} + \beta_{11}}.$$

For the overall prediction model, the conditional risk $E[\ell\{h(\mathbf{X}), Y\}|R = 0]$ is

$$E[\ell\{h(\mathbf{X}), Y\} \, w(Y) \mid R = 1, A = 0](\beta_{10} + \beta_{00})$$
$$+E[\ell\{h(\mathbf{X}), Y = 1\}|R = 0, A = 1](\beta_{01} + \beta_{11})$$
$$-E\left([\ell\{h(\mathbf{X}), Y = 1\} - \ell\{h(\mathbf{X}), Y = 0\}] \,|Y = 0, A = 1\right) \beta_{01}.$$

# 4 THEORETICAL RESULTS

For the interest of space, we only present the results for the background-specific prediction model with $A = 0$. The results for the other two prediction models are parallel and can be similarly developed. To facilitate the analysis, we begin by formally defining the population-level (expected) objective function:

$$\mathfrak{L}(\xi_0, b_1, \beta_{10}, \varrho) = E\left(\log[\xi_0(\mathbf{X})b_1^{-1}\beta_{10} + \{1 - \xi_0(\mathbf{X})\}(1 - b_1)^{-1}(\varrho - \beta_{10})]\big|R = 0, A = 0\right),$$

with its empirical version $\widehat{\mathfrak{L}}(\xi_0, b_1, \beta_{10}, \varrho)$.

**Assumption 4.1.** *Define $\boldsymbol{f}(\mathbf{x}) = \{f_0(\mathbf{x}), f_1(\mathbf{x})\}^{\mathrm{T}}$, where $f_0(\mathbf{x}) = \log\{\xi_0(\mathbf{x})\} - \frac{1}{2}[\log\{\xi_0(\mathbf{x})\} + \log\{1 - \xi_0(\mathbf{x})\}]$ and $f_1(\mathbf{x}) = \log\{1 - \xi_0(\mathbf{x})\} - \frac{1}{2}[\log\{\xi_0(\mathbf{x})\} + \log\{1 - \xi_0(\mathbf{x})\}]$, and the corresponding estimate is $\{\widehat{f}_k(\mathbf{x})\}_{k=0}^1$. There exist a constant $c > 0$ and a sequence $r_{n_{1\cdot0}} \to 0$ such that, for almost every $\mathbf{x}$, we have*

$$pr\left(\|\widehat{\boldsymbol{f}}(\mathbf{x}) - \boldsymbol{f}(\mathbf{x})\|_2 > t\right) \leq \exp\left\{-t^2/(c^2 r_{n_{1\cdot0}}^2)\right\}, \quad \forall\, t > 0.$$

**Theorem 4.2.** *Suppose Assumption 4.1 holds. Define $\chi_n = r_{n_{1\cdot0}}\sqrt{\log(n_{0\cdot0})} + n_{1\cdot0}^{-1/2} + n_{0\cdot0}^{-1/2}$. Then, there exists a constant $c_{10} > 0$ such that for any $\delta > 0$, with probability at least $1 - 6\delta$, we have*

$$\|\widehat{\boldsymbol{\beta}} - \boldsymbol{\beta}\|_1 \leq c_{10}\chi_n\sqrt{\log(1/\delta)}.$$

The proof of Theorem 4.2 is provided in Appendix A.2. Theorem 4.2 establishes the consistency of the estimator $\widehat{\boldsymbol{\beta}}$, provided that $r_{n_{1\cdot0}}\sqrt{\log(n_{0\cdot0})} \to 0$ as $n_{1\cdot0}, n_{0\cdot0} \to \infty$.

Next, we establish a generalization bound for the fitted model (14), which is obtained via weighted empirical risk minimization over the source subgroup. Let $\mathcal{F}$ denote the hypothesis class of classifiers. For any $h \in \mathcal{F}$ and a weight function $w(y) : y \to \mathbf{R}$, we define the population-level weighted loss and its empirical counterpart based on the source subgroup data as follows:

$$\begin{aligned}
\mathcal{L}_1(h, w) &= E\left[\ell\{h(\mathbf{X}), Y\}w(Y)\big|A = 0, R = 1\right], \\
\widehat{\mathcal{L}}_1(h, w) &= \widehat{E}\left[\ell\{h(\mathbf{X}), Y\}w(Y)\big|A = 0, R = 1\right].
\end{aligned}$$

We also define the population loss on the target subgroup as: $\mathcal{L}_0(h) = E\left[\ell\{h(\mathbf{X}), Y\}\big|R = 0, A = 0\right]$. Clearly, $\mathcal{L}_1(h, w) = \mathcal{L}_0(h)$.

To establish our generalization bound, we utilize the concept of Rademacher complexity (Bartlett & Mendelson, 2002), denoted as $\mathcal{R}_n(\mathcal{G})$ (see Appendix A.2 for details), and impose the following assumption on the loss function:

**Assumption 4.3.** *The loss function $\ell$ is uniformly bounded; that is, there exists a constant $B > 0$ such that*

$$|\ell\{h(\mathbf{x}), y\}| \leq B \text{ for any } h \in \mathcal{F}, \mathbf{x} \in \mathcal{X} \subset \mathbf{R}^q, \text{ and } y \in \{0, 1\}.$$

We now present the generalization bound for the learned model, with its proof provided in Appendix A.2.

**Proposition 4.4.** *Under Assumptions 4.1 and 4.3, let $\widehat{h}_{\widehat{w}} = argmin_{h \in \mathcal{F}}\widehat{\mathcal{L}}_1(h, \widehat{w})$ be the classifier obtained by minimizing the reweighted empirical risk on the source subgroup. Then, there exist constants $c, d > 0$ such that, with probability at least $1 - 7\delta$, the following generalization bound holds:*

$$\mathcal{L}_0(\widehat{h}_{\widehat{w}}) - \min_{h \in \mathcal{F}}\mathcal{L}_0(h) \leq 2\mathcal{R}_{n_{1\cdot0}}(\mathcal{G}) + dB\|\widehat{\boldsymbol{\beta}} - \boldsymbol{\beta}\|_1 + c\left\{\sqrt{\frac{\log(1/\delta)}{n_{1\cdot0}}} + \sqrt{\frac{\log(1/\delta)}{n_0}}\right\},$$

*where $\mathcal{G} = \{w(y)\ell\{h(\mathbf{x}), y\} : h \in \mathcal{F}\}$, and $\mathcal{R}_{n_{1\cdot0}}(\mathcal{G})$ denotes its Rademacher complexity as defined in Appendix A.2.*

## 5 EXPERIMENTS

We conduct extensive experiments on both synthetic and real-world datasets. Due to space limitations, we present the results on the Waterbirds dataset here and provide additional results in Appendix A.3.

The **Waterbirds** dataset (Sagawa et al., 2019) consists of 11,788 images. The label $Y = 1$ denotes a waterbird and $Y = 0$ landbird. The background $A = 1$ corresponds to a water background and $A = 0$ a land background. It yields four label–background subpopulations, as summarized in Table 3.

Table 3: Empirical joint distribution of $(Y, A)$ in the Waterbirds dataset, with varied values of $a$, $b$ and $c$, $0 < a, b, c < 1$.

| $Y$ | $A$ | Description | Count | Total Proportion | Proportion in Source | Proportion in Target |
|---|---|---|---|---|---|---|
| 1 | 1 | Waterbird on water | 1832 | 0.155 | 0 | 0.155 |
| 0 | 1 | Landbird on water | 2905 | 0.246 | $0.246a$ | $0.246(1 - a)$ |
| 1 | 0 | Waterbird on land | 831 | 0.071 | $0.071b$ | $0.071(1 - b)$ |
| 0 | 0 | Landbird on land | 6220 | 0.528 | $0.528c$ | $0.528(1 - c)$ |

To construct a structured domain adaptation problem, we partition the full dataset into a *source domain* ($R = 1$) and a *target domain* ($R = 0$). Specifically, we allocate samples from three subgroups—$(Y = 0, A = 1)$, $(Y = 1, A = 0)$, and $(Y = 0, A = 0)$—into the source domain, with allocation rates denoted by parameters $a$, $b$, and $c$, respectively. The remaining subgroup, $(Y = 1, A = 1)$, is deliberately *excluded* from the source domain and appears only in the target domain. This setting reflects real-world scenarios in which a specific combination of label and background is structurally missing from labeled datasets due to systematic data collection biases or constraints. In the target domain, all four subgroups are retained, but the label variable $Y$ is treated as unobserved.

To implement the proposed method, we apply the distribution matching approach to estimate the subclass proportions in the target domain. For feature extraction, we embed each image into a 512-dimensional feature vector using a ResNet-18 model (He et al., 2016) pre-trained on ImageNet (Deng et al., 2009), without additional fine-tuning. These embeddings serve as covariates $\mathbf{X} \in \mathbf{R}^{512}$ in our downstream analysis. Based on these feature vectors, we fit logistic regression models with $L_2$-regularization to estimate five key conditional probabilities required by both our proposed method and benchmark procedures: $\xi_0(\mathbf{x})$, $\xi(\mathbf{x})$, $\tau_0(\mathbf{x})$, $\tau_1(\mathbf{x})$ and $\kappa(\mathbf{x})$.

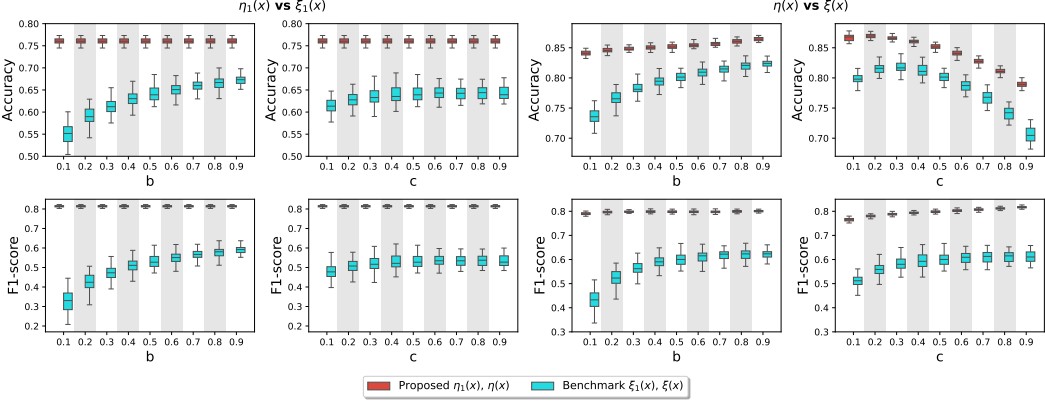

Figure 1: Performance comparison of our proposed estimators $\eta_1(\mathbf{x})$, $\eta(\mathbf{x})$, and the benchmark method $\xi_1(\mathbf{x})$, $\xi(\mathbf{x})$ under the setting $a = 0.5$ with either $c = 0.5$ and varying $b$ or $b = 0.5$ and varying $c$.

For empirical evaluation, we fix the subclass sampling rate at $a = 0.5$ in the source domain and systematically vary the remaining subclass inclusion rates by setting either $b = 0.5$ with $c \in \{0.1, 0.2, \ldots, 0.9\}$, or $c = 0.5$ with $b \in \{0.1, 0.2, \ldots, 0.9\}$. For each configuration, the data

generation process is repeated 50 times to account for sampling variability. We assess performance using two widely adopted classification metrics: accuracy and $F_1$ score. Figure 1 presents boxplots summarizing the distribution of these metrics across repetitions for our proposed estimators $\widehat{\eta}_1(\mathbf{x})$ and $\widehat{\eta}(\mathbf{x})$, alongside benchmark estimators $\widehat{\xi}_1(\mathbf{x})$ and $\widehat{\xi}(\mathbf{x})$. The results demonstrate that the proposed methods consistently outperform the benchmarks in both accuracy and $F_1$ score, suggesting enhanced robustness to structured subpopulation missingness in the unsupervised domain adaptation setting. Due to the space limitations, additional results are provided in Appendix A.4.

## 6 RELATED WORK

**Out-of-distribution (OOD) generalization** OOD generalization refers to the ability of a prediction model to perform well on test data drawn from a distribution that differs from the training data. In our context, the subpopulation $(Y = 1, A = 1)$ in the target can be regarded as the OOD data while the other three subpopulations are in-distribution data. For a comprehensive overview of OOD generalization, we refer the readers to the excellent survey (Liu et al., 2021), which reviewed real-world datasets, evaluation protocols, and key challenges in this area. In the OOD generalization literature, different methods were proposed with different emphases: (Arjovsky et al., 2019) emphasized the need to minimize invariant risk across different environments to ensure consistent model performance, whereas (Sagawa et al., 2019) underscored the importance of distributionally robust optimization (DRO) and various regularization techniques in reducing performance disparities across subgroups. In addition, (Bahng et al., 2020) introduced adversarial training as a method for learning de-biased representations, which is critical for promoting fairness in machine learning models, and (Sohoni et al., 2020) examined the issue of robustness in classification tasks involving coarse classes that contain finer subclasses, enhancing model performance across all subclasses.

**OOD detection** OOD detection is the task of identifying inputs at test time that do not come from the same distribution as the training data. Its goal is to prevent a model from making confident but incorrect predictions on unfamiliar or anomalous inputs by flagging them as OOD. There are a variety of techniques developed for OOD detection in the literature. For example, (Hendrycks & Gimpel, 2017) introduced a simple yet effective method for detecting both misclassified and OOD inputs in neural networks. Liang et al. (2018) (ODIN) proposed an improved method for detecting OOD inputs by applying temperature scaling to the softmax outputs and adding small input perturbations during inference. ODIN significantly outperformed previous baseline methods, including the maximum softmax probability approach, and set a new standard for OOD detection in classification tasks. Other techniques include but not limited to, outlier exposure (Hendrycks et al., 2018; Papadopoulos et al., 2021), ConfGAN (Sricharan & Srivastava, 2018) and OodGAN (Marek et al., 2021). In addition, Fort et al. (2021) provided an extensive empirical study of OOD detection methods across a wide range of datasets, architectures, and training regimes.

**Spurious correlation** Spurious correlation is a major obstacle to OOD generalization, where models often rely on non-causal features that can degrade performance, particularly when these correlations do not generalize across domains. For example, a model trained to classify cows might rely on green pastures (background) instead of the cow itself. On a desert background, it fails. This is also the case in the Waterbirds dataset where the spurious correlation exists between label $Y$ and background $A$. Different learning strategies were proposed to discover and mitigate the impact of spurious correlation on model performance, as well as to improve model robustness. For example, (Wu et al., 2023) introduced an attention-based approach to automatically identify spurious concepts and apply adversarial training to reduce reliance on them. Another approach proposed by (Kumar et al., 2023) used causal regularization to detect and discourage spurious dependencies, allowing for scalable robustness across shifts. In addition, (Sagawa et al., 2020) investigated why overparameterization exacerbates spurious correlations, and (Kirichenko et al., 2022) found that retraining only the final layer on a small, balanced dataset can restore robustness against spurious correlations. Also, (Wang & Wang, 2024) developed a theoretical model to analyze the influence of spurious correlation strength, sample size, and feature noise on learning. Spurious correlations were also investigated in feature learning (Izmailov et al., 2022; Qiu et al., 2024), reinforcement learning (Ding et al., 2023), OOD detection (Ming et al., 2022), and text classification (Wang & Culotta, 2020). One can also resort to a comprehensive survey paper (Ye et al., 2024) on this topic.

**Ethics statement**

We confirm that all authors of this manuscript have thoroughly read and strictly adhered to the ICLR Code of Ethics (available at: `https://iclr.cc/public/CodeOfEthics`). Our commitment to ethical research practices has been maintained throughout this study. The datasets utilized in this research are publicly available benchmark datasets. To the best of our knowledge, this work poses no foreseeable societal risks and is free from any potential conflicts of interest.

**Reproducibility statement**

We confirm the reproducibility of our work and will release a complete source code implementation along with detailed setup instructions upon acceptance of this paper. The implementation details of our proposed method are systematically outlined in Algorithm 1. All datasets used in this study are publicly accessible. For our theoretical contributions, we provide rigorous proofs in the Appendix, along with a detailed discussion of the underlying assumptions.

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

## A  APPENDIX

### A.1  PROOFS AND MORE DETAILS IN SECTION 3

*Proof of Proposition 3.1.* For $A = 1$ case, note that

$$p(\mathbf{x} \mid R = 0, A = 1)\mathrm{pr}(R = 0, A = 1)$$
$$= p(\mathbf{x} \mid R = 0, Y = 1, A = 1)p_{011} + p(\mathbf{x} \mid R = 0, Y = 0, A = 1)p_{001}.$$

Thus,

$$p_{11}(\mathbf{x}) = \frac{p(\mathbf{x} \mid R = 0, A = 1)\mathrm{pr}(R = 0, A = 1) - p_{01}(\mathbf{x})p_{001}}{p_{011}}.$$

Then,

$$\text{pr}(Y = 1 \mid \mathbf{x}, R = 0, A = 1) = \frac{p_{11}(\mathbf{x})p_{011}}{p(\mathbf{x}, R = 0, A = 1)}$$

$$= \frac{p(\mathbf{x} \mid R = 0, A = 1)\text{pr}(R = 0, A = 1) - p_{01}(\mathbf{x})p_{001}}{p(\mathbf{x} \mid R = 0, A = 1)\text{pr}(R = 0, A = 1)}$$

$$= \frac{p(\mathbf{x} \mid R = 0, A = 1)p_{0\cdot1} - p_{01}(\mathbf{x})\beta_{01}(1 - \pi)}{p(\mathbf{x} \mid R = 0, A = 1)p_{0\cdot1}}$$

Note that

$$\text{pr}(R = 1 \mid \mathbf{x}, A = 1) = \frac{p_{01}(\mathbf{x})\alpha_{01}\pi}{p_{01}(\mathbf{x})\alpha_{01}\pi + p(\mathbf{x} \mid R = 0, A = 1)p_{0\cdot1}}$$

gives

$$\frac{p_{01}(\mathbf{x})}{p(\mathbf{x} \mid R = 0, A = 1)p_{0\cdot1}} = \frac{\text{pr}(R = 1 \mid \mathbf{x}, A = 1)}{\alpha_{01}\pi\{1 - \text{pr}(R = 1 \mid \mathbf{x}, A = 1)\}}.$$

Hence,

$$\text{pr}(Y = 1 \mid \mathbf{x}, R = 0, A = 1) = 1 - \frac{\beta_{01}(1 - \pi)}{\alpha_{01}\pi} \frac{\text{pr}(R = 1 \mid \mathbf{x}, A = 1)}{1 - \text{pr}(R = 1 \mid \mathbf{x}, A = 1)}.$$

Note that

$$\text{pr}(R = 1 \mid \mathbf{x}, A = 1) = \frac{p(\mathbf{x} \mid R = 1, A = 1)\alpha_{01}\pi}{p(\mathbf{x} \mid R = 1, A = 1)\alpha_{01}\pi + p(\mathbf{x} \mid R = 0, A = 1)p_{0\cdot1}}$$

gives

$$\frac{p(\mathbf{x} \mid R = 1, A = 1)}{p(\mathbf{x} \mid R = 0, A = 1)p_{0\cdot1}} = \frac{\text{pr}(R = 1 \mid \mathbf{x}, A = 1)}{\alpha_{01}\pi\{1 - \text{pr}(R = 1 \mid \mathbf{x}, A = 1)\}}.$$

Hence,

$$\text{pr}(Y = 1 \mid \mathbf{x}, R = 0, A = 1) = 1 - \frac{\beta_{01}(1 - \pi)}{\alpha_{01}\pi} \frac{\text{pr}(R = 1 \mid \mathbf{x}, A = 1)}{1 - \text{pr}(R = 1 \mid \mathbf{x}, A = 1)} \left\{ \frac{p(\mathbf{x}|Y = 0, A = 1)}{p(\mathbf{x}|R = 1, A = 1)} \right\}.$$

For $A = 0$ case, note that

$$\text{pr}(Y = 1 \mid \mathbf{x}, R = 0, A = 0) = \frac{\text{pr}(Y = 1, \mathbf{x}, R = 0, A = 0)}{\text{pr}(Y = 1, \mathbf{x}, R = 0, A = 0) + \text{pr}(Y = 0, \mathbf{x}, R = 0, A = 0)}$$

$$= \frac{\text{pr}(\mathbf{x}, Y = 1, R = 1, A = 0)\frac{\text{pr}(Y=1,R=0,A=0)}{\text{pr}(Y=1,R=1,A=0)}}{\text{pr}(\mathbf{x}, Y = 1, R = 1, A = 0)\frac{\text{pr}(Y=1,R=0,A=0)}{\text{pr}(Y=1,R=1,A=0)} + \text{pr}(\mathbf{x}, Y = 0, R = 1, A = 0)\frac{\text{pr}(Y=0,R=0,A=0)}{\text{pr}(Y=0,R=1,A=0)}}$$

$$= \frac{\frac{\beta_{10}}{\alpha_{10}}\xi_0(\mathbf{x})}{\frac{\beta_{10}}{\alpha_{10}}\xi_0(\mathbf{x}) + \frac{\beta_{00}}{\alpha_{00}}\{1 - \xi_0(\mathbf{x})\}}.$$

By Bayes' rule, we obtain the following equation

$$\eta(\mathbf{x}) = \eta_1(\mathbf{x})\tau_0(\mathbf{x}) + \eta_0(\mathbf{x})\{1 - \tau_0(\mathbf{x})\}.$$

$\square$

*Proof of Lemma 3.2.* It is easy to see that, $\pi$, $\alpha_{10}$, $\alpha_{01}$, $p_{10}(\mathbf{x})$, $p_{01}(\mathbf{x})$ and $p_{00}(\mathbf{x})$ are all identifiable. Now suppose that there are two different sets $p_{11}(\mathbf{x})$, $\beta_{10}$, $\beta_{00}$ and $\widetilde{p}_{11}(\mathbf{x})$, $\widetilde{\beta}_{10}$, $\widetilde{\beta}_{00}$ such that

$$\beta_{11}p_{11}(\mathbf{x}) + (1 - \beta_{11} - \beta_{10} - \beta_{00})p_{01}(\mathbf{x}) = \beta_{11}\widetilde{p}_{11}(\mathbf{x}) + (1 - \beta_{11} - \widetilde{\beta}_{10} - \widetilde{\beta}_{00})p_{01}(\mathbf{x}),$$
$$\beta_{10}p_{10}(\mathbf{x}) + \beta_{00}p_{00}(\mathbf{x}) = \widetilde{\beta}_{10}p_{10}(\mathbf{x}) + \widetilde{\beta}_{00}p_{00}(\mathbf{x}). \tag{15}$$

Now taking the integral with respect to $\mathbf{x}$ on both sides of the second equation above, it is clear that

$$\beta_{10} + \beta_{00} = \widetilde{\beta}_{10} + \widetilde{\beta}_{00}.$$

Plugging in back to the first equation above, we obtain

$$\beta_{11}\{p_{11}(\mathbf{x}) - \widetilde{p}_{11}(\mathbf{x})\} = 0.$$

Since $\beta_{11} > 0$, we obtain $p_{11}(\mathbf{x}) = \widetilde{p}_{11}(\mathbf{x})$. Finally, (15) leads to $(\beta_{10} - \widetilde{\beta}_{10})p_{10}(\mathbf{x}) = (\widetilde{\beta}_{00} - \beta_{00})p_{00}(\mathbf{x})$, which can only hold if $\beta_{10} = \widetilde{\beta}_{10}$ and $\widetilde{\beta}_{00} = \beta_{00}$ since $p_{10}(\mathbf{x}) \neq p_{00}(\mathbf{x})$. This completes the proof. $\square$

*Proof of Lemma 3.3.*

$$D\left\{p(\mathbf{x}|R=0, A=0) \left\| \sum_{k=0}^{1} p(\mathbf{x}|Y=k, A=0)\beta_{k0}\frac{1-\pi}{\text{pr}(R=0, A=0)} \right.\right\}$$

$$= \int p(\mathbf{x}|R=0, A=0)\log\frac{p(\mathbf{x}|R=0, A=0)}{\sum_{k=0}^{1} p(\mathbf{x}|Y=k, A=0)\beta_{k0}\frac{1-\pi}{\text{pr}(R=0,A=0)}}\mathbf{dx}$$

$$= \int p(\mathbf{x}|R=0, A=0)\log\frac{p(\mathbf{x}|R=0, A=0)}{\sum_{k=0}^{1} p(\mathbf{x}|Y=k, A=0)\beta_{k0}\frac{1-\pi}{\text{pr}(R=0,A=0)}}\mathbf{dx}$$

$$= \int p(\mathbf{x}|R=0, A=0)\log\frac{p(\mathbf{x}|R=0, A=0)}{p(\mathbf{x}|R=1, A=0)}\mathbf{dx}$$

$$\quad - \int p(\mathbf{x}|R=0, A=0)\log\frac{\sum_{k=0}^{1} p(\mathbf{x}|Y=k, A=0)\beta_{k0}\frac{1-\pi}{\text{pr}(R=0,A=0)}}{p(\mathbf{x}|R=1, A=0)}\mathbf{dx}$$

$$= \int p(\mathbf{x}|R=0, A=0)\log\frac{p(\mathbf{x}|R=0, A=0)}{p(\mathbf{x}|R=1, A=0)}\mathbf{dx}$$

$$\quad - \int p(\mathbf{x}|R=0, A=0)\log\sum_{k=0}^{1}\frac{\text{pr}(Y=k|\mathbf{x}, R=1, A=0)\beta_{k0}(1-\pi)\text{pr}(R=1, A=0)}{\text{pr}(R=1, Y=k, A=0)\text{pr}(R=0, A=0)}\mathbf{dx}.$$

Minimizing the above equation is equivalent to maximizing

$$\text{argmax}_{\boldsymbol{\beta}} E\left\{\log\sum_{k=0}^{1}\text{pr}(Y=k|\mathbf{x}, R=1, A=0)\frac{\beta_{k0}}{\text{pr}(Y=k|R=1, A=0)} \right| R=0, A=0\right\},$$

subject to $\text{pr}(R=0, A=0) = \beta_{10}(1-\pi) + \beta_{00}(1-\pi)$.

We enforce this restriction as a constraint in the distribution matching problem: where $D$ is a discrepancy between probability distributions on $\mathcal{X}$.

Define

$$\mathfrak{L}(\xi_0, b_1, \beta_{10}, \varrho) = E\left(\log[\xi_0(\mathbf{X})b_1^{-1}\beta_{10} + \{1 - \xi_0(\mathbf{X})\}(1-b_1)^{-1}(\varrho - \beta_{10})]\big|R=0, A=0\right).$$

Its empirical version is

$$\widehat{\mathfrak{L}}(\xi_0, b_1, \beta_{10}, \varrho) = \widehat{E}\left(\log[\xi_0(\mathbf{X})b_1^{-1}\beta_{10} + \{1 - \xi_0(\mathbf{X})\}(1-b_1)^{-1}(\varrho - \beta_{10})]\big|R=0, A=0\right).$$

$\square$

AN ALTERNATIVE APPROACH FOR ESTIMATING $\boldsymbol{\beta}$

In the main text, we explore the use of distribution matching for estimating $\boldsymbol{\beta}$. Alternatively, it is sufficient to only consider some moments instead of the whole distribution. For any measurable function $\mathbf{m}(\mathbf{x})$, the law of total expectation yields the identity:

$$E\{\mathbf{m}(\mathbf{x}) \mid R=0, A=0\}\text{pr}(R=0, A=0)$$

$$= E\{\mathbf{m}(\mathbf{x}) \mid 1, 0\}\beta_{10}(1-\pi) + E\{\mathbf{m}(\mathbf{x}) \mid 0, 0\}\beta_{00}(1-\pi). \tag{16}$$

Rewriting equation (16), we obtain the following linear system:

$$(1-\pi)p_{0\cdot0}^{-1}[E\{\mathbf{m}(\mathbf{x}) \mid 1, 0\}, E\{\mathbf{m}(\mathbf{x}) \mid 0, 0\}]\boldsymbol{\beta} = E\{\mathbf{m}(\mathbf{x}) \mid R=0, A=0\},$$

which leads to the expression

$$\boldsymbol{\beta} = (1 - \pi)^{-1} p_{0\cdot 0} \left[ E\{\mathbf{m}(\mathbf{x}) \mid 1, 0\}, E\{\mathbf{m}(\mathbf{x}) \mid 0, 0\} \right]^{-1} E\{\mathbf{m}(\mathbf{x}) \mid R = 0, A = 0\},$$

provided that the $2 \times 2$ matrix $\left[ E\{\mathbf{m}(\mathbf{x}) \mid 1, 0\}, E\{\mathbf{m}(\mathbf{x}) \mid 0, 0\} \right]$ is invertible. To use the idea of moment matching, one has the flexibility of choosing different moments $\mathbf{m}(\mathbf{x})$. Certainly, a further research question of interest is to identify the optimal choice of this moment function, say, $\mathbf{m}_{\mathrm{opt}}(\mathbf{x})$, by borrowing the semiparametric techniques (Bickel et al., 1993; Tsiatis, 2006).

### A.2 PROOFS AND MORE DETAILS IN SECTION 4

Firstly, note that the tail bound described in Assumption 4.1 is intended to hold uniformly for every $n_{1\cdot 0}$ when estimating $\widehat{f}_k$ for $k = 0, 1$. In other words, for each subsample size $n_{1\cdot 0}$, we have a $r_{n_{1\dot 0}}$ such that the corresponding estimators $\widehat{f}_k$ for $k = 0, 1$ are required to satisfy the stated concentration inequality. This inequality is analogous to Hoeffding's inequality and provides a non-asymptotic concentration bound on the estimation error. Similar assumptions have also been adopted in recent work (e.g., Maity et al. (2022), Tsybakov & Audibert (2007)).

We next define the Rademacher complexity (Bartlett & Mendelson, 2002) that has been frequently used in machine learning literature to establish a generalization bound. Instead of considering the Rademacher complexity on $\mathcal{F}$ we define the class of weighted losses $\mathcal{G}(\ell, \mathcal{F}) = [w(x, y)\ell\{g(x), y\} : g \in \mathcal{F}]$ and $n \in N$ we define its Rademacher complexity measure as

$$\mathcal{R}_n(\mathcal{G}) := E_{u_i, v_i} \left( E_{\xi_i} \left[ \sup_{h \in \mathcal{F}} \frac{1}{n} \sum_{i=1}^{n} \xi_i w(u_i, v_i)\ell\{g(u_i), v_i\} \right] \right),$$

where $\{\xi_i\}_{i=1}^{n}$ are i.i.d. Rademacher random variables, taking values $\pm 1$ with equal probability $1/2$.

*Proof of Theorem 4.2.* For a probabilistic classifier: $\{\xi_0(\mathbf{x}), 1 - \xi_0(\mathbf{x})\} : \mathcal{X} \to \Delta^2$, and the parameter $\boldsymbol{\beta}^{\mathrm{T}} = (\beta_{10}, \beta_{00})$ and $b_1 = \mathrm{pr}(Y = 1 | R = 1, A = 0)$, we define the centered logit function $\boldsymbol{f} : \mathcal{X} \to \mathbf{R}^2$ as $f_0(\mathbf{x}) = \log \xi_0(\mathbf{x}) - \frac{1}{2}[\log \xi_0(\mathbf{x}) + \log\{1 - \xi_0(\mathbf{x})\}]$ and $f_1(\mathbf{x}) = \log\{1 - \xi_0(\mathbf{x})\} - \frac{1}{2}[\log \xi_0(\mathbf{x}) + \log\{1 - \xi_0(\mathbf{x})\}]$. We define the functions $\mu(f_0, b_1) = \xi_0(\mathbf{x})b_1^{-1} - \{1 - \xi_0(\mathbf{x})\}(1 - b_1)^{-1}$ and $\omega(f_0, b_1, \beta_{10}, \varrho) = \xi_0(\mathbf{x})b_1^{-1}\beta_{10} + \{1 - \xi_0(\mathbf{x})\}(1 - b_1)^{-1}(\varrho - \beta_{10})$, and notice that the objective is

$$\widehat{L}(f_0, b_1, \beta_{10}, \varrho) = \widehat{E}\left\{\log \omega(f_0, b_1, \beta_{10}, \varrho) | R = 0, A = 0\right\},$$

whereas the true objective is

$$L(f_0, b_1, \beta_{10}, \varrho) = E\left\{\log \omega(f_0, b_1, \beta_{10}, \varrho) | R = 0, A = 0\right\},$$

We see that the first-order optimality conditions in estimating $\widehat{\beta}_{10}$ are

$$\begin{aligned}
0 &= \partial_{\beta_{10}} \widehat{L}(\widehat{f}_0, \widehat{b}_1, \widehat{\beta}_{10}, \widehat{\varrho}) \qquad\qquad (17)\\
&= \partial_{\beta_{10}} \left[ \widehat{E}\left\{\log \omega(\widehat{f}_0, \widehat{b}_1, \widehat{\beta}_{10}, \widehat{\varrho}) \Big| R = 0, A = 0\right\} \right]\\
&= \widehat{E}\left[ \frac{\partial_{\beta_{10}}\{\omega(\widehat{f}_0, \widehat{b}_1, \widehat{\beta}_{10}, \widehat{\varrho})\}}{\omega(\widehat{f}_0, \widehat{b}_1, \widehat{\beta}_{10}, \widehat{\varrho})} \Big| R = 0, A = 0 \right].
\end{aligned}$$

Similarly, the first order optimality condition at truth (for $\beta_{10}$) are

$$\begin{aligned}
0 &= \partial_{\beta_{10}} L(f_0, b_1, \beta_{10}, \varrho)\\
&= \partial_{\beta_{10}} \left[ E\left\{\log \omega(f_0, b_1, \beta_{10}, \varrho) | R = 0, A = 0\right\} \right]\\
&= E\left[ \frac{\partial_{\beta_{10}}\{\omega(f_0, b_1, \beta_{10}, \varrho)\}}{\omega(f_0, b_1, \beta_{10}, \varrho)} \Big| R = 0, A = 0 \right].
\end{aligned}$$

We decompose (17) using the Taylor expansion and obtain:

$$0 = \partial_{\beta_{10}} \widehat{L}(f_0, \widehat{b}_1, \widehat{\beta}_{10}, \widehat{\varrho}) + \langle \widehat{f}_0 - f_0, \partial_{f_0} \partial_{\beta_{10}} \widehat{L}(\widetilde{f}_0, \widehat{b}_1, \widehat{\beta}_{10}, \widehat{\varrho}) \rangle$$

where $\widetilde{f}_0$ is a function in the bracket $[f_0, \widehat{f}_0]$, i.e. for every $\mathbf{x}$, $\widetilde{f}_0(\mathbf{x})$ is a number between $\widehat{f}_0(\mathbf{x})$ and $f_0(\mathbf{x})$.

**Bound on** $\langle \widehat{f}_0 - f_0, \partial_{f_0}\partial_{\beta_{10}}\widehat{L}(\widetilde{f}_0, \widehat{b}_1, \widehat{\beta}_{10}, \widehat{\varrho})\rangle$**:**

To bound the term, we define $\zeta_0 = \widehat{f}_0 - f_0$ and notice that

$$\langle \zeta_0, \partial_{f_0}\partial_{\beta_{10}}\widehat{L}(\widetilde{f}_0, \widehat{b}_1, \widehat{\beta}_{10}, \widehat{\varrho})\rangle$$

$$= \left\langle \zeta_0, \partial_{f_0}\left[\widehat{E}\left\{\frac{\mu(\widetilde{f}_0, \widehat{b}_1)}{\omega(\widetilde{f}_0, \widehat{b}_1, \widehat{\beta}_{10}, \widehat{\varrho})}\middle| R = 0, A = 0\right\}\right]\right\rangle$$

$$= \widehat{E}\left(\zeta_0 \frac{2\widetilde{\xi}_0(1 - \widetilde{\xi}_0)}{\omega(\widetilde{f}_0, \widehat{b}_1, \widehat{\beta}_{10}, \widehat{\varrho})}\left[\widehat{b}_1^{-1} + (1 - \widehat{b}_1)^{-1}\right.\right.$$

$$\left.\left. - \frac{\mu(\widetilde{f}_0, \widehat{b}_1)}{\omega(\widetilde{f}_0, \widehat{b}_1, \widehat{\beta}_{10}, \widehat{\varrho})}\left\{\widehat{b}_1^{-1}\widehat{\beta}_{10} - (1 - \widehat{b}_1)^{-1}(\widehat{\varrho} - \widehat{\beta}_{10})\right\}\right]\middle| R = 0, A = 0\right).$$

The derivative in third equality in the above display is calculated in Lemma A.1. Assume $\varrho - \epsilon > \beta_{10} > \epsilon > 0$ and $1 - \epsilon_1 > b_1 > \epsilon_1 > 0$, i.e., there exist a $c_1 > 0$ such that $\left|\frac{2\widetilde{\xi}_0(1-\widetilde{\xi}_0)}{\omega(\widetilde{f}_0, \widehat{b}_1, \widehat{\beta}_{10}, \widehat{\varrho})}\left[\widehat{b}_1^{-1} + (1-\widehat{b}_1)^{-1} - \frac{\mu(\widetilde{f}_0, \widehat{b}_1)}{\omega(\widetilde{f}_0, \widehat{b}_1, \widehat{\beta}_{10}, \widehat{\varrho})}\left\{\widehat{b}_1^{-1}\widehat{\beta}_{10} - (1-\widehat{b}_1)^{-1}(\widehat{\varrho} - \widehat{\beta}_{10})\right\}\right]\right| < c_1$. This implies the followings: we have

$$\left|\widehat{E}\left(\zeta_0 \frac{2\widetilde{\xi}_0(1 - \widetilde{\xi}_0)}{\omega(\widetilde{f}_0, \widehat{b}_1, \widehat{\beta}_{10}, \widehat{\varrho})}\left[\widehat{b}_1^{-1} + (1 - \widehat{b}_1)^{-1}\right.\right.\right.$$

$$\left.\left.\left. - \frac{\mu(\widetilde{f}_0, \widehat{b}_1)}{\omega(\widetilde{f}_0, \widehat{b}_1, \widehat{\beta}_{10}, \widehat{\varrho})}\left\{\widehat{b}_1^{-1}\widehat{\beta}_{10} - (1 - \widehat{b}_1)^{-1}(\widehat{\varrho} - \widehat{\beta}_{10})\right\}\right]\middle| R = 0, A = 0\right)\right|$$

$$\leq c_1 \widehat{E}\{|\zeta_0(\mathbf{x})||R = 0, A = 0\}.$$

It follows from Assumption 4.1 with probability at least $1 - \delta$ it holds $\sup_{i \in [n_{0.0}]}\|\widehat{\boldsymbol{f}}(\mathbf{x}_i) - \boldsymbol{f}(\mathbf{x}_i)\|_2 \leq cr_{n_{1.0}}\sqrt{\log(n_{0.0})\log(1/\delta)}$, we conclude that

$$|\langle \zeta_0, \partial_{f_0}\partial_{\beta_{10}}\widehat{L}(\widetilde{f}_0, \widehat{b}_1, \widehat{\beta}_{10}, \widehat{\varrho})\rangle| \leq cc_1 r_{n_{1.0}}\sqrt{\log(n_{0.0})\log(1/\delta)}$$

holds with probability at least $1 - \delta$.

**Bound on** $\partial_{\beta_{10}}\widehat{L}(f_0, \widehat{b}_1, \widehat{\beta}_{10}, \widehat{\varrho}) - \partial_{\beta_{10}}\widehat{L}(f_0, b_1, \widehat{\beta}_{10}, \widehat{\varrho}).$

Using the taylor expansion, we have

$$\partial_{\beta_{10}}\widehat{L}(f_0, \widehat{b}_1, \widehat{\beta}_{10}, \widehat{\varrho}) - \partial_{\beta_{10}}\widehat{L}(f_0, b_1, \widehat{\beta}_{10}, \widehat{\varrho}) = \langle \widehat{b}_1 - b_1, \partial_{b_1}\partial_{\beta_{10}}\widehat{L}(f_0, \widetilde{b}_1, \widehat{\beta}_{10}, \widehat{\varrho})\rangle$$

$$= \left\langle \widehat{b}_1 - b_1, \partial_{b_1}\left[\widehat{E}\left\{\frac{\mu(f_0, \widetilde{b}_1)}{\omega(f_0, \widetilde{b}_1, \widehat{\beta}_{10}, \widehat{\varrho})}\middle| R = 0, A = 0\right\}\right]\right\rangle$$

$$= \widehat{E}\left\{(\widehat{b}_1 - b_1)\left[\frac{-\xi_0(\mathbf{x})\widetilde{b}_1^{-2} - \{1 - \xi_0(\mathbf{x})\}(1 - \widetilde{b}_1)^{-2}}{\omega(f_0, \widetilde{b}_1, \widehat{\beta}_{10}, \widehat{\varrho})}\right.\right.$$

$$\left.\left. - \frac{\mu(f_0, \widetilde{b}_1)}{\omega(f_0, \widetilde{b}_1, \widehat{\beta}_{10}, \widehat{\varrho})}\frac{-\xi_0(\mathbf{x})\widetilde{b}_1^{-2}\widehat{\beta}_{10} + \{1 - \xi_0(\mathbf{x})\}(1 - \widetilde{b}_1)^{-2}(\widehat{\varrho} - \widehat{\beta}_{10})}{\omega(f_0, \widetilde{b}_1, \widehat{\beta}_{10}, \widehat{\varrho})}\right]\right\}.$$

Assume $\varrho - \epsilon > \beta_{10} > \epsilon > 0$ and $1 - \epsilon_1 > b_1 > \epsilon_1 > 0$, i.e., there exist a $c_2 > 0$ such that $\left|\left[\frac{-\xi_0(\mathbf{x})\widetilde{b}_1^{-2} - \{1-\xi_0(\mathbf{x})\}(1-\widetilde{b}_1)^{-2}}{\omega(f_0, \widetilde{b}_1, \widehat{\beta}_{10}, \widehat{\varrho})} - \frac{\mu(f_0, \widetilde{b}_1)}{\omega(f_0, \widetilde{b}_1, \widehat{\beta}_{10}, \widehat{\varrho})}\frac{-\xi_0(\mathbf{x})\widetilde{b}_1^{-2}\widehat{\beta}_{10} + \{1-\xi_0(\mathbf{x})\}(1-\widetilde{b}_1)^{-2}(\widehat{\varrho}-\widehat{\beta}_{10})}{\omega(f_0, \widetilde{b}_1, \widehat{\beta}_{10}, \widehat{\varrho})}\right]\right| < c_2$. This implies the followings: we have

$$\left|\widehat{E}\left\{(\widehat{b}_1 - b_1)\left[\frac{-\xi_0(\mathbf{x})\widetilde{b}_1^{-2} - \{1 - \xi_0(\mathbf{x})\}(1 - \widetilde{b}_1)^{-2}}{\omega(f_0, \widetilde{b}_1, \widehat{\beta}_{10}, \widehat{\varrho})}\right.\right.\right.$$

$$\left.\left.\left. - \frac{\mu(f_0, \widetilde{b}_1)}{\omega(f_0, \widetilde{b}_1, \widehat{\beta}_{10}, \widehat{\varrho})}\frac{-\xi_0(\mathbf{x})\widetilde{b}_1^{-2}\widehat{\beta}_{10} + \{1 - \xi_0(\mathbf{x})\}(1 - \widetilde{b}_1)^{-2}(\widehat{\varrho} - \widehat{\beta}_{10})}{\omega(f_0, \widetilde{b}_1, \widehat{\beta}_{10}, \widehat{\varrho})}\right]\right\}\right|$$

$$\leq c_2|\widehat{b}_1 - b_1|.$$

We apply Hoeffing's concentration inequality for a sample mean of i.i.d. sub-gaussian random variable $Y_i$ and obtain a $c_3 > 0$ such that for any $\delta > 0$ with probability at least $1 - \delta$ it holds

$$|\widehat{b}_1 - b_1| = |\widehat{\mathrm{pr}}(Y = 1|R = 1, A = 0) - \mathrm{pr}(Y = 1|R = 1, A = 0)| \leq c_3 \sqrt{\frac{\log(1/\delta)}{n_{1\cdot0}}}.$$

**Bound on** $\partial_{\beta_{10}}\widehat{L}(f_0, b_1, \widehat{\beta}_{10}, \widehat{\varrho}) - \partial_{\beta_{10}}\widehat{L}(f_0, b_1, \widehat{\beta}_{10}, \varrho)$.

Using the taylor expansion, we have

$$\partial_{\beta_{10}}\widehat{L}(f_0, b_1, \widehat{\beta}_{10}, \widehat{\varrho}) - \partial_{\beta_{10}}\widehat{L}(f_0, b_1, \widehat{\beta}_{10}, \varrho) = \langle \widehat{\varrho} - \varrho, \partial_\varrho \partial_{\beta_{10}}\widehat{L}(f_0, b_1, \widehat{\beta}_{10}, \widetilde{\varrho}) \rangle$$

$$= \widehat{E}\left[(\widehat{\varrho} - \varrho) \frac{-\mu(f_0, b_1)\{1 - \xi_0(\mathbf{x})\}(1 - b_1)^{-1}}{\omega^2(f_0, b_1, \widehat{\beta}_{10}, \widetilde{\varrho})}\bigg| R = 0, A = 0\right].$$

Assume $\varrho - \epsilon > \beta_{10} > \epsilon > 0$ and $1 - \epsilon_1 > b_1 > \epsilon_1 > 0$, i.e., there exist a $c_4 > 0$ such that $\left|\frac{-\mu(f_0, b_1)\{1 - \xi_0(\mathbf{x})\}(1 - b_1)^{-1}}{\omega^2(f_0, b_1, \widehat{\beta}_{10}, \widetilde{\varrho})}\right| < c_4$. This implies the followings: we have

$$\left|\widehat{E}\left[(\widehat{\varrho} - \varrho) \frac{-\mu(f_0, b_1)\{1 - \xi_0(\mathbf{x})\}(1 - b_1)^{-1}}{\omega^2(f_0, b_1, \widehat{\beta}_{10}, \widetilde{\varrho})}\bigg| R = 0, A = 0\right]\right| \leq c_4|\widehat{\varrho} - \varrho|.$$

We apply Hoeffing's concentration inequality for a sample mean of i.i.d. sub-gaussian random variable $A_i$ and obtain a $c_5 > 0$ such that for any $\delta > 0$ with probability at least $1 - \delta$ it holds

$$|\widehat{\varrho} - \varrho| = |\widehat{\mathrm{pr}}(A = 0|R = 0) - \mathrm{pr}(A = 0|R = 0)| \leq c_5 \sqrt{\frac{\log(1/\delta)}{n_0}}.$$

**The term** $\partial_{\beta_{10}}\widehat{L}(f_0, \widehat{b}_1, \widehat{\beta}_{10}, \widehat{\varrho})$: We have

$$\partial_{\beta_{10}}\widehat{L}(f_0, \widehat{b}_1, \widehat{\beta}_{10}, \widehat{\varrho}) - \partial_{\beta_{10}}\widehat{L}(f_0, b_1, \widehat{\beta}_{10}, \varrho) + \partial_{\beta_{10}}\widehat{L}(f_0, b_1, \widehat{\beta}_{10}, \varrho)$$

$$= \partial_{\beta_{10}}\widehat{L}(f_0, b_1, \widehat{\beta}_{10}, \varrho) + O_p(|\widehat{b}_1 - b_1| + |\widehat{\varrho} - \varrho|).$$

Now, we study the term $\partial_{\beta_{10}}\widehat{L}(f_0, b_1, \widehat{\beta}_{10}, \varrho)$, use strong convexity of $-L(f_0, b_1, \beta_{10}, \varrho)$ with $\beta_{10}$ and the convergence of the loss that

$$\sup_{\beta_{10} \in (0, \varrho)} |\widehat{L}(f_0, b_1, \beta_{10}, \varrho) - L(f_0, b_1, \beta_{10}, \varrho)| \xrightarrow{n_{0\cdot0} \to \infty} 0$$

for $\beta_{10} \in (0, \varrho)$ in Wellner et al. (2013)(see Corollary 3.2.3) to conclude that $\widehat{\beta}_{10} \to \beta_{10}$ in probability and hence $\widehat{\beta}_{10}$ is a consistent estimator for $\beta_{10}$.

Following the consistency of $\widehat{\beta}_{10}$ we see that for sufficiently large $n_{0\cdot0}$, we have $|\widehat{\beta}_{10} - \beta_{10}| \leq \delta_\beta(\delta_\beta$ is chosen bound by $\frac{\beta_{10}}{2} \wedge \frac{\varrho - \beta_{10}}{2})$ with probability at least $1 - \delta$ and on the event it holds: $\widehat{\beta}_{10} \in [\beta_{10} - \delta_\beta, \beta_{10} + \delta_\beta]$. We define empirical process

$$Z_{n_{0\cdot0}} = \sup_{\beta \in [\beta_{10} - \delta_\beta, \beta_{10} + \delta_\beta]} |\partial_\beta \widehat{L}(f_0, b_1, \beta, \varrho) - \partial_\beta L(f_0, b_1, \beta, \varrho)|$$

for which we shall provide a high probability upper bound. We denote $Z_{n_{0\cdot0}}(\beta) = \partial_\beta \widehat{L}(f_0, b_1, \beta, \varrho) - \partial_\beta L(f_0, b_1, \beta, \varrho)$ and notice that

$$\partial_\beta \widehat{L}(f_0, b_1, \beta, \varrho) - \partial_\beta L(f_0, b_1, \beta, \varrho)$$

$$= \widehat{E}\left\{\frac{\mu(f_0, b_1)}{\omega(f_0, b_1, \beta, \varrho)}\bigg| R = 0, A = 0\right\} - E\left\{\frac{\mu(f_0, b_1)}{\omega(f_0, b_1, \beta, \varrho)}\bigg| R = 0, A = 0\right\} := A(\beta)$$

where to bound $A(\beta)$ we notice that $\frac{\mu(f_0, b_1)}{\omega(f_0, b_1, \beta, \varrho)}$ are i.i.d. and bounded by $c_0(\beta^{-1} + (\varrho - \beta)^{-1} \leq \frac{2}{\beta_{10}} + \frac{2}{\varrho - \beta_{10}} \leq c_0$ for all $\mathbf{x} \in \mathcal{X}$) and hence sub-gaussian. We apply Hoeffding's concentration

inequality for a sample mean if i.i.d. sub-gaussian random variables and obtain a constant $c_6 > 0$ such that for any $\delta > 0$ with probability at least $1 - \delta$ it holds

$$
\begin{aligned}
A(\beta) &= \widehat{E}\left\{\frac{\mu(f_0, b_1)}{\omega(f_0, b_1, \beta, \varrho)}\middle| R = 0, A = 0\right\} - E\left\{\frac{\mu(f_0, b_1)}{\omega(f_0, b_1, \beta, \varrho)}\middle| R = 0, A = 0\right\} \\
&\leq c_0 c_6 \sqrt{\frac{\log(1/\delta)}{n_{0\cdot 0}}}.
\end{aligned}
$$

Use chained arguments for $\ell_1$ with interval length $2\delta_\beta$ we obtain a uniform bound as the following: there exists a constant $c_7 > 0$ such that for any $\delta > 0$ with probability at least $1 - \delta$ if it holds

$$
\sup_{\beta \in [\beta_{10} - \delta_\beta, \beta_{10} + \delta_\beta]} A(\beta) \leq c_0 c_6 c_7 \sqrt{\frac{\log(1/\delta)}{n_{0\cdot 0}}}.
$$

Therefore, with probability at least $1 - \delta$, we have

$$
Z_{n_{0\cdot 0}} \leq c_0 c_6 c_7 \sqrt{\frac{\log(1/\delta)}{n_{0\cdot 0}}}.
$$

Returning to the first order optimality condition for estimating $\widehat{\beta}_{10}$ we notice that

$$
\begin{aligned}
0 &= (\widehat{\beta}_{10} - \beta_{10})\left\{\partial_{\beta_{10}}\widehat{L}(f_0, \widehat{b}_1, \widehat{\beta}_{10}, \widehat{\varrho}) + \langle \widehat{f}_0 - f_0, \partial_{f_0}\partial_{\beta_{10}}\widehat{L}(\widetilde{f}_0, \widehat{b}_1, \widehat{\beta}_{10}, \widehat{\varrho})\rangle\right\} \\
&= (\widehat{\beta}_{10} - \beta_{10})\left\{\partial_{\beta_{10}}\widehat{L}(f_0, \widehat{b}_1, \widehat{\beta}_{10}, \widehat{\varrho}) - \partial_{\beta_{10}}\widehat{L}(f_0, b_1, \widehat{\beta}_{10}, \varrho) + \partial_{\beta_{10}}\widehat{L}(f_0, b_1, \widehat{\beta}_{10}, \varrho)\right. \\
&\quad \left. + \langle \widehat{f}_0 - f_0, \partial_{f_0}\partial_{\beta_{10}}\widehat{L}(\widetilde{f}_0, \widehat{b}_1, \widehat{\beta}_{10}, \widehat{\varrho})\rangle\right\} \\
&= (\widehat{\beta}_{10} - \beta_{10})\partial_{\beta_{10}}L(f_0, b_1, \widehat{\beta}_{10}, \varrho) \\
&\quad + (\widehat{\beta}_{10} - \beta_{10})\left\{\partial_{\beta_{10}}\widehat{L}(f_0, \widehat{b}_1, \widehat{\beta}_{10}, \widehat{\varrho}) - \partial_{\beta_{10}}\widehat{L}(f_0, b_1, \widehat{\beta}_{10}, \varrho) + Z_{n_{0\cdot 0}}(\widehat{\beta}_{10})\right. \\
&\quad \left. + \langle \widehat{f}_0 - f_0, \partial_{f_0}\partial_{\beta_{10}}\widehat{L}(\widetilde{f}_0, \widehat{b}_1, \widehat{\beta}_{10}, \widehat{\varrho})\rangle\right\}.
\end{aligned}
$$

We combine it with the first order optimality condition for $\boldsymbol{\beta}$ to obtain

$$
\begin{aligned}
&(\widehat{\beta}_{10} - \beta_{10})\left\{\partial_{\beta_{10}}L(f_0, b_1, \widehat{\beta}_{10}, \varrho) - \partial_{\beta_{10}}L(f_0, b_1, \beta_{10}, \varrho)\right\} \\
&+ (\widehat{\beta}_{10} - \beta_{10})\left\{\partial_{\beta_{10}}\widehat{L}(f_0, \widehat{b}_1, \widehat{\beta}_{10}, \widehat{\varrho}) - \partial_{\beta_{10}}\widehat{L}(f_0, b_1, \widehat{\beta}_{10}, \varrho) + Z_{n_{0\cdot 0}}(\widehat{\beta}_{10})\right. \\
&\left. + \langle \widehat{f}_0 - f_0, \partial_{f_0}\partial_{\beta_{10}}\widehat{L}(\widetilde{f}_0, \widehat{b}_1, \widehat{\beta}, \widehat{\varrho})\rangle\right\} = 0,
\end{aligned}
$$

which can be rewritten as

$$
\begin{aligned}
&-(\widehat{\beta}_{10} - \beta_{10})\left\{\partial_{\beta_{10}}L(f_0, b_1, \widehat{\beta}_{10}, \varrho) - \partial_{\beta_{10}}L(f_0, b_1, \beta_{10}, \varrho)\right\} \quad (18) \\
&= (\widehat{\beta}_{10} - \beta_{10})\left\{\partial_{\beta_{10}}\widehat{L}(f_0, \widehat{b}_1, \widehat{\beta}_{10}, \widehat{\varrho}) - \partial_{\beta_{10}}\widehat{L}(f_0, b_1, \widehat{\beta}_{10}, \varrho) + Z_{n_{0\cdot 0}}(\widehat{\beta}_{10})\right. \\
&\quad \left. + \langle \widehat{f}_0 - f_0, \partial_{f_0}\partial_{\beta_{10}}\widehat{L}(\widetilde{f}_0, \widehat{b}_1, \widehat{\beta}_{10}, \widehat{\varrho})\rangle\right\}.
\end{aligned}
$$

Using the strong convexity of function $-L$ at $\beta_{10}$, we obtain that the left-hand side in the above equation is lower bounded as

$$
-(\widehat{\beta}_{10} - \beta_{10})\left\{\partial_{\beta_{10}}L(f_0, b_1, \widehat{\beta}_{10}, \varrho) - \partial_{\beta_{10}}L(f_0, b_1, \beta_{10}, \varrho)\right\} \geq \mu(\widehat{\beta}_{10} - \beta_{10})^2. \quad (19)
$$

Let $\mathcal{E}$ be the event on which the following hold:

- $|\widehat{\beta}_{10} - \beta_{10}| \leq \delta_\beta$.

- $|\langle \widehat{f}_0 - f_0, \partial_{f_0}\partial_{\beta_{10}}\widehat{L}(\widetilde{f}_0, \widehat{b}_1, \widehat{\beta}_{10}, \widehat{\varrho})\rangle| \leq cc_1 r_{n_{1\cdot 0}}\sqrt{\log(n_{0\cdot 0})\log(1/\delta)}$.

- $Z_{n_{0 \cdot 0}} \le c_0 c_6 c_7 \sqrt{\frac{\log(1/\delta)}{n_{0 \cdot 0}}}$.

- $|\partial_{\beta_{10}} \widehat{L}(f_0, \widehat{b}_1, \widehat{\beta}_{10}, \widehat{\varrho}) - \partial_{\beta_{10}} \widehat{L}(f_0, b_1, \widehat{\beta}_{10}, \varrho)| \le (c_2 c_3 + c_4 c_5) \left\{ \sqrt{\frac{\log(1/\delta)}{n_{1 \cdot 0}}} + \sqrt{\frac{\log(1/\delta)}{n_0}} \right\}$.

We notice that the event $\mathcal{E}$ has probability $1 - 5\delta$. Under the event there exists a $c_8 > 0$ such that the right-hand side in (18) is upper bounded as

$$
\begin{aligned}
& \left| (\widehat{\beta}_{10} - \beta_{10}) \left\{ \partial_{\beta_{10}} \widehat{L}(f_0, \widehat{b}_1, \widehat{\beta}_{10}, \widehat{\varrho}) - \partial_{\beta_{10}} \widehat{L}(f_0, b_1, \widehat{\beta}_{10}, \varrho) + Z_{n_{0 \cdot 0}}(\widehat{\beta}_{10}) \right. \right. \\
& \left. \left. \qquad + \langle \widehat{f}_0 - f_0, \partial_{f_0} \partial_{\beta_{10}} \widehat{L}(\widetilde{f}_0, \widehat{b}_1, \widehat{\beta}_{10}, \varrho) \rangle \right\} \right| \\[4pt]
& \le \ |\widehat{\beta}_{10} - \beta_{10}| \left\{ |\partial_{\beta_{10}} \widehat{L}(f_0, \widehat{b}_1, \widehat{\beta}_{10}, \widehat{\varrho}) - \partial_{\beta_{10}} \widehat{L}(f_0, b_1, \widehat{\beta}_{10}, \varrho)| + |Z_{n_{0 \cdot 0}}(\widehat{\beta}_{10})| \right. \\
& \left. \qquad + |\langle \widehat{f}_0 - f_0, \partial_{f_0} \partial_{\beta_{10}} \widehat{L}(\widetilde{f}_0, \widehat{b}_1, \widehat{\beta}_{10}, \widehat{\varrho}) \rangle| \right\} \\[4pt]
& \le \ c_8 \left\{ r_{n_{1 \cdot 0}} \sqrt{\log(n_{0 \cdot 0})\log(1/\delta)} + \sqrt{\frac{\log(1/\delta)}{n_{1 \cdot 0}}} + \sqrt{\frac{\log(1/\delta)}{n_0}} + \sqrt{\frac{\log(1/\delta)}{n_{0 \cdot 0}}} \right\} |\widehat{\beta}_{10} - \beta_{10}|.
\end{aligned}
\tag{20}
$$

Combining the bounds (19) and (20) for left and right hand sides we obtain a $c_{10} > 0$ such that on the event $\mathcal{E}$ it holds

$$
|\widehat{\beta}_{10} - \beta_{10}| \le c_{10} \left\{ r_{n_{1 \cdot 0}} \sqrt{\log(n_{0 \cdot 0})\log(1/\delta)} + \sqrt{\frac{\log(1/\delta)}{n_{1 \cdot 0}}} + \sqrt{\frac{\log(1/\delta)}{n_0}} + \sqrt{\frac{\log(1/\delta)}{n_{0 \cdot 0}}} \right\}.
$$

Further, sine $A_i$ is bound random variable, then we can obtain a constant $c_9 > 0$ such that for any $\delta > 0$ with probability at least $1 - \delta$ it holds

$$
|\widehat{\beta}_{00} - \beta_{00}| = |(\widehat{\varrho} - \widehat{\beta}_{10}) - (\varrho - \beta_{10})| \le |\widehat{\varrho} - \varrho| + |\widehat{\beta}_{10} - \beta_{10}| \le c_9 \sqrt{\frac{\log(1/\delta)}{n_0}} + |\widehat{\beta}_{10} - \beta_{10}|.
$$

In summary, we have a constant $c_{10} > 0$ such that for any $\delta > 0$ with probability at least $1 - 6\delta$ it holds

$$
\|\widehat{\boldsymbol{\beta}} - \boldsymbol{\beta}\|_1 \le c_{10} \left\{ r_{n_{1 \cdot 0}} \sqrt{\log(n_{0 \cdot 0})\log(1/\delta)} + \sqrt{\frac{\log(1/\delta)}{n_{1 \cdot 0}}} + \sqrt{\frac{\log(1/\delta)}{n_0}} + + \sqrt{\frac{\log(1/\delta)}{n_{0 \cdot 0}}} \right\}.
$$

$\qquad\qquad\qquad\qquad\qquad\qquad\qquad\qquad\qquad\qquad\qquad\qquad\qquad\qquad\qquad\qquad\qquad\quad \square$

**Lemma A.1.** *(Derivatives).Th following holds:*

- $\partial_{f_0}(\xi_0) = 2\xi_0(1 - \xi_0)$;

- $\partial_{f_0}\{\mu(f_0, b_1)\} = \partial_{f_0}(\xi_0)\{b_1^{-1} + (1 - b_1)^{-1}\}$;

- $\partial_{f_0}\{\omega(f_1, b_1, \beta_{10}, \varrho)\} = \partial_{f_0}(\xi_0)\{b_1^{-1}\beta_{10} - (1 - b_1)^{-1}(\varrho - \beta_{10})\}$;

- $\partial_{f_0} \left\{ \frac{\mu(f_0, b_1)}{\omega(f_0, b_1, \beta_{10}, \varrho)} \right\} = \frac{2\xi_0(1 - \xi_0)}{\omega(f_0, b_1, \beta_{10}, \varrho)} \left[ b_1^{-1} + (1 - b_1)^{-1} - \frac{\mu(f_0, b_1)\{b_1^{-1}\beta_{10} - (1 - b_1)^{-1}(\varrho - \beta_{10})\}}{\omega(f_0, b_1, \beta_{10}, \varrho)} \right]$.

*Proof.*

$$
\begin{aligned}
\partial_{f_0}\xi_0 &= \partial_{f_0} \left( \frac{e^{f_0}}{e^{f_0} + e^{-f_0}} \right) = \frac{e^{f_0}\left(e^{f_0} + e^{-f_0}\right) - e^{f_0}\left(e^{f_0} - e^{-f_0}\right)}{\left(e^{f_0} + e^{-f_0}\right)^2} = 2\xi_0(1 - \xi_0), \\
\partial_{f_0}\{\mu(f_0, b_1)\} &= \partial_{f_0}(\xi_0)\{b_1^{-1} + (1 - b_1)^{-1}\}, \\
\partial_{f_0}\{\omega(f_0, b_1, \beta_{10}, \varrho)\} &= \partial_{f_0}(\xi_0)b_1^{-1}\beta_{10} - \partial_{f_0}(\xi_0)(1 - b_1)^{-1}(\varrho - \beta_{10}) \\
&= \partial_{f_0}(\xi_0)\{b_1^{-1}\beta_{10} - (1 - b_1)^{-1}(\varrho - \beta_{10})\}.
\end{aligned}
$$

Thus,

$$\partial_{f_0}\left\{\frac{\mu(f_0, b_1)}{\omega(f_0, b_1, \beta_{10}, \varrho)}\right\}$$

$$= \frac{\partial_{f_0}(\xi_0)[\{b_1^{-1} + (1-b_1)^{-1}\}\omega(f_0, b_1, \beta_{10}, \varrho) - \mu(f_0, b_1)\{b_1^{-1}\beta_{10} - (1-b_1)^{-1}(\varrho - \beta_{10})\}]}{\omega^2(f_0, b_1, \beta_{10}, \varrho)}$$

$$= \frac{\partial_{f_0}(\xi_0)}{\omega(f_0, b_1, \beta_{10}, \varrho)}\left[b_1^{-1} + (1-b_1)^{-1} - \frac{\mu(f_0, b_1)\{b_1^{-1}\beta_{10} - (1-b_1)^{-1}(\varrho - \beta_{10})\}}{\omega(f_0, b_1, \beta_{10}, \varrho)}\right]$$

$$= \frac{2\xi_0(1-\xi_0)}{\omega(f_0, b_1, \beta_{10}, \varrho)}\left[b_1^{-1} + (1-b_1)^{-1} - \frac{\mu(f_0, b_1)\{b_1^{-1}\beta_{10} - (1-b_1)^{-1}(\varrho - \beta_{10})\}}{\omega(f_0, b_1, \beta_{10}, \varrho)}\right].$$

□

*Proof of Proposition 4.4.* Define $w(y) = \frac{pr(y|A=0,R=0)}{pr(y|A=0,R=1)}$,

$$\begin{aligned}
\mathcal{L}_0(h) &= \mathrm{E}[\ell\{h(\mathbf{X}), Y\}|R=0, A=0]\\
&= \int \ell\{h(\mathbf{X}), Y\}p(\mathbf{x}, y|A=0, R=0)\mathbf{dx}dy\\
&= \int \ell\{h(\mathbf{X}), Y\}\frac{p(\mathbf{x}, y|A=0, R=0)}{p(\mathbf{x}, y|A=0, R=1)}p(\mathbf{x}, y|A=0, R=1)\mathbf{dx}dy\\
&= \int \ell\{h(\mathbf{x}), y\}\frac{pr(y|A=0, R=0)}{pr(y|A=0, R=1)}pr(\mathbf{x}, y|A=0, R=1)\mathbf{dx}dy\\
&= \mathrm{E}\left[\ell\{h(\mathbf{X}), Y\}w(Y)\big|A=0, R=1\right] =: \mathcal{L}_1(h, w).
\end{aligned}$$

Let $\mathcal{L}_1(h, w) = \mathrm{E}\left[\ell\{h(\mathbf{X}), Y\}w(Y)\big|A=0, R=1\right]$, then we have

$$\begin{aligned}
\mathcal{L}_0(\widehat{h}) - \mathcal{L}_0(h) &= \mathcal{L}_1(\widehat{h}, w) - \mathcal{L}_1(h, w)\\
&= \underbrace{\mathcal{L}_1(\widehat{h}, w) - \widehat{\mathcal{L}}_1(\widehat{h}, w)}_{(a)} + \underbrace{\widehat{\mathcal{L}}_1(\widehat{h}, w) - \widehat{\mathcal{L}}_1(\widehat{h}, \widehat{w})}_{(b)} \qquad (21)\\
&\quad + \underbrace{\widehat{\mathcal{L}}_1(\widehat{h}, \widehat{w}) - \widehat{\mathcal{L}}_1(h, \widehat{w})}_{\leq 0} + \underbrace{\widehat{\mathcal{L}}_1(h, \widehat{w}) - \widehat{\mathcal{L}}_1(h, w)}_{(c)} + \underbrace{\widehat{\mathcal{L}}_1(h, w) - \mathcal{L}_1(h, w)}_{(d)},
\end{aligned}$$

where $\widehat{h} \equiv \widehat{h}_{\widehat{w}}$.

**Uniform bound on (a)** To control (a) in (21) we establish a concentration bound on the following generalization error

$$\sup_{g \in \mathcal{F}}\{\mathcal{L}_1(g, w) - \widehat{\mathcal{L}}_1(g, w)\}$$

$$= \sup_{g \in \mathcal{F}}\left\{E\left[\ell\{g(\mathbf{X}), Y\}w(Y)\big|A=0, R=1\right] - \widehat{E}\left[\ell\{g(\mathbf{X}), Y\}w(Y)\big|A=0, R=1\right]\right\}$$

$$=: F(\mathbf{Z}_{1:n_{1\cdot 0}})$$

where, for $i > 1$ we denote $\mathbf{Z}_{1:i} = (\mathbf{Z}_1, \cdots, \mathbf{Z}_i)$ and $\mathbf{Z}_i = (\mathbf{X}_i, Y_i)$. First, we use a modification of McDiarmid concentration inequality to bound $F(\mathbf{Z}_{1:n_{1\cdot 0}})$ in terms of its expectation and a $O_p(1/\sqrt{n_{1\cdot 0}})$ term, as elucidated in the following lemma.

**Lemma A.2.** *There exists a constant $c_1 > 0$ such that with probability at least $1 - \delta$ the following holds*

$$F(\mathbf{Z}_{1:n_{1\cdot 0}}) \leq E\{F(\mathbf{Z}_{1:n_{1\cdot 0}})\} + c_1\sqrt{\frac{\log(1/\delta)}{n_{1\cdot 0}}}. \qquad (22)$$

*The proof is similar to Lemma A.3 of Maity et al. (2022), so we omit it.*

Next, we use a symmetrization argument (see Wellner et al. (2013), Chapter 2, Lemma 2.3.1) to bound the expectation $E\{F(\mathbf{Z}_{1:n_{1 \cdot 0}})\}$ by the Rademacher complexity of the hypothesis class $\mathcal{G}$, i.e.,

$$E\{F(\mathbf{Z}_{1:n_{1 \cdot 0}})\} \leq 2\mathcal{R}_{n_{1 \cdot 0}}(\mathcal{G}). \tag{23}$$

Combining (22) and (23) we obtain

$$(a) \leq 2\mathcal{R}_{n_{1 \cdot 0}}(\mathcal{G}) + c_1 \sqrt{\frac{\log(1/\delta)}{n_{1 \cdot 0}}} \tag{24}$$

with probability at least $1 - \delta$.

**Uniform bound on (b) and (c)** Denoting $\mathbf{Z}_i = (\mathbf{X}_i, Y_i)$ and $\ell_g(\mathbf{Z}_i) = \ell\{g(\mathbf{X}_i), Y_i\}$ we notice that for any $g \in \mathcal{F}$ we have

$$|\widehat{\mathcal{L}}_1(g, w) - \widehat{\mathcal{L}}_1(g, \widehat{w})|$$
$$= |\widehat{\mathrm{E}}\left[\ell\{g(\mathbf{X}), Y\}\{w(Y) - \widehat{w}(Y)\} \mid A = 0, R = 1\right]| \leq \frac{\|\ell_g\|_\infty}{n_{1 \cdot 0}} \sum_{i=1}^{n_{1 \cdot 0}} |w(y_i) - \widehat{w}(y_i)|.$$

Since $w(y) - \widehat{w}(y)$ is a sub-gaussian random variable, we use sub-gaussian concentration to establish that for some constant $c_2 > 0$,

$$\text{for any } g \in \mathcal{F}, |\widehat{\mathcal{L}}_1(g, w) - \widehat{\mathcal{L}}_1(g, \widehat{w})| \leq \|\ell_g\|_\infty \left\{ \mathrm{E}_Y |w(Y) - \widehat{w}(Y)| + c_2 \sqrt{\frac{\log(1/\delta)}{n_{1 \cdot 0}}} \right\}$$

with probability at least $1 - \delta$. This provides a simultaneous bound (on the same probability event) for both (b) and (c) with $g = \widehat{h}$ and $g = h$. Further, by Lemma A.3, for some constants $C_1$ and $C_2$ and any $g \in \mathcal{F}$, we have

$$|\widehat{\mathcal{L}}_1(g, w) - \widehat{\mathcal{L}}_1(g, \widehat{w})| \tag{25}$$
$$\leq \|\ell_g\|_\infty \left\{ C_1 \|\widehat{\boldsymbol{\beta}} - \boldsymbol{\beta}\|_1 + C_2 \left( \sqrt{\frac{\log(1/\delta)}{n_0}} + \sqrt{\frac{\log(1/\delta)}{n_1}} \right) + c_2 \sqrt{\frac{\log(1/\delta)}{n_{1 \cdot 0}}} \right\}$$

with probability at least $1 - 5\delta$.

**Uniform bound on (d)** We note that

$$\widehat{\mathcal{L}}_1(h, w) - \mathcal{L}_1(h, w)$$
$$= \widehat{E}\left[\ell\{h(\mathbf{X}), Y\}w(Y)\big|A = 0, R = 1\right] - E\left[\ell\{h(\mathbf{X}), Y\}w(Y)\big|A = 0, R = 1\right]$$

where $[\ell\{h(\mathbf{X}_i), Y_i\}w(Y_i)]_{i=1}^{n_{1 \cdot 0}}$ are i.i.d sub-gaussian random variables. Using Hoeffding concentration bound we conclude that there exists a constant $c_3 > 0$ such that for any $\delta > 0$ the following holds with probability at least $1 - \delta$,

$$\widehat{\mathcal{L}}_1(h, w) - \mathcal{L}_1(h, w) \leq c_3 \sqrt{\frac{\log(1/\delta)}{n_{1 \cdot 0}}}. \tag{26}$$

Finally, using (24) on (a) (which is true on an event of probability $\geq 1 - \delta$), (25) on (b) and (c) (simultaneously true on an event of probability $1 - 5\delta$), and (26) on (d) (holds on an event of probability $\geq 1 - \delta$) we conclude that with probability at least $1 - 7\delta$ the following holds

$$\mathcal{L}_0(\widehat{h}_{\widehat{w}}) - \mathcal{L}_0(h) \leq 2\mathcal{R}_{n_{1 \cdot 0}}(\mathcal{G}) + CB\|\widehat{\boldsymbol{\beta}} - \boldsymbol{\beta}\|_1 + c\left\{ \sqrt{\frac{\log(1/\delta)}{n_{1 \cdot 0}}} + \sqrt{\frac{\log(1/\delta)}{n_0}} + \sqrt{\frac{\log(1/\delta)}{n_1}} \right\}.$$

where $c = c_1 + \|\ell_g\|_\infty(C_2 + c_2) + c_3$. $\qquad \square$

**Lemma A.3.** *Assume $|\beta_{i0}/\alpha_{i0}| \le B_1$ for any $i = 0, 1$. There exist constants $C, c_1, c_2$, such that with probability at least $1 - 4\delta$,*

$$|\widehat{w}(y) - w(y)| \le CB_1\|\widehat{\boldsymbol{\beta}} - \boldsymbol{\beta}\|_1 + CB_1(c_1 + c_2)\left(\sqrt{\frac{\log(1/\delta)}{n_0}} + \sqrt{\frac{\log(1/\delta)}{n_1}}\right).$$

*Proof.*

$$|\widehat{w}(y) - w(y)| = \left|\frac{\widehat{\mathrm{pr}}(y|A = 0, R = 0)}{\widehat{\mathrm{pr}}(y|A = 0, R = 1)} - \frac{\mathrm{pr}(y|A = 0, R = 0)}{\mathrm{pr}(y|A = 0, R = 1)}\right|$$

$$= \left|\frac{\widehat{\mathrm{pr}}(y, A = 0|R = 0)}{\widehat{\mathrm{pr}}(y, A = 0|R = 1)}\frac{\widehat{\mathrm{pr}}(A = 0|R = 1)}{\widehat{\mathrm{pr}}(A = 0|R = 0)} - \frac{\mathrm{pr}(y, A = 0|R = 0)}{\mathrm{pr}(y, A = 0|R = 1)}\frac{\mathrm{pr}(A = 0|R = 1)}{\mathrm{pr}(A = 0|R = 0)}\right|$$

$$\le \left|\left\{\frac{\widehat{\mathrm{pr}}(y, A = 0|R = 0)}{\widehat{\mathrm{pr}}(y, A = 0|R = 1)} - \frac{\mathrm{pr}(y, A = 0|R = 0)}{\mathrm{pr}(y, A = 0|R = 1)}\right\}\frac{\widehat{\mathrm{pr}}(A = 0|R = 1)}{\widehat{\mathrm{pr}}(A = 0|R = 0)}\right|$$

$$+ \left|\frac{\mathrm{pr}(y, A = 0|R = 0)}{\mathrm{pr}(y, A = 0|R = 1)}\left\{\frac{\widehat{\mathrm{pr}}(A = 0|R = 1)}{\widehat{\mathrm{pr}}(A = 0|R = 0)} - \frac{\mathrm{pr}(A = 0|R = 1)}{\mathrm{pr}(A = 0|R = 0)}\right\}\right|.$$

For the first term, we have

$$\left|\frac{\widehat{\beta}_{y0}\alpha_{y0} - \beta_{y0}\widehat{\alpha}_{y0}}{\alpha_{y0}\widehat{\alpha}_{y0}}\frac{n_{1\cdot0}}{n_{0\cdot0}}\frac{n_0}{n_1}\right|$$

$$= \left|\frac{(\widehat{\beta}_{y0} - \beta_{y0})\alpha_{y0} + \beta_{y0}(\alpha_{y0} - \widehat{\alpha}_{y0})}{\alpha_{y0}\widehat{\alpha}_{y0}}\frac{n_{1\cdot0}}{n_{0\cdot0}}\frac{n_0}{n_1}\right|$$

$$\le \left|\frac{y\alpha_{y0}}{\alpha_{y0}\widehat{\alpha}_{y0}}\frac{n_{1\cdot0}}{n_{0\cdot0}}\frac{n_0}{n_1}\right||\widehat{\beta}_{10} - \beta_{10}| + \left|\frac{(1 - y)\alpha_{y0}}{\alpha_{y0}\widehat{\alpha}_{y0}}\frac{n_{1\cdot0}}{n_{0\cdot0}}\frac{n_0}{n_1}\right||\widehat{\beta}_{00} - \beta_{00}|$$

$$+ \left|\frac{y\beta_{y0}}{\alpha_{y0}\widehat{\alpha}_{y0}}\frac{n_{1\cdot0}}{n_{0\cdot0}}\frac{n_0}{n_1}\right|(\alpha_{10} - \widehat{\alpha}_{10}) + \left|\frac{(1 - y)\beta_{y0}}{\alpha_{y0}\widehat{\alpha}_{y0}}\frac{n_{1\cdot0}}{n_{0\cdot0}}\frac{n_0}{n_1}\right|(\alpha_{00} - \widehat{\alpha}_{00})$$

$$\le CB_1(\|\widehat{\boldsymbol{\beta}} - \boldsymbol{\beta}\|_1 + |\widehat{\alpha}_{00} - \alpha_{00}| + |\widehat{\alpha}_{10} - \alpha_{10}|).$$

Here $C$ is some constant. Since $Y_i$ and $A_i$ are sub-gaussian random variables, we use sub-gaussian concentration to establish that for some constant $c > 0$,

$$|\widehat{\alpha}_{00} - \alpha_{00}| + |\widehat{\alpha}_{10} - \alpha_{10}| \le c\left\{\sqrt{\frac{\log(1/\delta)}{n_1}}\right\}$$

with probability at least $1 - 2\delta$.

For the second term, we have

$$\left|\frac{\beta_{y0}}{\alpha_{y0}}\frac{\mathrm{pr}(A = 0|R = 1)\widehat{\mathrm{pr}}(A = 0|R = 0) - \mathrm{pr}(A = 0|R = 0)\widehat{\mathrm{pr}}(A = 0|R = 1)}{\mathrm{pr}(A = 0|R = 0)\widehat{\mathrm{pr}}(A = 0|R = 0)}\right|$$

$$\le \left|\frac{\beta_{y0}}{\alpha_{y0}}\right|\frac{\mathrm{pr}(A = 0|R = 1)}{\mathrm{pr}(A = 0|R = 0)\widehat{\mathrm{pr}}(A = 0|R = 0)}|\{\widehat{\mathrm{pr}}(A = 0|R = 0) - \mathrm{pr}(A = 0|R = 0)\}|$$

$$+ \left|\frac{\beta_{y0}}{\alpha_{y0}}\right|\frac{\mathrm{pr}(A = 0|R = 0)}{\mathrm{pr}(A = 0|R = 0)\widehat{\mathrm{pr}}(A = 0|R = 0)}|\{\mathrm{pr}(A = 0|R = 1) - \widehat{\mathrm{pr}}(A = 0|R = 1)\}|$$

$$\le \left|\frac{\beta_{y0}}{\alpha_{y0}}\right|\frac{p_{1\cdot0}(1 - \pi)n_0}{p_{0\cdot0}\pi n_{0\cdot0}}|\{\widehat{\mathrm{pr}}(A = 0|R = 0) - \mathrm{pr}(A = 0|R = 0)\}|$$

$$+ \left|\frac{\beta_{y0}}{\alpha_{y0}}\right|\frac{n_0}{n_{0\cdot0}}|\{\mathrm{pr}(A = 0|R = 1) - \widehat{\mathrm{pr}}(A = 0|R = 1)\}|$$

$$\le CB_1(|\{\widehat{\mathrm{pr}}(A = 0|R = 0) - \mathrm{pr}(A = 0|R = 0)\}| + |\{\mathrm{pr}(A = 0|R = 1) - \widehat{\mathrm{pr}}(A = 0|R = 1)\}|).$$

Here $C$ is some constant. Since $A_i$ is a sub-gaussian random variable, we use sub-gaussian concentration to establish that for some constant $c > 0$,

$$|\{\widehat{\mathrm{pr}}(A = 0|R = 0) - \mathrm{pr}(A = 0|R = 0)\}| + |\{\mathrm{pr}(A = 0|R = 1) - \widehat{\mathrm{pr}}(A = 0|R = 1)\}|$$

$$\leq c \left\{ \sqrt{\frac{\log(1/\delta)}{n_0}} + \sqrt{\frac{\log(1/\delta)}{n_1}} \right\}$$

with probability at least $1 - 2\delta$. $\qquad\square$

**Remark A.4.** *Proposition 4.4 indicates that the generalization bound depends on the estimation error $\|\widehat{\boldsymbol{\beta}} - \boldsymbol{\beta}\|_1$, which can be directly controlled based on the conditions listed in Assumption 4.1, implying that different estimation procedures for $\boldsymbol{\beta}$ will yield different upper bounds. In Theorem 4.2, we established an upper bound for the estimation error of $\widehat{\boldsymbol{\beta}}$, which directly leads to a refined generalization bound for the learned classifier $\widehat{h}_{\widehat{w}}$. Specifically, for any $\delta > 0$, with probability at least $1 - 13\delta$, the following inequality holds:*

$$\mathcal{L}_0(\widehat{h}_{\widehat{w}}) - \min_{h \in \mathcal{F}} \mathcal{L}_0(h) \leq 2\mathcal{R}_{n_{1\cdot 0}}(\mathcal{G}) + dBc_{10}\chi_n \sqrt{\log(1/\delta)} + c\left\{ \sqrt{\frac{\log(1/\delta)}{n_{1\cdot 0}}} + \sqrt{\frac{\log(1/\delta)}{n_0}} \right\},$$

*where $c_{10}$ is the constant appearing in Theorem 4.2, and $\alpha_n$ characterizes the convergence rate of $\widehat{\boldsymbol{\beta}}$.*

### A.3 SYNTHETIC DATA RESULTS

We consider a structured data-generating process in which the covariates $\mathbf{X} \in \mathbf{R}^4$ are drawn from a distribution conditioned on a latent pair $(Y, A)$, where $Y \in \{0, 1\}$ denotes the class label and $A \in \{0, 1\}$ denotes the background. The generation begins by sampling $(Y, A)$ according to a predefined distribution.

In the *source* domain, we consider $(Y, A) \in \{(0, 0), (0, 1), (1, 0)\}$, each occurring with probability $1/3$. The covariates $\mathbf{X} \in \mathbf{R}^4$ are generated as $\mathbf{X} \sim N(\boldsymbol{\mu}_{\mathrm{YA}}, \mathbf{I}_4)$, where $\boldsymbol{\mu}_{\mathrm{YA}}$ denotes the mean vector for each combination and $\mathbf{I}_4$ is the $4 \times 4$ identity matrix. The stratum $(1, 1)$ is excluded from the source. In the *target* domain, all four combinations $(Y, A) \in \{0, 1\}^2$ appear with equal probability $1/4$, and $\mathbf{X}$ is drawn from the same distribution $N(\boldsymbol{\mu}_{\mathrm{YA}}, \mathbf{I}_4)$ with distinct means:

$$\boldsymbol{\mu}_{00} = (1, 0, 0, 0)^{\mathrm{T}}, \quad \boldsymbol{\mu}_{01} = (0, 0, 1, 0)^{\mathrm{T}}, \quad \boldsymbol{\mu}_{10} = (0, 1, 0, 0)^{\mathrm{T}}, \quad \boldsymbol{\mu}_{11} = (0, 0, 0, 1)^{\mathrm{T}}.$$

For model estimation, we utilize the known data-generating distribution to compute the conditional probability models required by both our proposed estimators and the benchmark method. Specifically, we calculate the five key conditional probabilities needed for implementation: $\xi_0(\mathbf{x})$, $\xi(\mathbf{x})$, $\tau_0(\mathbf{x})$, $\tau_1(\mathbf{x})$, and $\kappa(\mathbf{x})$, which together determine the models $\eta_0(\mathbf{x})$, $\eta_1(\mathbf{x})$ and $\eta(\mathbf{x})$ for our method, and $\xi_0(\mathbf{x})$, $\xi_1(\mathbf{x})$ and $\xi(\mathbf{x})$ for the benchmark. Additionally, the parameters $\{\beta_{ya} : y = 0, 1; a = 0, 1\}$ and $\{\alpha_{ya} : y = 0, 1; a = 0, 1\}$ as known. The classification threshold is set to $0.5$, consistent with the standard Bayesian decision rule—that is, a sample is classified as positive if the predicted probability exceeds $0.5$.

To assess the performance of the proposed estimators, we conduct 100 simulations for each configuration and report the results using box plots that compare $\{\widehat{\eta}_1(\mathbf{x}), \widehat{\xi}_1(\mathbf{x})\}$ and $\{\widehat{\eta}(\mathbf{x}), \widehat{\xi}(\mathbf{x})\}$ across varying sample sizes. The left panel of Figure 2 illustrates the performance of $\{\widehat{\eta}_1(\mathbf{x}), \widehat{\xi}_1(\mathbf{x})\}$ for $n_0 = 1000$ and $6000$, with $n_1$ ranging from 1000 to 8000. The right panel of Figure 2 shows the corresponding performance of $\{\widehat{\eta}(\mathbf{x}), \widehat{\xi}(\mathbf{x})\}$ under the same settings. Performance is evaluated using two standard metrics: accuracy and $F_1$ score. In both cases, the proposed estimators consistently outperform the benchmark estimators. Moreover, as $n_0$ increases, the variance of the estimators decreases, suggesting improved stability and reliability. In particular, for the $F_1$ score of $\widehat{\eta}(\mathbf{x})$, performance steadily improves as $n_0$ increases, further demonstrating the robustness and effectiveness of the proposed method. Comparable patterns are also observed when $n_1$ is fixed at 1000 and 6000 while $n_0$ varies from 1000 to 8000, as shown in Figure 3.

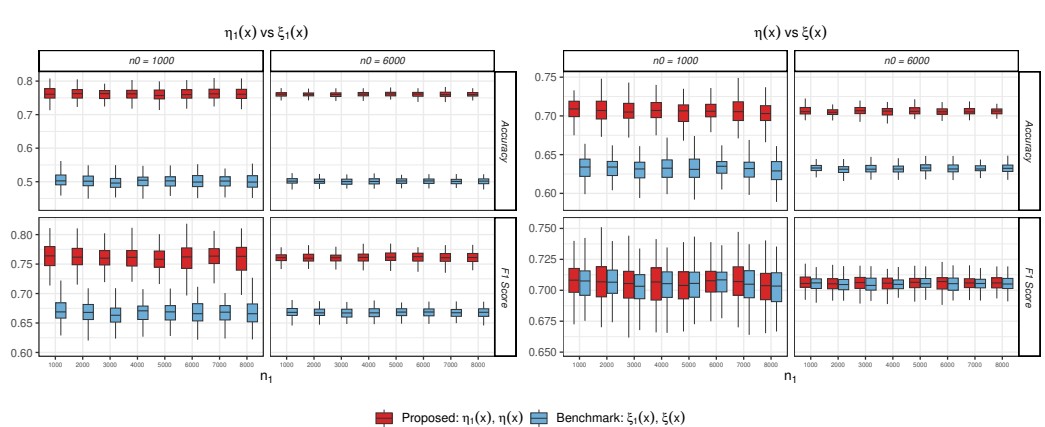

Figure 2: The left panel displays the performance of the $F_1$ score and accuracy for $\eta_1(\mathbf{x})$ and $\xi_1(\mathbf{x})$ across different scenarios, while the right panel presents the corresponding results for $\eta(\mathbf{x})$ and $\xi(\mathbf{x})$.

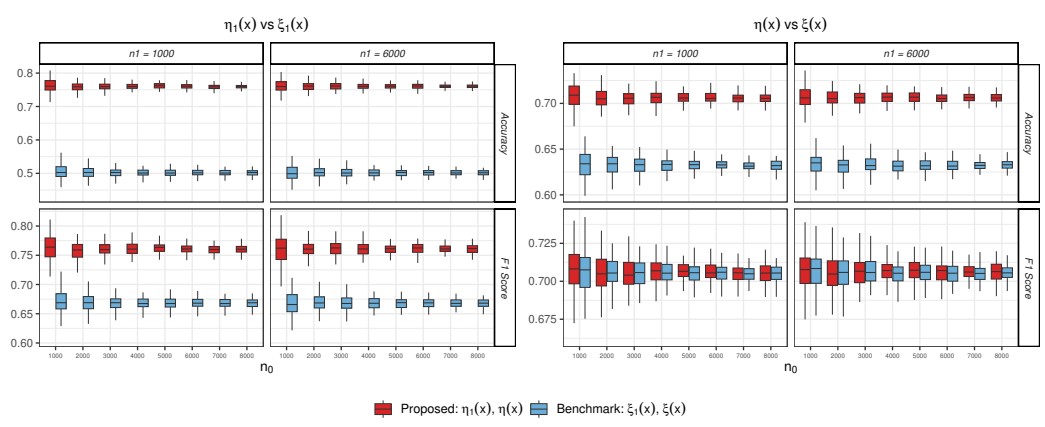

Figure 3: The left panel displays the performance of the $F_1$ score and accuracy for $\eta_1(\mathbf{x})$ and $\xi_1(\mathbf{x})$ across different scenarios, while the right panel presents the corresponding results for $\eta(\mathbf{x})$ and $\xi(\mathbf{x})$.

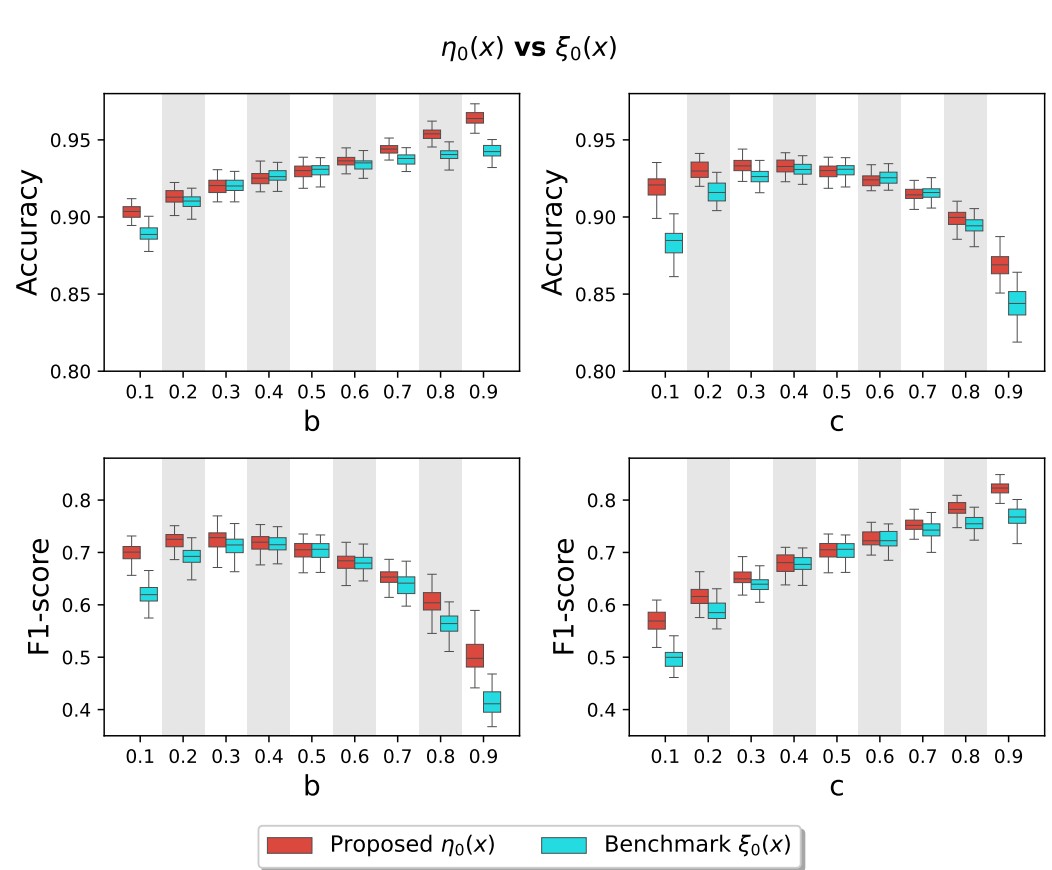

Figure 4: Performance comparison of our proposed estimator $\eta_0(\mathbf{x})$, and the benchmark method $\xi_0(\mathbf{x})$ under the setting $a = 0.5$ with either $c = 0.5$ and varying $b$ or $b = 0.5$ and varying $c$.

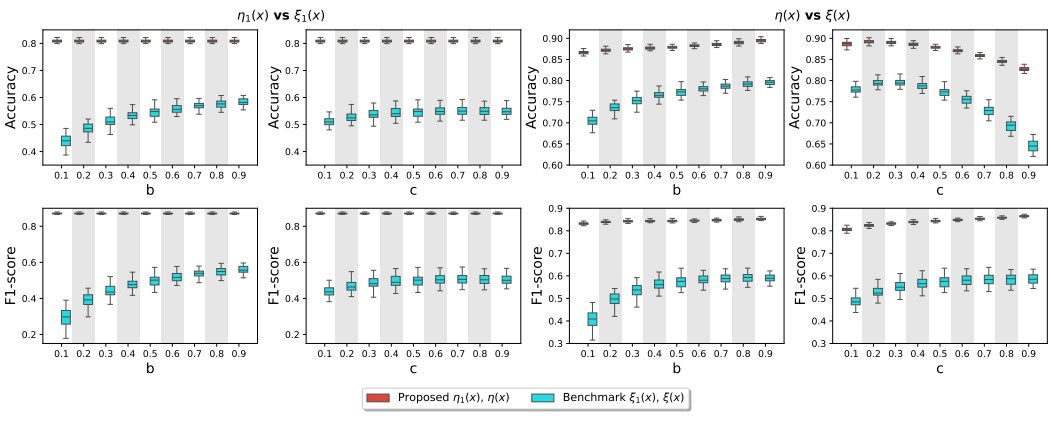

Figure 5: Performance comparison of our proposed estimators $\eta_1(\mathbf{x})$, $\eta(\mathbf{x})$, and the benchmark method $\xi_1(\mathbf{x})$, $\xi(\mathbf{x})$ under the setting $a = 0.7$ with either $c = 0.5$ and varying $b$ or $b = 0.5$ and varying $c$.

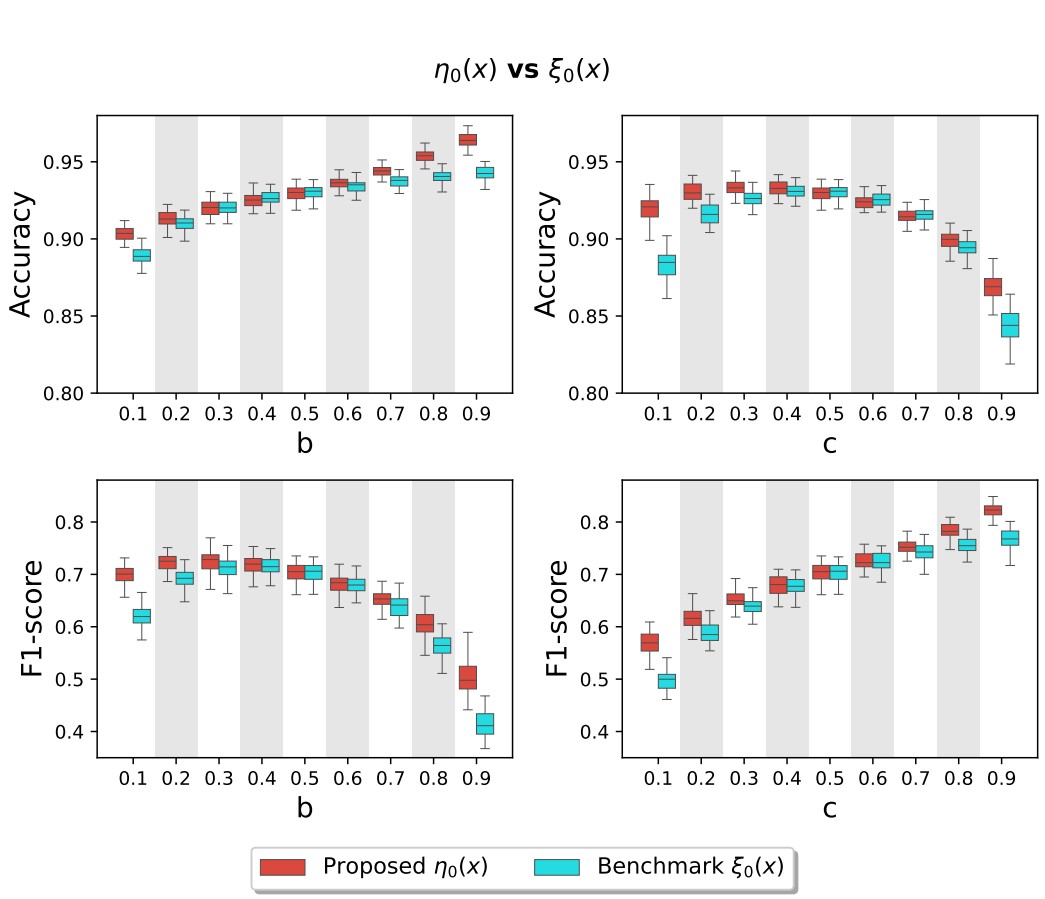

Figure 6: Performance comparison of our proposed estimator $\eta_0(\mathbf{x})$, and the benchmark method $\xi_0(\mathbf{x})$ under the setting $a = 0.7$ with either $c = 0.5$ and varying $b$ or $b = 0.5$ and varying $c$.

### A.4 ADDITIONAL BENCHMARK DATA RESULTS

In addition to the estimators analyzed in the main text, we further evaluate the performance of the proposed estimator $\widehat{\eta}_0(\mathbf{x})$ and its corresponding benchmark $\widehat{\xi}_0(\mathbf{x})$. Figure 4 presents the results with $a = 0.5$ in the source domain and systematically vary the remaining subclass inclusion rates by setting either $b = 0.5$ with $c \in \{0.1, 0.2, \ldots, 0.9\}$, or $c = 0.5$ with $b \in \{0.1, 0.2, \ldots, 0.9\}$. Performance is assessed using the $F_1$ score and accuracy across 50 independent repetitions. The proposed estimator $\widehat{\eta}_0(\mathbf{x})$ consistently demonstrates superior performance compared to the benchmark $\widehat{\xi}_0(\mathbf{x})$, achieving higher accuracy and $F_1$ scores across all configurations. Similar results are observed when $a = 0.7$ in Figures 5 and 6, further validating the robustness of our method.

### A.5 DISCUSSIONS AND CONCLUSIONS

In this paper, we introduce a novel unsupervised domain adaptation setting where an entire label-background subpopulation is absent from the source domain, a scenario motivated by real-world data collection constraints. Despite this structured missingness, we show that accurate prediction in the target domain is still achievable. We develop a theoretical framework that enables such prediction by estimating subpopulation proportions in the target through distribution matching. We provide rigorous guarantees, including statistical consistency as well as upper bounds on the target-domain prediction error. Empirically, our method outperforms standard baselines that overlook structured missingness, especially in prediction performance for the unobserved subpopulation. Overall, our framework provides a rigorous characterization of model adaptation under subpopulation structured missingness, and enables robust domain adaptation in such a challenging scenario.

Our theoretical framework is built upon structured conditional invariance and mixture proportion estimation. These tools naturally generalize to multi-class labels for $n_y$ species and multi-level (or even continuous) environment variables for $n_a$ species. In fact, the identification strategy and distribution-matching estimation carry over to larger joint label-environment spaces, though at the cost of heavier notation and more complex optimization. Technically, at this general multi-label and multi-background situation, the model identification considerations (see discussion in Section 3.3) becomes more complex. At this situation, one can identify both $\mathrm{pr}(\mathbf{X}, A = a | R = 0)$ as well as $\mathrm{pr}(A = a | R = 0)$, which in total $2n_a - 1$ quantities, while one has in total $n_y n_a$ unknown quantities, including $\mathrm{pr}(Y = y, A = a | R = 0)$ and the unobservable subpopulation distribution $\mathrm{pr}(\mathbf{X} | Y = 1, A = 1)$. To make sure this model is identifiable, one needs to make $(n_y - 2)n_a + 1$ anchor set assumptions. For example, when $n_y = 3$ and $n_a = 2$, 3 anchor set assumptions are needed. Interestingly, as long as the label is binary $n_y = 2$, one anchor set assumption is sufficient if only one subpopulation is missing in the source. In the setting we consider in the paper, $n_y = n_a = 2$, so we only need to make one anchor set assumption.

## B  THE USE OF LARGE LANGUAGE MODELS (LLMS)

We declare that large language models (LLMs) were used in a limited capacity solely to aid in polishing the writing and improving the linguistic clarity of this manuscript. The core research ideation, methodological development, experimental execution, data analysis, and technical content generation were conducted entirely by the human authors. All LLM-assisted refinements were carefully reviewed, validated, and approved by the authors, who take full responsibility for the final content. No LLMs were involved in the conceptualization of research ideas or the generation of technical claims, and LLMs were not used in any capacity that would constitute substantive intellectual contribution.

