# OpenReview forum: "Unsupervised Domain Adaptation with an Unobservable Source Subpopulation"
_ICLR.cc/2026/Conference — ICLR 2026 Conference Withdrawn Submission_

### Official Review · Reviewer_1GtB · 2025-10-30

**Soundness:** 3
**Presentation:** 2
**Contribution:** 2
**Rating:** 6
**Confidence:** 3

**Summary:**

The paper introduces a new unsupervised domain adaptation (UDA) setting where one source subpopulation, defined by a combination of label and background variables, is completely missing. This structured missingness makes classical domain adaptation invalid. The authors derive target-domain prediction formulas based on a conditional invariance assumption and propose a KL-divergence-based distribution matching estimator to recover unobserved subgroup proportions. They provide theoretical guarantees and validate the method on synthetic and real datasets. Results show consistent improvements over naïve benchmarks that ignore the unobserved subpopulation

**Strengths:**

1. The paper defines a previously unstudied but realistic UDA scenario: structured subpopulation absence in the source. This bridges gaps between label shift, open-set label shift, and missing-data frameworks.

2. The KL-based distribution matching avoids high-dimensional density estimation by relying only on classification probabilities, which is computationally elegant. Algorithm 1 is practical and well structured.

3. The paper situates itself well within OOD generalization, spurious correlation, and open-set adaptation literature.

**Weaknesses:**

1. Only Waterbirds is shown in the main text; synthetic and other results are relegated to the appendix. The method’s robustness across diverse settings (e.g., CelebA or tabular data) remains unclear.

2. The experiments seem rely on a handcrafted partition (missing (Y=1, A=1)) rather than a naturally occurring absence.

3. The key conditional invariance assumption is strong: it requires identical conditional distributions across domains. The authors provide examples but no empirical test or sensitivity analysis. In practice, small violations could substantially bias the estimator.

**Questions:**

1. How robust is the method if the conditional invariance assumption is slightly violated?

2. Can this framework extend to multi-class or continuous background variables A?

3. Can this be framed as a causal identification problem under missing mechanisms?

---

> ### Author Response · Authors · 2025-11-21
>
> >W1: Only Waterbirds is shown in the main text; synthetic and other results are relegated to the appendix. The method’s robustness across diverse settings (e.g., CelebA or tabular data) remains unclear.
>
> **Response**
> - Following your suggestion, we now compare the proposed method to a standard UDA method based solely on the label shift assumption without considering the missing subgroup. We simply denote it as "New". We evaluate all three methods (proposed, naive benchmark, and "New") on three representative datasets: Waterbirds, CelebA, and MIMIC. Results show that while "New" outperforms the naive benchmark $\xi(x)$, our proposal consistently achieves the best performance, validating its effectiveness.
>
> ### Table: Waterbirds (a = c = 0.5, Mean(SD))
> | Metric | Method | b=0.1 | 0.2 | 0.3 | 0.4 | 0.5 | 0.6 | 0.7 | 0.8 | 0.9 |
> |-------|-------|-------|-------|-------|-------|-------|-------|-------|-------|-------|
> | **Accuracy** | $\eta(x)$(proposal) | 0.841(0.003) | 0.845(0.003) | 0.848(0.003) | 0.850(0.003) | 0.851(0.003) | 0.853(0.003) | 0.856(0.003) | 0.859(0.003) | 0.863(0.003) |
> |  | $\xi(x)$(naive) | 0.741(0.013) | 0.769(0.011) | 0.786(0.009) | 0.796(0.007) | 0.803(0.007) | 0.811(0.008) | 0.816(0.007) | 0.820(0.006) | 0.824(0.005) |
> |  | New | 0.799(0.015) | 0.806(0.012) | 0.812(0.008) | 0.815(0.007) | 0.819(0.008) | 0.823(0.007) | 0.826(0.008) | 0.828(0.007) | 0.831(0.005) |
> | **F1** | $\eta(x)$(proposal) | 0.790(0.005) | 0.796(0.003) | 0.798(0.004) | 0.799(0.004) | 0.797(0.004) | 0.796(0.003) | 0.797(0.004) | 0.797(0.004) | 0.798(0.004) |
> |  | $\xi(x)$(naive) | 0.450(0.042) | 0.532(0.032) | 0.574(0.025) | 0.594(0.019) | 0.605(0.018) | 0.617(0.020) | 0.621(0.019) | 0.624(0.016) | 0.625(0.013) |
> |  | New | 0.640(0.038) | 0.655(0.030) | 0.665(0.022) | 0.669(0.017) | 0.671(0.018) | 0.674(0.016) | 0.672(0.018) | 0.668(0.017) | 0.665(0.013) |
>
> ### Table: Waterbirds (a = b = 0.5, Mean(SD))
> | Metric | Method | c=0.1 | 0.2 | 0.3 | 0.4 | 0.5 | 0.6 | 0.7 | 0.8 | 0.9 |
> |-------|-------|-------|-------|-------|-------|-------|-------|-------|-------|-------|
> | **Accuracy** | $\eta(x)$(proposal) | 0.869(0.002) | 0.870(0.003) | 0.866(0.001) | 0.860(0.002) | 0.851(0.003) | 0.840(0.004) | 0.826(0.005) | 0.809(0.005) | 0.788(0.006) |
> |  | $\xi(x)$(naive) | 0.801(0.006) | 0.816(0.006) | 0.818(0.008) | 0.813(0.008) | 0.803(0.007) | 0.787(0.007) | 0.765(0.006) | 0.737(0.007) | 0.703(0.008) |
> |  | New | 0.796(0.006) | 0.814(0.006) | 0.821(0.007) | 0.821(0.008) | 0.819(0.008) | 0.813(0.009) | 0.805(0.009) | 0.796(0.010) | 0.786(0.011) |
> | **F1** | $\eta(x)$(proposal) | 0.768(0.004) | 0.780(0.004) | 0.787(0.002) | 0.793(0.003) | 0.797(0.004) | 0.801(0.004) | 0.805(0.006) | 0.809(0.005) | 0.815(0.005) |
> |  | $\xi(x)$(naive) | 0.519(0.015) | 0.557(0.021) | 0.582(0.025) | 0.596(0.023) | 0.605(0.018) | 0.608(0.017) | 0.607(0.013) | 0.605(0.015) | 0.606(0.016) |
> |  | New | 0.531(0.015) | 0.581(0.020) | 0.620(0.020) | 0.647(0.021) | 0.671(0.018) | 0.689(0.019) | 0.710(0.017) | 0.730(0.017) | 0.753(0.017) |
>
> ### Table: CeleA (a = c = 0.5, Mean(SD))
> | Metric | Method | b=0.1 | 0.2 | 0.3 | 0.4 | 0.5 | 0.6 | 0.7 | 0.8 | 0.9 |
> |-------|-------|-------|-------|-------|-------|-------|-------|-------|-------|-------|
> | **Accuracy** | $\eta(x)$(proposal) | 0.655(0.004) | 0.662(0.005) | 0.667(0.004) | 0.671(0.004) | 0.675(0.004) | 0.681(0.004) | 0.690(0.003) | 0.698(0.003) | 0.708(0.003) |
> |  | $\xi(x)$(naive) | 0.395(0.002) | 0.412(0.003) | 0.419(0.003) | 0.420(0.003) | 0.422(0.003) | 0.421(0.003) | 0.419(0.003) | 0.416(0.002) | 0.412(0.002) |
> |  | New | 0.449(0.003) | 0.455(0.004) | 0.456(0.005) | 0.455(0.005) | 0.454(0.004) | 0.452(0.003) | 0.448(0.004) | 0.444(0.004) | 0.439(0.003) |
> | **F1** | $\eta(x)$(proposal) | 0.756(0.003) | 0.762(0.003) | 0.766(0.003) | 0.768(0.002) | 0.771(0.002) | 0.774(0.002) | 0.779(0.002) | 0.783(0.004) | 0.788(0.002) |
> |  | $\xi(x)$(naive) | 0.170(0.006) | 0.212(0.009) | 0.224(0.009) | 0.224(0.007) | 0.224(0.008) | 0.215(0.007) | 0.203(0.008) | 0.189(0.006) | 0.172(0.005) |
> |  | New | 0.324(0.010) | 0.335(0.012) | 0.337(0.013) | 0.331(0.013) | 0.326(0.010) | 0.316(0.006) | 0.302(0.010) | 0.287(0.009) | 0.270(0.005) |

---

> ### Author Response · Authors · 2025-11-21
>
> ### Table: CeleA (a = b = 0.5, Mean(SD))
> | Metric | Method | c=0.1 | 0.2 | 0.3 | 0.4 | 0.5 | 0.6 | 0.7 | 0.8 | 0.9 |
> |-------|-------|-------|-------|-------|-------|-------|-------|-------|-------|-------|
> | **Accuracy** | $\eta(x)$(proposal) | 0.659(0.003) | 0.657(0.002) | 0.659(0.003) | 0.664(0.003) | 0.675(0.004) | 0.697(0.004) | 0.730(0.003) | 0.776(0.001) | 0.835(0.001) |
> |  | $\xi(x)$(naive) | 0.533(0.003) | 0.509(0.003) | 0.483(0.003) | 0.454(0.003) | 0.422(0.003) | 0.386(0.002) | 0.344(0.002) | 0.297(0.002) | 0.242(0.002) |
> |  | New | 0.540(0.003) | 0.520(0.003) | 0.500(0.004) | 0.478(0.003) | 0.454(0.004) | 0.429(0.004) | 0.401(0.004) | 0.371(0.004) | 0.344(0.005) |
> | **F1** | $\eta(x)$(proposal) | 0.669(0.003) | 0.691(0.002) | 0.716(0.002) | 0.743(0.002) | 0.771(0.002) | 0.802(0.002) | 0.835(0.002) | 0.870(0.001) | 0.909(0.001) |
> |  | $\xi(x)$(naive) | 0.275(0.010) | 0.257(0.009) | 0.244(0.009) | 0.232(0.007) | 0.224(0.008) | 0.218(0.005) | 0.209(0.007) | 0.203(0.005) | 0.197(0.004) |
> |  | New | 0.310(0.011) | 0.310(0.009) | 0.313(0.012) | 0.318(0.009) | 0.326(0.010) | 0.338(0.009) | 0.350(0.009) | 0.366(0.008) | 0.393(0.009) |
>
> ### Table: MIMIC-III (a = c = 0.5, Mean(SD))
> | Metric | Method | b=0.1 | 0.2 | 0.3 | 0.4 | 0.5 | 0.6 | 0.7 | 0.8 | 0.9 |
> |--|--|--|--|--|--|--|--|--|-------|-------|
> | **Accuracy** | $\eta(x)$(proposal) | 0.904(0.001) | 0.902(0.001) | 0.899(0.001) | 0.895(0.001) | 0.892(0.001) | 0.889(0.001) | 0.885(0.001) | 0.881(0.002) | 0.877(0.001) |
> |  | $\xi(x)$(naive) | 0.529(0.006) | 0.679(0.005) | 0.761(0.006) | 0.810(0.006) | 0.837(0.004) | 0.855(0.003) | 0.865(0.004) | 0.871(0.002) | 0.873(0.002) |
> |  | New | 0.900(0.002) | 0.900(0.001) | 0.897(0.001) | 0.895(0.001) | 0.891(0.001) | 0.888(0.001) | 0.884(0.001) | 0.880(0.001) | 0.875(0.001) |
> | **F1** | $\eta(x)$(proposal) | 0.949(0.000) | 0.947(0.001) | 0.945(0.001) | 0.944(0.001) | 0.942(0.001) | 0.939(0.001) | 0.937(0.001) | 0.935(0.001) | 0.932(0.001) |
> |  | $\xi(x)$(naive) | 0.648(0.006) | 0.786(0.004) | 0.850(0.005) | 0.885(0.004) | 0.904(0.003) | 0.916(0.002) | 0.923(0.002) | 0.926(0.002) | 0.927(0.001) |
> |  | New | 0.946(0.001) | 0.946(0.001) | 0.945(0.000) | 0.943(0.001) | 0.941(0.001) | 0.939(0.001) | 0.936(0.001) | 0.934(0.001) | 0.931(0.001) |
>
> ### Table: MIMIC-III (a = b = 0.5,Mean(SD))
> | Metric | Method | c=0.1 | 0.2 | 0.3 | 0.4 | 0.5 | 0.6 | 0.7 | 0.8 | 0.9 |
> |--|---|--|--|--|--|--|--|--|--|--|
> |**Accuracy**| $\eta(x)$(proposal) | 0.867(0.001) | 0.873(0.001) | 0.879(0.001) | 0.886(0.001) | 0.892(0.001) | 0.900(0.001) | 0.907(0.001) | 0.915(0.001) | 0.924(0.001) |
> || $\xi(x)$(naive) | 0.855(0.003) | 0.854(0.004) | 0.849(0.004) | 0.844(0.004) | 0.837(0.004) | 0.830(0.004) | 0.822(0.004) | 0.814(0.005) | 0.804(0.004) |
> || New |0.864(0.001)|0.870(0.001)|0.877(0.001)|0.884(0.001)|0.891(0.001)|0.899(0.001)|0.906(0.001)|0.914(0.001)| 0.922(0.001)|
> |**F1**|$\eta(x)$(proposal)|0.926(0.001)|0.930(0.001)|0.933(0.001)|0.938(0.001)|0.942(0.001)|0.946(0.001)| 0.952(0.001)|0.955(0.001)|0.960(0.000)|
> || $\xi(x)$(naive) | 0.916(0.002) | 0.914(0.003) |0.911(0.003)|0.908(0.003)|0.904(0.003)|0.900(0.003)|0.895(0.003)| 0.890(0.003)|0.885(0.003)|
> ||New|0.924(0.001)|0.928(0.001)|0.932(0.001)|0.937(0.001) | 0.941(0.001) |0.945(0.001)|0.950(0.001)|0.954(0.001)| 0.959(0.001)|
>
> >W2: The experiments seem rely on a handcrafted partition (missing (Y=1, A=1)) rather than a naturally occurring absence.
>
> **Response**
> - To comprehensively demonstrate the performance of the proposed method with different proportions of each subpopulation, the setting used in the experiments is indeed not a naturally occuring absence.
>
> >W3: The key conditional invariance assumption is strong: it requires identical conditional distributions across domains. The authors provide examples but no empirical test or sensitivity analysis. In practice, small violations could substantially bias the estimator.
>
> **Response**
> - First of all, in the context of UDA, without making any assumption, it is difficult or even infeasible to understand the relation between the source domain and the target domain. Commonly seen assumptions include covariate shift, label shift, or their variants.
> - The assumption we make is considerably weaker than standard label shift assumption. In many applications, features are causally generated from $(Y,A)$, while domain differences primarily arise from sampling, making this assumption realistic. We further justify this point in lines 122–136 and provide examples illustrating its practical relevance.
> - In theory, without making such an assumption, the target posterior $p(Y∣X,A,R=0)$ would be unidentifiable. Thus, this assumption is theoretically necessary.
> - That being said, we agree that this assumption might be susceptible to real-world perturbations. In our response to Q1 below, we will provide a detailed analysis of this issue, including a sensitivity analysis to assess the robustness of our estimator to potential violations of this assumption.

---

> ### Author Response · Authors · 2025-11-21
>
> > Q1: How robust is the method if the conditional invariance assumption is slightly violated?
>
> **Response**
>
> - To evaluate the robustness, we introduce a perturbation in the target domain by shifting $\mu_{YA}$ to $\mu_{YA} + 0.2e$, where $e$ is a four-dimensional standard normal vector, keeping all other settings unchanged. The results remain consistent with the main text, showing that our method is robust to mild violations of equation (3).
>
> ### Accuracy Results(Mean(SD))
>
> | $n_0$      | Model       | $n_1=1000$     | $2000$        | $3000$        | $4000$        | $5000$        | $6000$        | $7000$        | $8000$        |
> |------------|--------------|----------------|----------------|----------------|----------------|----------------|----------------|----------------|----------------|
> | **1000**   | $\eta_1(x)$  | 0.751 (0.052) | 0.762 (0.050) | 0.756 (0.048) | 0.757 (0.045) | 0.755 (0.046) | 0.756 (0.048) | 0.761 (0.047) | 0.757 (0.045) |
> |            | $\xi_1(x)$   | 0.504 (0.023) | 0.502 (0.023) | 0.498 (0.022) | 0.500 (0.020) | 0.501 (0.022) | 0.502 (0.022) | 0.501 (0.023) | 0.501 (0.023) |
> |            | $\eta(x)$    | 0.694 (0.041) | 0.697 (0.024) | 0.699 (0.052) | 0.695 (0.039) | 0.698 (0.037) | 0.698 (0.036) | 0.701 (0.035) | 0.693 (0.034) |
> |            | $\xi(x)$     | 0.629 (0.033) | 0.631 (0.030) | 0.630 (0.030) | 0.627 (0.032) | 0.631 (0.035) | 0.630 (0.031) | 0.634 (0.033) | 0.624 (0.030) |
> | **6000**   | $\eta_1(x)$  | 0.761 (0.043) | 0.766 (0.044) | 0.762 (0.044) | 0.758 (0.041) | 0.756 (0.047) | 0.749 (0.045) | 0.761 (0.043) | 0.748 (0.049) |
> |            | $\xi_1(x)$   | 0.502 (0.010) | 0.501 (0.009) | 0.501 (0.009) | 0.501 (0.009) | 0.501 (0.009) | 0.502 (0.009) | 0.501 (0.009) | 0.501 (0.009) |
> |            | $\eta(x)$    | 0.699 (0.032) | 0.702 (0.032) | 0.702 (0.031) | 0.698 (0.034) | 0.700 (0.033) | 0.694 (0.034) | 0.703 (0.031) | 0.696 (0.036) |
> |            | $\xi(x)$     | 0.633 (0.026) | 0.631 (0.029) | 0.632 (0.028) | 0.632 (0.028) | 0.633 (0.027) | 0.628 (0.029) | 0.631 (0.026) | 0.630 (0.030) |
> ### F1 Score Results (Mean(SD))
> | $n_0$      | Model     | $n_1=1000$     | $2000$        | $3000$        | $4000$        | $5000$        | $6000$        | $7000$        | $8000$        |
> |------------|--------------|----------------|----------------|----------------|----------------|----------------|----------------|----------------|----------------|
> | **1000**   | $\eta_1(x)$  | 0.748 (0.059) | 0.761 (0.055) | 0.755 (0.052) | 0.753 (0.048) | 0.753 (0.052) | 0.753 (0.055) | 0.761 (0.049) | 0.756 (0.048) |
> |  | $\xi_1(x)$   | 0.670 (0.020) | 0.668 (0.020) | 0.665 (0.019) | 0.666 (0.018) | 0.667 (0.019) | 0.668 (0.020) | 0.667 (0.020) | 0.667 (0.020) |
> || $\eta(x)$    | 0.703 (0.024) | 0.704 (0.023) | 0.702 (0.024) | 0.701 (0.025) | 0.704 (0.027) | 0.703 (0.025) | 0.707 (0.026) | 0.699 (0.023) |
> |  | $\xi(x)$     | 0.691 (0.049) | 0.693 (0.051) | 0.695 (0.043) | 0.693 (0.054) | 0.693 (0.050) | 0.696 (0.047) | 0.703 (0.043) | 0.688 (0.050) |
> | **6000**   | $\eta_1(x)$  | 0.761 (0.044) | 0.765 (0.047) | 0.762 (0.047) | 0.758 (0.042) | 0.758 (0.048) | 0.750 (0.045) | 0.762 (0.048) | 0.750 (0.050) |
> | | $\xi_1(x)$   | 0.668 (0.008) | 0.668 (0.008) | 0.667 (0.008) | 0.668 (0.008) | 0.668 (0.008) | 0.668 (0.008) | 0.667 (0.008) | 0.667 (0.008) |
> |   | $\eta(x)$    | 0.706 (0.020) | 0.705 (0.022) | 0.704 (0.020) | 0.705 (0.020) | 0.706 (0.019) | 0.701 (0.021) | 0.704 (0.020) | 0.703 (0.022) |
> |   | $\xi(x)$     | 0.700 (0.037) | 0.702 (0.041) | 0.702 (0.035) | 0.696 (0.042) | 0.700 (0.038) | 0.690 (0.043) | 0.701 (0.047) | 0.693 (0.047) |
> ### Accuracy Results(Mean(SD))
>
> | $n_1$    | model       | $n_0=1000$     | $2000$  | $3000$  | $4000$   | $5000$  | $6000$ | $7000$   | $8000$|
> |--|--|--|--|--|--|--|--|--|--|
> | ***1000*** | $\eta_1(x)$ | 0.751 (0.052) | 0.748 (0.049) | 0.754 (0.045) | 0.756 (0.042) | 0.762 (0.044) | 0.761 (0.043) | 0.753 (0.043) | 0.757 (0.043) |
> | | $\xi_1(x)$  | 0.504 (0.023) | 0.501 (0.009) | 0.502 (0.014) | 0.502 (0.012) | 0.501 (0.010) | 0.502 (0.010) | 0.502 (0.009) | 0.501 (0.008) |
> || $\eta(x)$ | 0.694 (0.041) | 0.696 (0.036) | 0.690 (0.033) | 0.693 (0.030) | 0.702 (0.030) | 0.699 (0.032) | 0.695 (0.035) | 0.694 (0.029) |
> | | $\xi(x)$| 0.629 (0.033) | 0.630 (0.030) | 0.628 (0.026) | 0.626 (0.027) | 0.631 (0.028) | 0.633 (0.026) | 0.634 (0.027) | 0.628 (0.028) |
> | ***6000***| $\eta_1(x)$ | 0.756 (0.048) | 0.757 (0.042) | 0.750 (0.041) | 0.761 (0.044) | 0.751 (0.044) | 0.749 (0.045) | 0.748 (0.046) | 0.759 (0.047) |
> | | $\xi_1(x)$  | 0.502 (0.022) | 0.502 (0.016) | 0.502 (0.013) | 0.502 (0.011) | 0.502 (0.010) | 0.502 (0.009) | 0.501 (0.008) | 0.501 (0.008) |
> || $\eta(x)$| 0.698 (0.036) | 0.705 (0.033) | 0.694 (0.031) | 0.702 (0.034) | 0.697 (0.035) | 0.694 (0.034) | 0.692 (0.036) | 0.695 (0.032) |
> || $\xi(x)$ | 0.630 (0.031) | 0.637 (0.029) | 0.633 (0.026) | 0.635 (0.030) | 0.631 (0.026) | 0.628 (0.029) | 0.626 (0.030) | 0.627 (0.031) |

---

> > ### Author Response · Authors · 2025-11-21
> >
> > ### F1 Score Results(Mean(SD))
> >
> > | $n_1$    | model       | $n_0=1000$     | 1000        | 2000        | 3000        | 4000        | 5000        | 6000        | 7000        | 8000        |
> > |----------------|----------------|----------------|-------------|-------------|-------------|-------------|-------------|-------------|-------------|-------------|
> > | ***1000***     | $\eta_1(x)$    | 0.748 (0.059) | 0.750 (0.050) | 0.754 (0.048) | 0.757 (0.045) | 0.764 (0.047) | 0.761 (0.044) | 0.753 (0.046) | 0.758 (0.045) |
> > || $\xi_1(x)$     | 0.670 (0.020) | 0.667 (0.008) | 0.668 (0.013) | 0.668 (0.010) | 0.668 (0.009) | 0.668 (0.008) | 0.668 (0.008) | 0.668 (0.007) |
> > || $\eta(x)$      | 0.703 (0.024) | 0.703 (0.022) | 0.701 (0.019) | 0.700 (0.019) | 0.704 (0.020) | 0.706 (0.020) | 0.706 (0.021) | 0.702 (0.020) |
> > || $\xi(x)$       | 0.691 (0.049) | 0.693 (0.047) | 0.682 (0.044) | 0.690 (0.040) | 0.697 (0.041) | 0.700 (0.037) | 0.691 (0.049) | 0.693 (0.038) |
> > |***6000***| $\eta_1(x)$    | 0.753 (0.055) | 0.757 (0.043) | 0.750 (0.041) | 0.760 (0.045) | 0.752 (0.044) | 0.750 (0.045) | 0.748 (0.050) | 0.758 (0.049) |
> > || $\xi_1(x)$     | 0.668 (0.020) | 0.669 (0.014) | 0.669 (0.011) | 0.668 (0.010) | 0.668 (0.008) | 0.668 (0.008) | 0.668 (0.007) | 0.668 (0.007) |
> > || $\eta(x)$       | 0.703 (0.025) | 0.709 (0.022) | 0.705 (0.019) | 0.707 (0.020) | 0.705 (0.020) | 0.701 (0.021) | 0.701 (0.022) | 0.702 (0.022) |
> > || $\xi(x)$      | 0.696 (0.047) | 0.703 (0.046) | 0.691 (0.042) | 0.699 (0.038) | 0.695 (0.043) | 0.690 (0.043) | 0.691 (0.047) | 0.697 (0.036) |
> >
> >
> > >Q2: Can this framework extend to multi-class or continuous background variables A?
> >
> >
> > - Yes. We have already included a discussion of this generalization in Appendix A.5. Our setting can be extended to the situation where label $Y$ has $n_y$ categories and background $A$ has $n_a$ categories, $n_y>2, n_a>2$. The difficulty is that the model identification (presented in Section 3.3) becomes more complex. To be brief, at this general situation, we can identify both $p(x, A=a|R=0)$ as well as $pr(A=a|R=0)$, which in total $2n_a-1$ quantities, while we have in total $n_y*n_a$ unknown quantities ($pr(Y=y,A=a|R=0)$ and the unobservable subpopulation distribution $p(x|Y=1,A=1)$). To make sure this model is identifiable, we need to make $(n_y-2)n_a+1$ anchor set assumptions. In the setting we consider in the paper, $n_y=n_a=2$, so we only need to make 1 anchor set assumption (see lines 214-215). Interestingly, as long as the label is binary $n_y=2$, one anchor set assumption is sufficient if only one subpopulation is missing in the source. In the general multi-label and multi-background setting, one needs to make more anchor set assumptions to make the model identifiable.
> > - Extension to a continuous background is also of interest, but we feel in practice it might be challenging to define an appropriate subpopulation that is missing in the source domain. Nevertheless, this situation has been analyzed in the statistical literature for missing not at random data, see, e.g., Miao and Tchetgen Tchetgen (2016).
> >
> > **References**
> > - Miao，W and Tchetgen Tchetgen, E.J. (2016). On varieties of doubly robust estimators under missingness not at random with a shadow variable. Biometrika, 103(2), 475–482.
> >
> > >Q3:Can this be framed as a causal identification problem under missing mechanisms?
> >
> > - This is a great question but we do not know the answer for now. To answer this question, some more background or information might be needed, such as what variable serves as the treatment and which variable has missing values. Based on the setting considered in our paper, we believe this question is beyond the scope and capacity of our current work, and unfortunately, we do not have an answer for this rebuttal.

---

> > > ### Comment · Reviewer_1GtB · 2025-11-21
> > >
> > > Thanks for the detailed responses and the extra experiments. Most of my concerns are resolved. My only remaining reservation is about the assumptions, especially the conditional invariance part. The new experiments with the additive Gaussian perturbation are informative but still not fully sufficient to justify that this holds in complex real-world settings, given the simple form of the perturbation. I will keep my positive rating for now and remain open to further improvements in later discussions with other reviewers.

---

> > > > ### Author Response · Authors · 2025-12-02
> > > >
> > > > **Response**
> > > >
> > > > - We understand the reviewer's concerns regarding the assumption of conditional invariance. This assumption shares the similar spirit to the classical label-shift or covariate-shift assumption, and is intended to model the stable structural mechanism $p(X∣Y,A)$ while allowing substantial changes in $p(Y∣A)$ and $p(X∣A)$ (see lines 123–136). Such stability is often reasonable in practice—for instance, in medical diagnostics where feature measurements are largely determined by disease status and demographic variables, even though disease prevalence and population composition may vary across datasets.
> > > >
> > > > - The additive Gaussian perturbation experiments were designed as an initial stress test to illustrate robustness under moderate deviations from ideal invariance. We agree that richer perturbation models and more systematic empirical diagnostics would further strengthen the justification.

---

### Official Review · Reviewer_3cwJ · 2025-11-01

**Soundness:** 2
**Presentation:** 3
**Contribution:** 2
**Rating:** 2
**Confidence:** 3

**Summary:**

This paper proposes a new setting for domain adaptation, in which they assume the data is rare to obtain even in the source domain, and thus some cases exist in the target domain but not exist in the source domain.

**Strengths:**

This paper is easy to read. Table 1 and Figure 1 are helpful in understanding the motivation.

**Weaknesses:**

It is hard to understand the application of this setting, so I can't support the idea in this paper. In this paper, they first define Y ∈ {0, 1} as the binary label observed in the source domain, and then define A ∈ {0, 1} as a binary background or environment variable. But why should Y and A be binary? In the reviewer's sense, in every application, many categories exist in the distribution of Y, as well as the shared information A across both domains. It can be observed in Figure 2 that Y is defined as "Blond hair" and "Black Hair" in this setting, but hairs with other colors should also be considered. Similar conflicts with different variables, like Water/Land/Sky and Water/Land/Sky bird.

With the initial setting being hard to assess, it is hard to consider the contribution of this paper further.

**Questions:**

Please clarify the weakness above.

---

> ### Author Response · Authors · 2025-11-21
>
> >W1: It is hard to understand the application of this setting, so I can't support the idea in this paper. In this paper, they first define $Y\in{0, 1}$ as the binary label observed in the source domain, and then define $A\in{0, 1}$ as a binary background or environment variable. But why should Y and A be binary? In the reviewer's sense, in every application, many categories exist in the distribution of Y, as well as the shared information A across both domains. It can be observed in Figure 2 that Y is defined as "Blond hair" and "Black Hair" in this setting, but hairs with other colors should also be considered. Similar conflicts with different variables, like Water/Land/Sky and Water/Land/Sky bird. With the initial setting being hard to assess, it is hard to consider the contribution of this paper further.
>
> **Response**
>
> - **Why start with the binary case:** The binary setting is a principled first step: it already yields four subpopulations and captures the main challenge of an entirely missing structured group (e.g., Waterbirds with $Y=1,A=1$). This is a canonical scenario in practice (e.g., healthcare subgroups or spurious correlation benchmarks), and analyzing it carefully provides valuable insights before scaling up.
>
>
> - **Extension beyond binary labels/environments:** Our theoretical framework (see Sections 3.2–3.4 and Appendix A.1) is built upon structured conditional invariance and mixture proportion estimation. These tools naturally generalize to multi-class labels and multi-level (or even continuous) environment variables. In fact, the identification strategy and distribution-matching estimation carry over to larger joint label–environment spaces, though at the cost of heavier notation and more complex optimization. We have emphasized this in our discussions and conclusions (Appendix A.5, line 1481–1492: "These tools naturally generalize to multi-class labels for $n_y$ species and multi-level (or even continuous) environment variables for $n_a$ species..."). Technically, at this general multi-label and multi-background situation, the model identification considerations (see discussion in Section 3.3) becomes more complex. At this situation, one can identify both $p(x,A=a|R=0)$ as well as $p(A=a|R=0)$, which in total $2n_a-1$ quantities, while one has in total $n_yn_a$ unknown quantities, including $p(Y=y,A=a|R=0)$ and the unobservable subpopulation distribution $p(x|Y=1,A=1)$. To make sure this model is identifiable, one needs to make $(n_y-2)n_a+1$ anchor set assumptions. For example, when $n_y=3$ and $n_a=2$, 3 anchor set assumptions are needed. Interestingly, as long as the label is binary $n_y=2$, one anchor set assumption is sufficient if only one subpopulation is missing in the source. In the setting we consider in the paper, $n_y=n_a=2$, so we only need to make one anchor set assumption (see lines 133-134).
>
>
>
> - **Scalability in practice:** Although we focus on the case where one subgroup is missing, our results and proofs (e.g., Lemma 3.2, Proposition 3.1) extend to cases where multiple subpopulations are underrepresented or absent. The methodology hinges on estimating missing proportions via mixture decomposition, which remains valid in higher-dimensional joint label–environment settings. Furthermore, distribution-matching approaches such as KL-based objectives or moment-matching (Appendix A.1) are flexible enough to handle larger state spaces.
>
>
>
> - We have already included a discussion of this generalization in Appendix A.5. Here, we would like to reiterate and clarify the key point: although our main exposition focuses on binary labels and binary environment variables, this choice was intentional—it represents the simplest non-trivial setting that highlights the core difficulty arising from structured subpopulation absence. Importantly, our methodology is not inherently limited to the binary case and can be extended to scenarios where label $Y$ has $n_y$ categories and background $A$ has $n_a$ categories, $n_y>2$, $n_a>2$.

---

> > ### Comment · Reviewer_3cwJ · 2025-11-28
> >
> > I appreciate the authors' rebuttal, but my initial concern persists. Both the theoretical analysis and the experimental evaluation rest entirely on the assumption that only binary labels {0, 1} exist. This assumption severely narrows the method's applicability; without empirical evidence on multi-label datasets, its broader effectiveness remains unverified. Consequently, I respectfully retain my original rating.

---

> > > ### Author Response · Authors · 2025-12-02
> > >
> > > We thank the reviewer for their comments. We would like to clarify that, in classification problems, extending methods from the binary-label setting to the multi-label setting can be straightforward, or it is substantially more complex. Our work falls into the latter category, primarily due to the structured missingness present in the source population.
> > >
> > > As explained in our rebuttal above, we provided reasons why focusing on the binary-label case is appropriate for this paper and outlined the theoretical challenges that arise when extending the framework beyond the binary label/environment setting. Given the scope and capacity of a single paper, we have chosen to concentrate on the binary case here, and we plan to investigate more complex extensions in future work.

---

### Official Review · Reviewer_xz9M · 2025-11-01

**Soundness:** 3
**Presentation:** 3
**Contribution:** 2
**Rating:** 4
**Confidence:** 3

**Summary:**

The paper studies a new and fine-grained unsupervised domain adaptation problem where a structured label–background subpopulation is missing in the source domain. Based on the assumption of identical conditional distributions across the two domains, the authors propose a theoretical framework to estimate subpopulation proportions in the target domain, with theoretical guarantees and error bounds provided. Experiments on both synthetic and real-world datasets demonstrate that the proposed method is effective and outperforms baselines that ignore structured missingness.

**Strengths:**

The subpopulation shift problem in unsupervised domain adaptation is common in realistic data collection scenarios, making it both significant and practical to study. This paper considers an extreme case where an entire label–background subpopulation is missing from the source domain.

The proposed subpopulation estimation method is solid, supported by rigorous theoretical guarantees and error bounds. It provides a comprehensive solution to the specific subpopulation-missing problem.

The experimental design aligns well with the problem setting, and empirical results on both synthetic and real-world datasets demonstrate the effectiveness of the proposed method.

**Weaknesses:**

The problem setting lacks sufficient novelty. Subpopulation shift has been studied previously [1,2,3,4,5], and it is not convincing to claim novelty merely by considering an extreme case of the open-set subpopulation problem—where one subpopulation is entirely missing from the source domain but may exist in the target domain—and presenting it as a new problem setting.

The proposed method relies on the assumption that the conditional distribution is identical across the two domains, which can be easily satisfied under simplified conditions. However, this assumption oversimplifies real-world domain adaptation scenarios, where both severe domain shifts and label shifts often exist, making it difficult to ensure an invariant and target-discriminative conditional distribution.

To align with this restricted problem setting, the experiments are also limited in practicality and scope. Although the paper discusses historical and facial datasets to motivate the problem, only one real-world dataset (Waterbirds) is actually used. Moreover, the selected baseline adaptation methods are not representative, overlooking both prior subpopulation shift approaches and several mainstream adaptation methods such as distribution matching and self-training.

References:

[1] A Theory of Label Propagation for Subpopulation Shift. ICML@2021

[2] Adaptive Risk Minimization: Learning to Adapt to Domain Shift. Neurips@2021

[3] Domain Adaptation meets Individual Fairness. And they get along. Neurips@2022

[4] RLSbench: Domain Adaptation Under Relaxed Label Shift. ICML@2023

[5] Change is Hard: A Closer Look at Subpopulation Shift. ICML@2023

**Questions:**

The paper provides a solid solution to the label–background subpopulation missingness problem in unsupervised domain adaptation. However, the main concern lies in the practicality of the setting and experiments, which limits the overall scope and applicability of the work.

---

> ### Author Response · Authors · 2025-11-21
>
> >W1: The problem setting lacks sufficient novelty. Subpopulation shift has been studied previously [1,2,3,4,5], and it is not convincing to claim novelty merely by considering an extreme case of the open-set subpopulation problem—where one subpopulation is entirely missing from the source domain but may exist in the target domain—and presenting it as a new problem setting.
>
> **Response**
> - We respectfully disagree that our problem setting lacks sufficient novelty.
> - Although our setting is related to Open Set Label Shift (OSLS), it differs fundamentally in both structure and identifiability. In OSLS, the target domain contains a truly new class whose distribution is entirely unobserved in the source domain. This introduces an unrestricted latent component in the mixture model, making the problem unidentifiable without strong auxiliary assumptions.
> - In contrast, the “unobserved” group in our setting is not a new class, but a structured subpopulation, namely $(Y=1,A=1)$. Crucially, the background variable $A$ is always observed in both domains. This observable structure partitions the target domain and reduces the missingness to a binary conditional mixture within the subset $A=1$, which is equivalent to a PU-learning setting. Because the latent component is fully parameterized by known conditional distributions, the model becomes identifiable, and we can derive closed-form expressions for the target-domain predictor.
> - Therefore, our problem corresponds to a much more structured and identifiable special case that OSLS frameworks do not cover. Unlike OSLS, where the new class is fundamentally unknown, our setting leverages the observable variable $A$ to constrain the latent distribution, making identification feasible and enabling explicit solutions that general OSLS methods cannot obtain.
>
> >W2: The proposed method relies on the assumption that the conditional distribution is identical across the two domains, which can be easily satisfied under simplified conditions. However, this assumption oversimplifies real-world domain adaptation scenarios, where both severe domain shifts and label shifts often exist, making it difficult to ensure an invariant and target-discriminative conditional distribution.
>
>
> **Response**
>
> - First of all, in the context of UDA, without making any assumption, it is difficult or even infeasible to understand the relation between the source domain and the target domain. Commonly seen assumptions include covariate shift, label shift, or their variants.
> - The assumption we make is considerably weaker than standard label shift assumption. In many applications, features are causally generated from $(Y,A)$, while domain differences primarily arise from sampling, making this assumption realistic. We further justify this point in lines 122–136 and provide examples illustrating its practical relevance.
> - In theory, without making such an assumption, the target posterior $p(Y∣X,A,R=0)$ would be unidentifiable. Thus, this assumption is theoretically necessary.
>
> >W3: To align with this restricted problem setting, the experiments are also limited in practicality and scope. Although the paper discusses historical and facial datasets to motivate the problem, only one real-world dataset (Waterbirds) is actually used. Moreover, the selected baseline adaptation methods are not representative, overlooking both prior subpopulation shift approaches and several mainstream adaptation methods such as distribution matching and self-training.
>
> **Response**
> - Following your suggestion, we now compare the proposed method to a standard UDA method based solely on the label shift assumption without considering the missing subgroup. We simply denote it as "New". We evaluate all three methods (proposed, naive benchmark, and "New") on three representative datasets: Waterbirds, CelebA, and MIMIC. Results show that while "New" outperforms the naive benchmark $\xi(x)$, our proposal consistently achieves the best performance, validating its effectiveness.

---

> ### Author Response · Authors · 2025-11-21
>
> ### Table: Waterbirds (a = c = 0.5, Mean(SD))
> | Metric | Method | b=0.1 | 0.2 | 0.3 | 0.4 | 0.5 | 0.6 | 0.7 | 0.8 | 0.9 |
> |-------|-------|-------|-------|-------|-------|-------|-------|-------|-------|-------|
> | **Accuracy** | $\eta(x)$(proposal) | 0.841(0.003) | 0.845(0.003) | 0.848(0.003) | 0.850(0.003) | 0.851(0.003) | 0.853(0.003) | 0.856(0.003) | 0.859(0.003) | 0.863(0.003) |
> |  | $\xi(x)$(naive) | 0.741(0.013) | 0.769(0.011) | 0.786(0.009) | 0.796(0.007) | 0.803(0.007) | 0.811(0.008) | 0.816(0.007) | 0.820(0.006) | 0.824(0.005) |
> |  | New | 0.799(0.015) | 0.806(0.012) | 0.812(0.008) | 0.815(0.007) | 0.819(0.008) | 0.823(0.007) | 0.826(0.008) | 0.828(0.007) | 0.831(0.005) |
> | **F1** | $\eta(x)$(proposal) | 0.790(0.005) | 0.796(0.003) | 0.798(0.004) | 0.799(0.004) | 0.797(0.004) | 0.796(0.003) | 0.797(0.004) | 0.797(0.004) | 0.798(0.004) |
> |  | $\xi(x)$(naive) | 0.450(0.042) | 0.532(0.032) | 0.574(0.025) | 0.594(0.019) | 0.605(0.018) | 0.617(0.020) | 0.621(0.019) | 0.624(0.016) | 0.625(0.013) |
> |  | New | 0.640(0.038) | 0.655(0.030) | 0.665(0.022) | 0.669(0.017) | 0.671(0.018) | 0.674(0.016) | 0.672(0.018) | 0.668(0.017) | 0.665(0.013) |
>
> ### Table: Waterbirds (a = b = 0.5, Mean(SD))
> | Metric | Method | c=0.1 | 0.2 | 0.3 | 0.4 | 0.5 | 0.6 | 0.7 | 0.8 | 0.9 |
> |-------|-------|-------|-------|-------|-------|-------|-------|-------|-------|-------|
> | **Accuracy** | $\eta(x)$(proposal) | 0.869(0.002) | 0.870(0.003) | 0.866(0.001) | 0.860(0.002) | 0.851(0.003) | 0.840(0.004) | 0.826(0.005) | 0.809(0.005) | 0.788(0.006) |
> |  | $\xi(x)$(naive) | 0.801(0.006) | 0.816(0.006) | 0.818(0.008) | 0.813(0.008) | 0.803(0.007) | 0.787(0.007) | 0.765(0.006) | 0.737(0.007) | 0.703(0.008) |
> |  | New | 0.796(0.006) | 0.814(0.006) | 0.821(0.007) | 0.821(0.008) | 0.819(0.008) | 0.813(0.009) | 0.805(0.009) | 0.796(0.010) | 0.786(0.011) |
> | **F1** | $\eta(x)$(proposal) | 0.768(0.004) | 0.780(0.004) | 0.787(0.002) | 0.793(0.003) | 0.797(0.004) | 0.801(0.004) | 0.805(0.006) | 0.809(0.005) | 0.815(0.005) |
> |  | $\xi(x)$(naive) | 0.519(0.015) | 0.557(0.021) | 0.582(0.025) | 0.596(0.023) | 0.605(0.018) | 0.608(0.017) | 0.607(0.013) | 0.605(0.015) | 0.606(0.016) |
> |  | New | 0.531(0.015) | 0.581(0.020) | 0.620(0.020) | 0.647(0.021) | 0.671(0.018) | 0.689(0.019) | 0.710(0.017) | 0.730(0.017) | 0.753(0.017) |
>
> ### Table: CeleA (a = c = 0.5, Mean(SD))
> | Metric | Method | b=0.1 | 0.2 | 0.3 | 0.4 | 0.5 | 0.6 | 0.7 | 0.8 | 0.9 |
> |-------|-------|-------|-------|-------|-------|-------|-------|-------|-------|-------|
> | **Accuracy** | $\eta(x)$(proposal) | 0.655(0.004) | 0.662(0.005) | 0.667(0.004) | 0.671(0.004) | 0.675(0.004) | 0.681(0.004) | 0.690(0.003) | 0.698(0.003) | 0.708(0.003) |
> |  | $\xi(x)$(naive) | 0.395(0.002) | 0.412(0.003) | 0.419(0.003) | 0.420(0.003) | 0.422(0.003) | 0.421(0.003) | 0.419(0.003) | 0.416(0.002) | 0.412(0.002) |
> |  | New | 0.449(0.003) | 0.455(0.004) | 0.456(0.005) | 0.455(0.005) | 0.454(0.004) | 0.452(0.003) | 0.448(0.004) | 0.444(0.004) | 0.439(0.003) |
> | **F1** | $\eta(x)$(proposal) | 0.756(0.003)|0.762(0.003)|0.766(0.003) | 0.768(0.002) | 0.771(0.002) | 0.774(0.002) | 0.779(0.002) | 0.783(0.004) | 0.788(0.002) |
> |  | $\xi(x)$(naive) | 0.170(0.006) | 0.212(0.009) | 0.224(0.009) | 0.224(0.007) | 0.224(0.008) | 0.215(0.007) | 0.203(0.008) | 0.189(0.006) | 0.172(0.005) |
> |  | New | 0.324(0.010) | 0.335(0.012) | 0.337(0.013) | 0.331(0.013) | 0.326(0.010) | 0.316(0.006) | 0.302(0.010) | 0.287(0.009) | 0.270(0.005) |
>
> ### Table: CeleA (a = b = 0.5, Mean(SD))
> | Metric | Method | c=0.1 | 0.2 | 0.3 | 0.4 | 0.5 | 0.6 | 0.7 | 0.8 | 0.9 |
> |-------|-------|-------|-------|-------|-------|-------|-------|-------|-------|-------|
> | **Accuracy** | $\eta(x)$(proposal) | 0.659(0.003) | 0.657(0.002) | 0.659(0.003) | 0.664(0.003) | 0.675(0.004) | 0.697(0.004) | 0.730(0.003) | 0.776(0.001) | 0.835(0.001) |
> |  | $\xi(x)$(naive) | 0.533(0.003) | 0.509(0.003) | 0.483(0.003) | 0.454(0.003) | 0.422(0.003) | 0.386(0.002) | 0.344(0.002) | 0.297(0.002) | 0.242(0.002) |
> |  | New | 0.540(0.003) | 0.520(0.003) | 0.500(0.004) | 0.478(0.003) | 0.454(0.004) | 0.429(0.004) | 0.401(0.004) | 0.371(0.004) | 0.344(0.005) |
> | **F1** | $\eta(x)$(proposal) | 0.669(0.003) | 0.691(0.002) | 0.716(0.002) | 0.743(0.002) | 0.771(0.002) | 0.802(0.002) | 0.835(0.002) | 0.870(0.001) | 0.909(0.001) |
> |  | $\xi(x)$(naive) | 0.275(0.010) | 0.257(0.009) | 0.244(0.009) | 0.232(0.007) | 0.224(0.008) | 0.218(0.005) | 0.209(0.007) | 0.203(0.005) | 0.197(0.004) |
> |  | New | 0.310(0.011) | 0.310(0.009) | 0.313(0.012) | 0.318(0.009) | 0.326(0.010) | 0.338(0.009) | 0.350(0.009) | 0.366(0.008) | 0.393(0.009) |

---

> ### Author Response · Authors · 2025-11-21
>
> ### Table: MIMIC-III (a = c = 0.5, Mean(SD))
> | Metric | Method | b=0.1 | 0.2 | 0.3 | 0.4 | 0.5 | 0.6 | 0.7 | 0.8 | 0.9 |
> |-------|-------|-------|-------|-------|-------|-------|-------|-------|-------|-------|
> | **Accuracy** | $\eta(x)$(proposal) | 0.904(0.001) | 0.902(0.001) | 0.899(0.001) | 0.895(0.001) | 0.892(0.001) | 0.889(0.001) | 0.885(0.001) | 0.881(0.002) | 0.877(0.001) |
> |  | $\xi(x)$(naive) | 0.529(0.006) | 0.679(0.005) | 0.761(0.006) | 0.810(0.006) | 0.837(0.004) | 0.855(0.003) | 0.865(0.004) | 0.871(0.002) | 0.873(0.002) |
> |  | New | 0.900(0.002) | 0.900(0.001) | 0.897(0.001) | 0.895(0.001) | 0.891(0.001) | 0.888(0.001) | 0.884(0.001) | 0.880(0.001) | 0.875(0.001) |
> | **F1** | $\eta(x)$(proposal) | 0.949(0.000) | 0.947(0.001) | 0.945(0.001) | 0.944(0.001) | 0.942(0.001) | 0.939(0.001) | 0.937(0.001) | 0.935(0.001) | 0.932(0.001) |
> |  | $\xi(x)$(naive) | 0.648(0.006) | 0.786(0.004) | 0.850(0.005) | 0.885(0.004) | 0.904(0.003) | 0.916(0.002) | 0.923(0.002) | 0.926(0.002) | 0.927(0.001) |
> |  | New | 0.946(0.001) | 0.946(0.001) | 0.945(0.000) | 0.943(0.001) | 0.941(0.001) | 0.939(0.001) | 0.936(0.001) | 0.934(0.001) | 0.931(0.001) |
>
> ### Table: MIMIC-III (a = b = 0.5,Mean(SD))
> | Metric | Method | c=0.1 | 0.2 | 0.3 | 0.4 | 0.5 | 0.6 | 0.7 | 0.8 | 0.9 |
> |-------|-------|-------|-------|-------|-------|-------|-------|-------|-------|-------|
> | **Accuracy** | $\eta(x)$(proposal) | 0.867(0.001) | 0.873(0.001) | 0.879(0.001) | 0.886(0.001) | 0.892(0.001) | 0.900(0.001) | 0.907(0.001) | 0.915(0.001) | 0.924(0.001) |
> |  | $\xi(x)$(naive) | 0.855(0.003) | 0.854(0.004) | 0.849(0.004) | 0.844(0.004) | 0.837(0.004) | 0.830(0.004) | 0.822(0.004) | 0.814(0.005) | 0.804(0.004) |
> |  | New | 0.864(0.001) | 0.870(0.001) | 0.877(0.001) | 0.884(0.001) | 0.891(0.001) | 0.899(0.001) | 0.906(0.001) | 0.914(0.001) | 0.922(0.001) |
> | **F1** | $\eta(x)$(proposal) | 0.926(0.001) | 0.930(0.001) | 0.933(0.001) | 0.938(0.001) | 0.942(0.001) | 0.946(0.001) | 0.952(0.001) | 0.955(0.001) | 0.960(0.000) |
> |  | $\xi(x)$(naive) | 0.916(0.002) | 0.914(0.003) | 0.911(0.003) | 0.908(0.003) | 0.904(0.003) | 0.900(0.003) | 0.895(0.003) | 0.890(0.003) | 0.885(0.003) |
> |  | New | 0.924(0.001) | 0.928(0.001) | 0.932(0.001) | 0.937(0.001) | 0.941(0.001) | 0.945(0.001) | 0.950(0.001) | 0.954(0.001) | 0.959(0.001) |
>
> > Q1: The paper provides a solid solution to the label–background subpopulation missingness problem in unsupervised domain adaptation. However, the main concern lies in the practicality of the setting and experiments, which limits the overall scope and applicability of the work.
>
> **Response**:
> We respectfully disagree with the reviewer’s concern that the proposed setting is impractical.
>
> **Practicality of the setting**
> - Structured subpopulation missingness is not artificial. It is a well-established and widely documented data collection phenomenon. In many real applications, a specific $(Y,A)$ subgroup is systematically absent from labeled historical datasets but appears in the unlabeled target domain. This occurs in healthcare (new disease–demographic combinations missing in previous cohorts), medical imaging (unseen scanner–pathology pairs), ecology (rare species–environment combinations), and temporal shifts in online platforms (emergent user–behavior patterns). As highlighted in Table 2 of our paper (description is on lines 41–44), the Waterbirds dataset is a concrete, real-world example of this exact issue. Existing UDA methods cannot handle this structured missingness, as they implicitly assume all subpopulations are present in the source domain.
> - The binary case is not a limitation. As discussed in Appendix A.5, our method naturally extends to $n_y>2$ labels and $n_a>2$ background categories. In our reply to Reviewer 3cwJ, we provided a detailed explanation of why we initially focused on $n_y=2$ and $n_a=2$, and how it can be generalized to $n_y>2$ and $n_a>2$.
> - The assumption in equation (3) of the paper is weaker than the standard label shift assumption. Lines 123–136 provide concrete real examples where this assumption naturally holds.
>
> **Practicality experiments**
> - Due to space constraints, in Appendix A.3, to demonstrate the effectiveness of our method, we evaluated its performance on synthetic data; in Appendix A.4, we also provide a performance comparison of other estimators on real-world data.
> - In W3, we added three representative datasets: Waterbirds, CelebA, and MIMIC. Across all of them, our method consistently achieves the best performance, showcasing its practicality and broad applicability.
> - To evaluate settings where conditional invariance (3) may be violated, we introduce a perturbation in the target domain by shifting $\mu_{YA}$ to $\mu_{YA} + 0.2e$, where $e$ is a four-dimensional standard normal vector, keeping all other settings unchanged. The results remain consistent with the main text, demonstrating that the proposed method is not overly sensitive to mild violations of the assumption.

---

> > ### Author Response · Authors · 2025-11-21
> >
> > ### Accuracy Results(Mean(SD))
> >
> > | $n_0$      | Model       | $n_1=1000$     | $2000$        | $3000$        | $4000$        | $5000$        | $6000$        | $7000$        | $8000$        |
> > |------------|--------------|----------------|----------------|----------------|----------------|----------------|----------------|----------------|----------------|
> > | **1000**   | $\eta_1(x)$  | 0.751 (0.052) | 0.762 (0.050) | 0.756 (0.048) | 0.757 (0.045) | 0.755 (0.046) | 0.756 (0.048) | 0.761 (0.047) | 0.757 (0.045) |
> > |            | $\xi_1(x)$   | 0.504 (0.023) | 0.502 (0.023) | 0.498 (0.022) | 0.500 (0.020) | 0.501 (0.022) | 0.502 (0.022) | 0.501 (0.023) | 0.501 (0.023) |
> > |            | $\eta(x)$    | 0.694 (0.041) | 0.697 (0.024) | 0.699 (0.052) | 0.695 (0.039) | 0.698 (0.037) | 0.698 (0.036) | 0.701 (0.035) | 0.693 (0.034) |
> > |            | $\xi(x)$     | 0.629 (0.033) | 0.631 (0.030) | 0.630 (0.030) | 0.627 (0.032) | 0.631 (0.035) | 0.630 (0.031) | 0.634 (0.033) | 0.624 (0.030) |
> > | **6000**   | $\eta_1(x)$  | 0.761 (0.043) | 0.766 (0.044) | 0.762 (0.044) | 0.758 (0.041) | 0.756 (0.047) | 0.749 (0.045) | 0.761 (0.043) | 0.748 (0.049) |
> > |            | $\xi_1(x)$   | 0.502 (0.010) | 0.501 (0.009) | 0.501 (0.009) | 0.501 (0.009) | 0.501 (0.009) | 0.502 (0.009) | 0.501 (0.009) | 0.501 (0.009) |
> > |            | $\eta(x)$    | 0.699 (0.032) | 0.702 (0.032) | 0.702 (0.031) | 0.698 (0.034) | 0.700 (0.033) | 0.694 (0.034) | 0.703 (0.031) | 0.696 (0.036) |
> > |            | $\xi(x)$     | 0.633 (0.026) | 0.631 (0.029) | 0.632 (0.028) | 0.632 (0.028) | 0.633 (0.027) | 0.628 (0.029) | 0.631 (0.026) | 0.630 (0.030) |
> >
> > ### F1 Score Results (Mean(SD) )
> >
> > | $n_0$      | Model     | $n_1=1000$     | $2000$        | $3000$        | $4000$        | $5000$        | $6000$        | $7000$        | $8000$        |
> > |------------|--------------|----------------|----------------|----------------|----------------|----------------|----------------|----------------|----------------|
> > | **1000**   | $\eta_1(x)$  | 0.748 (0.059) | 0.761 (0.055) | 0.755 (0.052) | 0.753 (0.048) | 0.753 (0.052) | 0.753 (0.055) | 0.761 (0.049) | 0.756 (0.048) |
> > |            | $\xi_1(x)$   | 0.670 (0.020) | 0.668 (0.020) | 0.665 (0.019) | 0.666 (0.018) | 0.667 (0.019) | 0.668 (0.020) | 0.667 (0.020) | 0.667 (0.020) |
> > |            | $\eta(x)$    | 0.703 (0.024) | 0.704 (0.023) | 0.702 (0.024) | 0.701 (0.025) | 0.704 (0.027) | 0.703 (0.025) | 0.707 (0.026) | 0.699 (0.023) |
> > |            | $\xi(x)$     | 0.691 (0.049) | 0.693 (0.051) | 0.695 (0.043) | 0.693 (0.054) | 0.693 (0.050) | 0.696 (0.047) | 0.703 (0.043) | 0.688 (0.050) |
> > | **6000**   | $\eta_1(x)$  | 0.761 (0.044) | 0.765 (0.047) | 0.762 (0.047) | 0.758 (0.042) | 0.758 (0.048) | 0.750 (0.045) | 0.762 (0.048) | 0.750 (0.050) |
> > |            | $\xi_1(x)$   | 0.668 (0.008) | 0.668 (0.008) | 0.667 (0.008) | 0.668 (0.008) | 0.668 (0.008) | 0.668 (0.008) | 0.667 (0.008) | 0.667 (0.008) |
> > |            | $\eta(x)$    | 0.706 (0.020) | 0.705 (0.022) | 0.704 (0.020) | 0.705 (0.020) | 0.706 (0.019) | 0.701 (0.021) | 0.704 (0.020) | 0.703 (0.022) |
> > |            | $\xi(x)$     | 0.700 (0.037) | 0.702 (0.041) | 0.702 (0.035) | 0.696 (0.042) | 0.700 (0.038) | 0.690 (0.043) | 0.701 (0.047) | 0.693 (0.047) |
> >
> > ### Accuracy Results(Mean(SD))
> >
> > | $n_1$    | model       | $n_0=1000$     | $2000$        | $3000$        | $4000$        | $5000$        | $6000$        | $7000$        | $8000$        |
> > |----------|------------|----------------|----------------|----------------|----------------|----------------|----------------|----------------|----------------|
> > | ***1000***     | $\eta_1(x)$ | 0.751 (0.052) | 0.748 (0.049) | 0.754 (0.045) | 0.756 (0.042) | 0.762 (0.044) | 0.761 (0.043) | 0.753 (0.043) | 0.757 (0.043) |
> > |                | $\xi_1(x)$  | 0.504 (0.023) | 0.501 (0.009) | 0.502 (0.014) | 0.502 (0.012) | 0.501 (0.010) | 0.502 (0.010) | 0.502 (0.009) | 0.501 (0.008) |
> > |                | $\eta(x)$   | 0.694 (0.041) | 0.696 (0.036) | 0.690 (0.033) | 0.693 (0.030) | 0.702 (0.030) | 0.699 (0.032) | 0.695 (0.035) | 0.694 (0.029) |
> > |                | $\xi(x)$    | 0.629 (0.033) | 0.630 (0.030) | 0.628 (0.026) | 0.626 (0.027) | 0.631 (0.028) | 0.633 (0.026) | 0.634 (0.027) | 0.628 (0.028) |
> > | ***6000***     | $\eta_1(x)$ | 0.756 (0.048) | 0.757 (0.042) | 0.750 (0.041) | 0.761 (0.044) | 0.751 (0.044) | 0.749 (0.045) | 0.748 (0.046) | 0.759 (0.047) |
> > |                | $\xi_1(x)$  | 0.502 (0.022) | 0.502 (0.016) | 0.502 (0.013) | 0.502 (0.011) | 0.502 (0.010) | 0.502 (0.009) | 0.501 (0.008) | 0.501 (0.008) |
> > |                | $\eta(x)$   | 0.698 (0.036) | 0.705 (0.033) | 0.694 (0.031) | 0.702 (0.034) | 0.697 (0.035) | 0.694 (0.034) | 0.692 (0.036) | 0.695 (0.032) |
> > |                | $\xi(x)$    | 0.630 (0.031) | 0.637 (0.029) | 0.633 (0.026) | 0.635 (0.030) | 0.631 (0.026) | 0.628 (0.029) | 0.626 (0.030) | 0.627 (0.031) |

---

> > > ### Author Response · Authors · 2025-11-21
> > >
> > > ### F1 Score Results(Mean(SD) )
> > >
> > > | $n_1$    | model       | $n_0=1000$     | 1000        | 2000        | 3000        | 4000        | 5000        | 6000        | 7000        | 8000        |
> > > |----------------|----------------|----------------|-------------|-------------|-------------|-------------|-------------|-------------|-------------|-------------|
> > > | ***1000***     | $\eta_1(x)$    | 0.748 (0.059) | 0.750 (0.050) | 0.754 (0.048) | 0.757 (0.045) | 0.764 (0.047) | 0.761 (0.044) | 0.753 (0.046) | 0.758 (0.045) |
> > > || $\xi_1(x)$     | 0.670 (0.020) | 0.667 (0.008) | 0.668 (0.013) | 0.668 (0.010) | 0.668 (0.009) | 0.668 (0.008) | 0.668 (0.008) | 0.668 (0.007) |
> > > || $\eta(x)$      | 0.703 (0.024) | 0.703 (0.022) | 0.701 (0.019) | 0.700 (0.019) | 0.704 (0.020) | 0.706 (0.020) | 0.706 (0.021) | 0.702 (0.020) |
> > > || $\xi(x)$       | 0.691 (0.049) | 0.693 (0.047) | 0.682 (0.044) | 0.690 (0.040) | 0.697 (0.041) | 0.700 (0.037) | 0.691 (0.049) | 0.693 (0.038) |
> > > |***6000***| $\eta_1(x)$    | 0.753 (0.055) | 0.757 (0.043) | 0.750 (0.041) | 0.760 (0.045) | 0.752 (0.044) | 0.750 (0.045) | 0.748 (0.050) | 0.758 (0.049) |
> > > || $\xi_1(x)$     | 0.668 (0.020) | 0.669 (0.014) | 0.669 (0.011) | 0.668 (0.010) | 0.668 (0.008) | 0.668 (0.008) | 0.668 (0.007) | 0.668 (0.007) |
> > > || $\eta(x)$       | 0.703 (0.025) | 0.709 (0.022) | 0.705 (0.019) | 0.707 (0.020) | 0.705 (0.020) | 0.701 (0.021) | 0.701 (0.022) | 0.702 (0.022) |
> > > || $\xi(x)$      | 0.696 (0.047) | 0.703 (0.046) | 0.691 (0.042) | 0.699 (0.038) | 0.695 (0.043) | 0.690 (0.043) | 0.691 (0.047) | 0.697 (0.036) |

---

> > > > ### Comment · Reviewer_xz9M · 2025-11-25
> > > >
> > > > I thank the authors for their efforts in addressing my concerns.
> > > >
> > > > Whether the setting or assumptions are practical is largely subjective, and—as also noted by other reviewers—remains debatable. I would like to respectfully maintain my initial opinion.
> > > >
> > > > On the empirical side, I appreciate that the authors incorporated new experiments in W3 based on my suggestions. However, I now have a question regarding what the “New” method specifically refers to. Moreover, the additional experiments on MIMIC-III indicate that the proposed method does not outperform the existing “New” general label-shift approach. To strengthen the empirical justification, more “New” baselines should also be included in the comparison.

---

> > > > > ### Author Response · Authors · 2025-12-02
> > > > >
> > > > > **Response**
> > > > >
> > > > > - We sincerely thank the reviewer for the follow-up comments. We provide additional clarification below.
> > > > >
> > > > > **Practical relevance of the setting and assumptions**
> > > > > - Our setting is motivated by real-world scenarios in which a structured  $(Y,A)$ subgroup is systematically missing from the labeled data—a phenomenon that appears in medical, ecological, and vision datasets but has not been studied within a unified statistical framework. Table 2 provides two examples, and similar cases are widely observed in practice, where the missingness is tied to interpretable factors such as demographics, environments rather than occurring at random. Our conditional invariance assumption is designed precisely to handle such structured missingness. Unlike classical label-shift or covariate-shift assumptions(see lines 123-136), we require only that the structured feature-generation mechanism $p(X∣Y,A)$ remains stable, while allowing substantial variation in $p(A)$, $p(Y∣A)$, and $p(X∣A)$. This assumption is often reasonable in practice—for example, in medical diagnostics, feature measurements are primarily determined by disease status and demographic variables, even though disease prevalence and population composition may differ across datasets. Thus, the conditional invariance assumption is not intended to claim universal validity; rather, it captures a broad class of realistic data-generation mechanisms that lie between classical shift models and arise frequently in real applications.
> > > > >
> > > > > **Clarification of the “New” method**
> > > > > - The method labeled “New” in our tables corresponds to a standard unsupervised domain adaptation approach that relies solely on the label-shift assumption and without considering the missing subgroup. This was previously described in our W2 response, and we will clarify the labeling further in the revised manuscript.
> > > > >
> > > > > **Results on the MIMIC-III dataset**
> > > > > - On MIMIC-III, the naive baseline performs the worst, as expected. Both the “New” method and our proposed method perform well, with our method showing a slight numerical advantage. This is consistent with the fact that MIMIC-III is closer to a label-shift regime, under which both approaches are applicable and tend to behave similarly. The experimental results on these three datasets further demonstrate that our method naturally adapts to this special case while maintaining validity in the more general setting involving a missing structured subgroup.

---

### Official Review · Reviewer_Btck · 2025-11-03

**Soundness:** 3
**Presentation:** 3
**Contribution:** 2
**Rating:** 4
**Confidence:** 3

**Summary:**

The paper provides a theoretical framework for Unsupervised Domain Adaptation (UDA) that specifically accounts for the scenario where an entire subpopulation defined by the binary background A is unobservable in the source domain.

**Strengths:**

1. Presents a robust theoretical framework with strong theoretical guarantees and generalization bounds. I have not verified the correctness in depth.

**Weaknesses:**

1. It is a niche problem under unsupervised domain adaptation limiting its scope and usefulness. This is the major drawback.
2. Inadequate experimental results - just 1 page are insufficient to evaluate the effectiveness of this model.
3. The reviewer is only concerned about the usefulness of the proposed approach.

**Questions:**

None.

---

> ### Author Response · Authors · 2025-11-21
>
> >W1: It is a niche problem under unsupervised domain adaptation limiting its scope and usefulness. This is the major drawback.
>
> **Response**
> - We respectfully disagree with the assessment that our setting is a niche problem. Structured subpopulation missingness is not a niche scenario. In many real applications, a specific $(Y,A)$ subgroup is systematically missing from labeled historical data but appears in the unlabeled target domain. This occurs in healthcare (new disease–demographic groups not included in past datasets), medical imaging (new scanner–pathology combinations), ecology (rare species–environment pairs), and temporal shifts in online platforms (new user–behavior patterns). Existing UDA and label-shift methods assume complete source support and therefore cannot handle such cases. Our framework directly addresses these practically common situations, making the problem both relevant and impactful rather than narrow in scope.
>
> >W2: Inadequate experimental results - just 1 page are insufficient to evaluate the effectiveness of this model.
>
> **Response**
> - We respectfully disagree with judging our experimental results as "inadequate" solely based on the number of pages reported in the main paper.
> - Due to the space constraint, in Appendix A.3, to demonstrate the effectiveness of our method, we evaluated its performance on synthetic data; in Appendix A.4, we also provide a performance comparison of other estimators on real-world data.
> - Below we provide more experimental results: we now compare the proposed method to a standard UDA method based solely on the label shift assumption without considering the missing subgroup. We simply denote it as "New". We evaluate all three methods (proposed, naive benchmark, and "New") on three representative datasets: Waterbirds, CelebA, and MIMIC. Results show that while "New" outperforms the naive benchmark $\xi(x)$, our proposal consistently achieves the best performance, validating its effectiveness.
>
> ### Table: Waterbirds (a = c = 0.5, Mean(SD))
> | Metric | Method | b=0.1 | 0.2 | 0.3 | 0.4 | 0.5 | 0.6 | 0.7 | 0.8 | 0.9 |
> |-------|-------|-------|-------|-------|-------|-------|-------|-------|-------|-------|
> | **Accuracy** | $\eta(x)$(proposal) | 0.841(0.003) | 0.845(0.003) | 0.848(0.003) | 0.850(0.003) | 0.851(0.003) | 0.853(0.003) | 0.856(0.003) | 0.859(0.003) | 0.863(0.003) |
> |  | $\xi(x)$(naive) | 0.741(0.013) | 0.769(0.011) | 0.786(0.009) | 0.796(0.007) | 0.803(0.007) | 0.811(0.008) | 0.816(0.007) | 0.820(0.006) | 0.824(0.005) |
> |  | New | 0.799(0.015) | 0.806(0.012) | 0.812(0.008) | 0.815(0.007) | 0.819(0.008) | 0.823(0.007) | 0.826(0.008) | 0.828(0.007) | 0.831(0.005) |
> | **F1** | $\eta(x)$(proposal) | 0.790(0.005) | 0.796(0.003) | 0.798(0.004) | 0.799(0.004) | 0.797(0.004) | 0.796(0.003) | 0.797(0.004) | 0.797(0.004) | 0.798(0.004) |
> |  | $\xi(x)$(naive) | 0.450(0.042) | 0.532(0.032) | 0.574(0.025) | 0.594(0.019) | 0.605(0.018) | 0.617(0.020) | 0.621(0.019) | 0.624(0.016) | 0.625(0.013) |
> |  | New | 0.640(0.038) | 0.655(0.030) | 0.665(0.022) | 0.669(0.017) | 0.671(0.018) | 0.674(0.016) | 0.672(0.018) | 0.668(0.017) | 0.665(0.013) |
>
> ### Table: Waterbirds (a = b = 0.5, Mean(SD))
> | Metric | Method | c=0.1 | 0.2 | 0.3 | 0.4 | 0.5 | 0.6 | 0.7 | 0.8 | 0.9 |
> |-------|-------|-------|-------|-------|-------|-------|-------|-------|-------|-------|
> | **Accuracy** | $\eta(x)$(proposal) | 0.869(0.002) | 0.870(0.003) | 0.866(0.001) | 0.860(0.002) | 0.851(0.003) | 0.840(0.004) | 0.826(0.005) | 0.809(0.005) | 0.788(0.006) |
> |  | $\xi(x)$(naive) | 0.801(0.006) | 0.816(0.006) | 0.818(0.008) | 0.813(0.008) | 0.803(0.007) | 0.787(0.007) | 0.765(0.006) | 0.737(0.007) | 0.703(0.008) |
> |  | New | 0.796(0.006) | 0.814(0.006) | 0.821(0.007) | 0.821(0.008) | 0.819(0.008) | 0.813(0.009) | 0.805(0.009) | 0.796(0.010) | 0.786(0.011) |
> | **F1** | $\eta(x)$(proposal) | 0.768(0.004) | 0.780(0.004) | 0.787(0.002) | 0.793(0.003) | 0.797(0.004) | 0.801(0.004) | 0.805(0.006) | 0.809(0.005) | 0.815(0.005) |
> |  | $\xi(x)$(naive) | 0.519(0.015) | 0.557(0.021) | 0.582(0.025) | 0.596(0.023) | 0.605(0.018) | 0.608(0.017) | 0.607(0.013) | 0.605(0.015) | 0.606(0.016) |
> |  | New | 0.531(0.015) | 0.581(0.020) | 0.620(0.020) | 0.647(0.021) | 0.671(0.018) | 0.689(0.019) | 0.710(0.017) | 0.730(0.017) | 0.753(0.017) |

---

> ### Author Response · Authors · 2025-11-21
>
> ### Table: CeleA (a = c = 0.5, Mean(SD))
> | Metric | Method | b=0.1 | 0.2 | 0.3 | 0.4 | 0.5 | 0.6 | 0.7 | 0.8 | 0.9 |
> |-------|-------|-------|-------|-------|-------|-------|-------|-------|-------|-------|
> | **Accuracy** | $\eta(x)$(proposal) | 0.655(0.004) | 0.662(0.005) | 0.667(0.004) | 0.671(0.004) | 0.675(0.004) | 0.681(0.004) | 0.690(0.003) | 0.698(0.003) | 0.708(0.003) |
> |  | $\xi(x)$(naive) | 0.395(0.002) | 0.412(0.003) | 0.419(0.003) | 0.420(0.003) | 0.422(0.003) | 0.421(0.003) | 0.419(0.003) | 0.416(0.002) | 0.412(0.002) |
> |  | New | 0.449(0.003) | 0.455(0.004) | 0.456(0.005) | 0.455(0.005) | 0.454(0.004) | 0.452(0.003) | 0.448(0.004) | 0.444(0.004) | 0.439(0.003) |
> |**F1**| $\eta(x)$(proposal) | 0.756(0.003) | 0.762(0.003) | 0.766(0.003) | 0.768(0.002) | 0.771(0.002) | 0.774(0.002) | 0.779(0.002) | 0.783(0.004) | 0.788(0.002) |
> |  | $\xi(x)$(naive) | 0.170(0.006) | 0.212(0.009) | 0.224(0.009) | 0.224(0.007) | 0.224(0.008) | 0.215(0.007) | 0.203(0.008) | 0.189(0.006) | 0.172(0.005) |
> |  | New | 0.324(0.010) | 0.335(0.012) | 0.337(0.013) | 0.331(0.013) | 0.326(0.010) | 0.316(0.006) | 0.302(0.010) | 0.287(0.009) | 0.270(0.005) |
>
> ### Table: CeleA (a = b = 0.5, Mean(SD))
> | Metric | Method | c=0.1 | 0.2 | 0.3 | 0.4 | 0.5 | 0.6 | 0.7 | 0.8 | 0.9 |
> |-------|-------|-------|-------|-------|-------|-------|-------|-------|-------|-------|
> | **Accuracy** | $\eta(x)$(proposal) | 0.659(0.003) | 0.657(0.002) | 0.659(0.003) | 0.664(0.003) | 0.675(0.004) | 0.697(0.004) | 0.730(0.003) | 0.776(0.001) | 0.835(0.001) |
> |  | $\xi(x)$(naive) | 0.533(0.003) | 0.509(0.003) | 0.483(0.003) | 0.454(0.003) | 0.422(0.003) | 0.386(0.002) | 0.344(0.002) | 0.297(0.002) | 0.242(0.002) |
> |  | New | 0.540(0.003) | 0.520(0.003) | 0.500(0.004) | 0.478(0.003) | 0.454(0.004) | 0.429(0.004) | 0.401(0.004) | 0.371(0.004) | 0.344(0.005) |
> | **F1** | $\eta(x)$(proposal) | 0.669(0.003) | 0.691(0.002) | 0.716(0.002) | 0.743(0.002) | 0.771(0.002) | 0.802(0.002) | 0.835(0.002) | 0.870(0.001) | 0.909(0.001) |
> |  | $\xi(x)$(naive) | 0.275(0.010) | 0.257(0.009) | 0.244(0.009) | 0.232(0.007) | 0.224(0.008) | 0.218(0.005) | 0.209(0.007) | 0.203(0.005) | 0.197(0.004) |
> |  | New | 0.310(0.011) | 0.310(0.009) | 0.313(0.012) | 0.318(0.009) | 0.326(0.010) | 0.338(0.009) | 0.350(0.009) | 0.366(0.008) | 0.393(0.009) |
>
> ### Table: MIMIC-III (a = c = 0.5, Mean(SD))
> | Metric | Method | b=0.1 | 0.2 | 0.3 | 0.4 | 0.5 | 0.6 | 0.7 | 0.8 | 0.9 |
> |-------|-------|-------|-------|-------|-------|-------|-------|-------|-------|-------|
> | **Accuracy** | $\eta(x)$(proposal) | 0.904(0.001) | 0.902(0.001) | 0.899(0.001) | 0.895(0.001) | 0.892(0.001) | 0.889(0.001) | 0.885(0.001) | 0.881(0.002) | 0.877(0.001) |
> |  | $\xi(x)$(naive) | 0.529(0.006) | 0.679(0.005) | 0.761(0.006) | 0.810(0.006) | 0.837(0.004) | 0.855(0.003) | 0.865(0.004) | 0.871(0.002) | 0.873(0.002) |
> |  | New | 0.900(0.002) | 0.900(0.001) | 0.897(0.001) | 0.895(0.001) | 0.891(0.001) | 0.888(0.001) | 0.884(0.001) | 0.880(0.001) | 0.875(0.001) |
> | **F1** | $\eta(x)$(proposal) | 0.949(0.000) | 0.947(0.001) | 0.945(0.001) | 0.944(0.001) | 0.942(0.001) | 0.939(0.001) | 0.937(0.001) | 0.935(0.001) | 0.932(0.001) |
> |  | $\xi(x)$(naive) | 0.648(0.006) | 0.786(0.004) | 0.850(0.005) | 0.885(0.004) | 0.904(0.003) | 0.916(0.002) | 0.923(0.002) | 0.926(0.002) | 0.927(0.001) |
> |  | New | 0.946(0.001) | 0.946(0.001) | 0.945(0.000) | 0.943(0.001) | 0.941(0.001) | 0.939(0.001) | 0.936(0.001) | 0.934(0.001) | 0.931(0.001) |
>
> ### Table: MIMIC-III (a = b = 0.5,Mean(SD))
> | Metric | Method | c=0.1 | 0.2 | 0.3 | 0.4 | 0.5 | 0.6 | 0.7 | 0.8 | 0.9 |
> |-------|-------|-------|-------|-------|-------|-------|-------|-------|-------|-------|
> | **Accuracy** | $\eta(x)$(proposal) | 0.867(0.001) | 0.873(0.001) | 0.879(0.001) | 0.886(0.001) | 0.892(0.001) | 0.900(0.001) | 0.907(0.001) | 0.915(0.001) | 0.924(0.001) |
> |  | $\xi(x)$(naive) | 0.855(0.003) | 0.854(0.004) | 0.849(0.004) | 0.844(0.004) | 0.837(0.004) | 0.830(0.004) | 0.822(0.004) | 0.814(0.005) | 0.804(0.004) |
> |  | New | 0.864(0.001) | 0.870(0.001) | 0.877(0.001) | 0.884(0.001) | 0.891(0.001) | 0.899(0.001) | 0.906(0.001) | 0.914(0.001) | 0.922(0.001) |
> | **F1** | $\eta(x)$(proposal) | 0.926(0.001) | 0.930(0.001) | 0.933(0.001) | 0.938(0.001) | 0.942(0.001) | 0.946(0.001) | 0.952(0.001) | 0.955(0.001) | 0.960(0.000) |
> |  | $\xi(x)$(naive) | 0.916(0.002) | 0.914(0.003) | 0.911(0.003) | 0.908(0.003) | 0.904(0.003) | 0.900(0.003) | 0.895(0.003) | 0.890(0.003) | 0.885(0.003) |
> |  | New | 0.924(0.001) | 0.928(0.001) | 0.932(0.001) | 0.937(0.001) | 0.941(0.001) | 0.945(0.001) | 0.950(0.001) | 0.954(0.001) | 0.959(0.001) |

---

> ### Author Response · Authors · 2025-11-21
>
> >W3: The reviewer is only concerned about the usefulness of the proposed approach.
>
> **Response**
>
> - Sorry, we do not quite understand this *weakness*. Are you **only** concerning the *usefullness* of our work? If your indication is that our work is not useful, we respectfully disagree.
> - The structured subpopulation-missing scenario we study is not an artificial or rare edge case—it appears frequently in real applications where certain $(Y,A)$ combinations are systematically absent in labeled data but present in deployment. This happens in healthcare (new disease–demographic subgroups not included in historical training data), medical imaging (new scanner–pathology combinations), wildlife monitoring (rare species–environment pairs), and evolving online systems (emerging user–behavior patterns). Existing UDA and label-shift methods fail precisely in these settings because they require complete source support. Our method introduces a novel framework that enables effective prediction even in the presence of pervasive structural uncertainty. Therefore, the proposed approach is not only practical but also addresses an important and previously unexplored class of real-world domain adaptation problems.

---

### Note · Authors · 2025-12-02

I have read and agree with the venue's withdrawal policy on behalf of myself and my co-authors.